

# The representation of solar cycle signals in stratospheric ozone – Part 1: A comparison of satellite observations

A. Maycock[1,2], K. Matthes[3,4], S. Tegtmeier[3], R. Thiéblemont[3], and L. Hood[5]

[1]Centre for Atmospheric Science, University of Cambridge, Cambridge, UK
[2]National Centre for Atmospheric Science, UK
[3]GEOMAR Helmholtz for Ocean Research, Kiel, Germany
[4]Christian-Albrechts Universität zu Kiel, Kiel, Germany
[5]Lunar and Planetary Laboratory, University of Arizona, Tucson, Arizona, USA

Received: 27 October 2015 – Accepted: 1 December 2015 – Published: 15 January 2016

Correspondence to: A. Maycock (acm204@cam.ac.uk)

Published by Copernicus Publications on behalf of the European Geosciences Union.

**ACPD**

doi:10.5194/acp-2015-882

**Solar cycle signals in stratospheric ozone – Part 1: Satellite observations**

A. Maycock et al.

Title Page

| Abstract | Introduction |
| Conclusions | References |
| Tables | Figures |

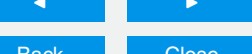

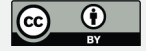

# Abstract

The impact of changes in incoming solar ultraviolet irradiance on stratospheric ozone forms an important part of the climate response to solar variability. To realistically simulate the climate response to solar variability using climate models, a minimum requirement is that they should include a solar cycle ozone component that has a realistic amplitude and structure, and which varies with season. For climate models that do not include interactive ozone chemistry, this component must be derived from observations and/or chemistry–climate model simulations and included in an externally prescribed ozone database that also includes the effects of all major external forcings. Part 1 of this two part study presents the solar-ozone responses in a number of updated satellite datasets for the period 1984–2004, including the Stratospheric Aerosol and Gas Experiment (SAGE) II version 6.2 and version 7.0 data, and the Solar Backscatter Ultraviolet Instrument (SBUV) version 8.0 and version 8.6 data. A number of combined datasets, which have extended SAGE II using more recent satellite measurements, are also analysed for the period 1984–2011. It is shown that SAGE II derived solar-ozone signals are sensitive to the independent temperature measurements used to convert ozone number density to mixing ratio units. A change in these temperature measurements in the recent SAGE II v7.0 data leads to substantial differences in the mixing ratio solar-ozone response compared to the previous v6.2, particularly in the tropical upper stratosphere. We also show that alternate satellite ozone datasets have issues (e.g., sparse spatial and temporal sampling, low vertical resolution, and shortness of measurement record), and that the methods of accounting for instrument offsets and drifts in merged satellite datasets can have a substantial impact on the solar cycle signal in ozone. For example, the magnitude of the solar-ozone response varies by around a factor of two across different versions of the SBUV VN8.6 record, which appears to be due to the methods used to combine the separate SBUV timeseries. These factors make it difficult to extract more than an annual-mean solar-ozone response from the available satellite observations. It is therefore unlikely that satellite ozone measure-

## ACPD

doi:10.5194/acp-2015-882

## Solar cycle signals in stratospheric ozone – Part 1: Satellite observations

A. Maycock et al.

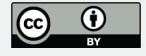

ments alone can be applied to estimate the necessary solar cycle ozone component of the prescribed ozone database for future coupled model intercomparison projects (e.g., CMIP6).

# 1 Introduction

Whilst changes in total solar irradiance (TSI) between the maximum and minimum phases of the approximately 11 year solar cycle are known to be small ($< 0.1\,\%$), there is enhanced variability in the ultraviolet (UV) spectral region ($> 6\,\%$) (e.g. Ermolli et al., 2013). An increase in UV irradiance impacts stratospheric heating rates, and thus temperatures, through two main mechanisms: (1) enhanced absorption of radiation by ozone, and (2) enhanced production of ozone through the photolysis of oxygen at wavelengths less than $\sim 242\,nm$. Consistent with these mechanisms, past studies using models, reanalysis data and observations have identified an increase in annual mean temperature in the upper stratosphere of up to $\sim 1.5\,K$ between solar maximum and minimum (e.g. Ramaswamy et al., 2001; Austin et al., 2008; Mitchell et al., 2015a), and an increase in ozone abundances of a few percent (Haigh, 1994; Soukharev and Hood, 2006). These radiative changes modify the meridional temperature gradients in the upper stratosphere, which can lead to a modulation of planetary wave propagation and breaking, and changes in the strength of the stratospheric polar vortex (e.g. Kuroda and Kodera, 2002; Matthes et al., 2004, 2006; Gray et al., 2010; Ineson et al., 2011). Such feedback mechanisms can lead to amplified changes in regional surface climate via stratosphere–troposphere dynamical coupling (e.g. Gray et al., 2010). Constraining the stratospheric temperature response to solar forcing is therefore important for understanding solar–climate coupling and sources of decadal variability in the climate system (e.g. Thiéblemont et al., 2015).

Recent analysis of simulations from the fifth phase of the Coupled Model Intercomparison Project (CMIP5) as part of the Stratospheric and Tropospheric Processes and their Role in Climate (SPARC) Solar Model Intercomparison Project (SolarMIP) un-

Discussion Paper | Discussion Paper | Discussion Paper | Discussion Paper | Discussion Paper |

**ACPD**

doi:10.5194/acp-2015-882

**Solar cycle signals in stratospheric ozone – Part 1: Satellite observations**

A. Maycock et al.

**ACPD**

doi:10.5194/acp-2015-882

## Solar cycle signals in stratospheric ozone – Part 1: Satellite observations

A. Maycock et al.

der the WCRP/SPARC SOLARIS-HEPPA activity has shown that the peak temperature response to the 11 year solar cycle near the tropical stratopause ($\sim 1$ hPa) ranges from 0.3–1.2 K across stratosphere resolving "high-top" models (Mitchell et al., 2015b). Since the majority of these models used the same spectral solar irradiance dataset (Wang et al., 2005), the reasons for this spread are likely to include differences in the models' shortwave radiation schemes (Nissen et al., 2007; Forster et al., 2011), and/or whether they prescribe or interactively simulate stratospheric ozone (Hood et al., 2015). For models which include interactive chemistry, the details of their photolysis schemes will also be important for simulating the impact of solar variability on ozone. The CMIP5 models also show considerable differences in their high latitude dynamical responses to solar variability (Mitchell et al., 2015b), and it is not yet clear what role the "direct" (i.e. irradiance + ozone) part of the response to solar forcing may play in this spread.

Whilst past studies have identified solar-ozone signals in observations (e.g. Soukharev and Hood, 2006; Randel and Wu, 2007; Remsberg and Lingenfelser, 2010; Remsberg, 2014), there are differences in the magnitudes and structures between individual satellite records. It is not clear whether these are due to inter-instrument differences in observational periods and/or differences in instrument resolution, sampling or drifts. There are also differences in the structure and magnitude of the solar-ozone responses between observations and atmospheric models with interactive chemistry (e.g. Haigh, 1994; Soukharev and Hood, 2006; Austin et al., 2008; Dhomse et al., 2011). These issues are compounded by uncertainties in the characteristics of spectral solar irradiance variability (e.g. Ermolli et al., 2013), which have implications for the magnitude and structure of the solar-ozone response because of its dependence on photochemical processes (Haigh et al., 2010; Dhomse et al., 2015). These factors present a challenge for understanding and evaluating the climate response to solar variability, particularly since dynamical feedbacks may amplify the effects of an initially small forcing (e.g. Matthes et al., 2006).

The focus of this two part study (see also Maycock et al., 2015) is on the representation of the solar-ozone signal in observations and global models. The goal is to syn-

thesise current knowledge to inform a recommendation for including the solar-ozone signal in the prescribed ozone dataset being created for CMIP6 (Hegglin et al., 2015); one motivation is to improve upon the approach adopted for CMIP5 (Cionni et al., 2011). The CMIP5 ozone database provided monthly mean global ozone profiles at

5 a horizontal resolution of $5° \times 5°$ (lon/lat) on 24 pressure levels covering 1000–1 hPa for the period 1850–2100 (Cionni et al., 2011). Data were given on the following levels: 1000, 850, 700, 600, 500, 400, 300, 250, 200, 150, 100, 80, 70, 50, 30, 20, 15, 10, 7, 5, 3, 2, 1.5, 1 hPa. Stratospheric ozone data (at $p \leq 300$ hPa) were given as zonal mean values. An ozone dataset for CMIP6 will have similar specifications, but is

10 likely to extend to lower pressure ($\sim 0.01$ hPa) and may be defined for higher temporal and spatial resolutions (M. Hegglin, personal communication, 2015). Therefore a description of a solar-ozone signal derived from observations and/or chemistry–climate models must fulfil these criteria (i.e. global coverage at monthly resolution and sufficient vertical resolution throughout the stratosphere).

The present Part 1 of the study focuses on the solar-ozone responses in current satellite datasets. A number of long-term satellite ozone datasets previously used to analyse solar-ozone signals (e.g. Soukharev and Hood, 2006) have recently been updated. We therefore take the opportunity to describe the solar-ozone responses in these revised datasets and compare them to previous versions. In particular, we

present the latest version 7.0 of the Stratospheric Aerosol and Gas Experiment (SAGE) II data and compare it to the former version 6.2 data, which has been used in several solar–climate studies (e.g. Soukharev and Hood, 2006; Gray et al., 2009). A number of merged satellite datasets, which extend SAGE II using more recent records, have also been created and recently analysed as part of the SPARC SI[2]N ozone trends activity

(e.g. Tummon et al., 2015); here we analyse the solar-ozone signals in five such combined satellite datasets. We also present the updated VN8.6 of the Solar Backscatter Ultraviolet Instrument (SBUV) data and compare this to the former VN8.0 data.

Part 2 of this study (Maycock et al., 2015) describes the solar-ozone responses in simulations of the recent past from the Chemistry-Climate Model Validation exer-

**ACPD**

doi:10.5194/acp-2015-882

**Solar cycle signals in stratospheric ozone – Part 1: Satellite observations**

A. Maycock et al.

Discussion Paper | Discussion Paper | Discussion Paper | Discussion Paper |

cise (CCMVal-2) (SPARC CCMVal, 2010) and compares them to the observations described here. Part 2 also discusses the representation of the solar-ozone response in the ozone dataset used for CMIP5 models without interactive chemistry (Cionni et al., 2011). The results from Parts 1 and 2 are synthesised to make a final recommendation for the CMIP6 ozone dataset, which is presented at the end of Part 2.

## 2   Methods

### 2.1   Ozone datasets

The ozone datasets examined in this study are summarised in Table 1. Unless otherwise stated, the analysis in Sect. 3 focuses on the 21 year period January 1984 to December 2004, which is a sampling period approximately common to most of the datasets considered. A detailed overview of the satellite records, their spatial and temporal sampling characteristics and, where appropriate, their merging procedures is provided by Soukharev and Hood (2006), Tummon et al. (2015), and the references listed below. Here we briefly summarise their main properties.

### 2.1.1   SAGE II based records

The SAGE II record forms the basis of many long-term ozone datasets (see e.g. Tummon et al., 2015). As a limb-viewing instrument, the spatial and temporal sampling of SAGE is fairly sparse, with a given latitude measured approximately once per month; however, it is recognised as having good long-term stability and a vertical resolution of $\sim 1\,\mathrm{km}$ in the stratosphere, which are both key for studying the structures of decadal and multi-decadal variations in ozone. Data are available from October 1984 to August 2005. We use zonal and monthly mean ozone data provided through the SPARC Data Inititive (SDI) (Tegtmeier et al., 2013). The native retrieval of SAGE II is in units of number density on altitude levels; the data are post-processed to volume mixing ratios (vmr) on pressure surfaces (see https://eosweb.larc.nasa.gov/project/sage2/sage2_table).

**ACPD**

doi:10.5194/acp-2015-882

**Solar cycle signals in stratospheric ozone – Part 1: Satellite observations**

A. Maycock et al.

Discussion Paper | Discussion Paper | Discussion Paper | Discussion Paper

**ACPD**

doi:10.5194/acp-2015-882

**Solar cycle signals in stratospheric ozone – Part 1: Satellite observations**

A. Maycock et al.

As a solar occultation instrument, all SAGE II profiles can be categorised as a sunrise (SR) or sunset (SS) measurements. There are known SR/SS sampling biases associated with the SAGE II retrievals, which impact on the estimates of climatological ozone values (Toohey et al., 2013), but could also affect temporal variability. For example, SAGE II obtained profiles in two narrow latitude bands each day, 15 each at sunrise and sunset, but after November 2000 SAGE II measured only one profile per orbit at either SR or SS. Therefore the SR/SS sampling may impact differently on the formation of monthly mean data as a function of time through the record. However, the results for the solar-ozone signal in SAGE II data presented in Sect. 3.1 are found to be similar for the period 1984–1999, which excludes the post-2000 period that includes markedly different SR/SS sampling. Therefore the results for SAGE II also focus on the 1984–2004 period.

The representation of the solar-ozone response in SAGE II version 6.2 data has been discussed in a number of studies: e.g. Randel and Wu (2007), Soukharev and Hood (2006), and Gray et al. (2009) for mixing ratios, and Remsberg and Lingenfelser (2010) for number densities. The SAGE II retrieval algorithm was recently updated as part of the version 7.0 release (Damadeo et al., 2013). Here this most recent release is compared to the previous version 6.2 in both number density and mixing ratio units. This comparison is important because the temperature dataset used to convert SAGE II from its native units of number density on altitude levels to mixing ratios on pressure levels was changed from National Meteorological Center/National Center for Environmental Prediction (NMC/NCEP) to Modern Era-Retrospective Analysis for Research and Applications (MERRA) reanalysis data.

The SAGE II mission stopped measuring in 2005. Since then several satellites have continued to measure ozone, and there are now a number of combined datasets that have extended the SAGE II record to near the present day using recent measurements. Here we analyse five such extended datasets that have been recently analysed as part of the SPARC SI$^2$N assessment of long-term stratospheric ozone trends, and which are described in detail by Tummon et al. (2015) and the references given below.

Discussion Paper | Discussion Paper | Discussion Paper | Discussion Paper

All of the extended ozone datasets considered here include SAGE II v7.0 vmr data, with the exception of the Global OZone Chemistry And Related trace gas Data records for the Stratosphere (GOZCARDS) dataset, which uses SAGE II v6.2 (Froidevaux et al., 2015). Therefore the results in Sect. 3.1 about differences in the solar-ozone signals in versions of SAGE II should be kept in mind for some of the comparisons of the extended datasets. Differences between the combined datasets are also likely to arise from the data sources used to extend SAGE and from how the various satellite records are merged.

Two datasets are analysed that extend SAGE II using GOMOS (Global Ozone Monitoring by Occultation of Stars), which flew on the ENVISAT satellite and covers 2002–2012. Two combined SAGE-GOMOS datasets have been constructed so far, which take different approaches for combining the two records. Kyrölä et al. (2015) use GOMOS as a reference and adjust SAGE II sunrise and sunset profiles separately at each latitude and altitude; this dataset will be referred to as SAGE-GOMOS 1. Conversely, Penckwitt et al. (2015) use SAGE II as a reference and adjust GOMOS data using seasonally-varying offsets at each latitude and altitude; this dataset will be referred to as SAGE-GOMOS 2.

Another dataset is analysed which extends SAGE II with OSIRIS (Optical Spectrograph and Infrared Imager System) data covering 2001-present (Bourassa et al., 2014; Sioris et al., 2014). Latitude and altitude dependent offsets are calculated for the deseasonalised data during the overlap period (January 2002–August 2005), and the OSIRIS data are adjusted to produce a consistent combined SAGE II and OSIRIS timeseries.

Two datasets that are comprised of more than two satellite records are also analysed. The SWOOSH (Stratospheric Water and OzOne Satellite Homogenized) record includes SAGE II (v7.0), SAGE III (2002–2005), HALOE (1991–2005), UARS MLS (1991–1999), and Aura MLS (2004 onwards), with Aura MLS used as a reference from which offsets for the other records are calculated (Davis et al., 2015). Finally, GOZCARDS includes data from SAGE I (1979–1982), HALOE, UARS MLS, Aura MLS, and ACE-FTS (2003 onwards), in addition to the SAGE II v6.2 data, which is used as a refer-

**ACPD**

doi:10.5194/acp-2015-882

**Solar cycle signals in stratospheric ozone – Part 1: Satellite observations**

A. Maycock et al.

ence to which the other records are adjusted (Froidevaux et al., 2015). The solar-ozone signals in these five extended records are analysed for the period 1984–2011.

### 2.1.2 SBUV based records

In addition to SAGE II, the other main long-term internally-calibrated satellite ozone dataset is comprised of data from the Solar Backscatter Ultraviolet Radiometer (SBUV) instrument on the Nimbus satellite and the SBUV/2 instruments on various National Oceanic and Atmospheric Administration (NOAA) satellites. Data are available as mixing ratios on pressure levels from January 1970 to near the present day. As nadir-viewing instruments, SBUV has more frequent global coverage than the limb-viewing SAGE II, but its vertical resolution is at least an order of magnitude poorer at pressures greater than $\sim$ 15 hPa rendering it more difficult to resolve detailed ozone structures in the mid and lower stratosphere. Since SBUV is comprised of multiple separate records from different satellites, inter-instrument biases and drifts must also be accounted for.

We analyse zonal and monthly mean data from the longstanding SBUV version 8.0 (VN8.0) dataset and the latest SBUV VN8.6 (McPeters et al., 2013; Bhartia et al., 2013), thereby complementing the analysis of SBUV VN8.0 by Soukharev and Hood (2006). The SBUV VN8.0 was accessed from http://acd-ext.gsfc.nasa.gov/Data_services/merged/data/sbuv.70-09.za.v8_prof.vmr.rev1.txt. Two versions of the VN8.6 record have been produced so far: the SBUV Merged Ozone Dataset (SBUVMOD) VN8.6 dataset from NASA (Frith et al., 2014) and the SBUV Merged Cohesive dataset from NOAA (Wild and Long, 2015). These are identical to the datasets analysed as part of the SI$^2$N ozone trend activity (e.g. Tummon et al., 2015). The SBUV VN8.6 datasets differ in terms of which SBUV instruments are included in particular time periods, and in the averaging and merging carried out. The SBUV Merged Cohesive VN8.6 dataset uses data from a single instrument in any time period; the individual records are then bias-corrected to produce a continuous record (Wild and Long, 2015). In contrast, the SBUVMOD VN8.6 data set is constructed by averaging all available data for a particular period (Frith et al., 2014).

**ACPD**

doi:10.5194/acp-2015-882

**Solar cycle signals in stratospheric ozone – Part 1: Satellite observations**

A. Maycock et al.

In addition to the "raw" SBUV versions described above, we also analyse the ozone dataset from McLinden et al. (2009), which uses SAGE I and SAGE II v6.2 data to correct for instrument drifts and inter-instrument biases in the SBUV VN8.0 dataset; this therefore benefits from the improved long-term stability of the SAGE II data, but retains the improved spatial sampling of SBUV. Note that the SAGE II v6.2 data employed by McLinden et al. (2009) differs from the version described in Sect. 2.1.1, which has been used in most previous solar cycle studies (e.g. Soukharev and Hood, 2006; Randel and Wu, 2007; Gray et al., 2009; Cionni et al., 2011). McLinden et al. (2009) discuss how the temperature profiles provided in the SAGE data files, which as described above came from the NMC/NCEP (re)analysis, contain apparently spurious long-term trends in the upper stratosphere, as compared to observations. McLinden et al. (2009) therefore convert the SAGE II data to mixing ratios using a temperature climatology with an estimate of the long-term stratospheric temperature trend from observations superposed. Section 3.1 addresses the importance of uncertainties in past stratospheric temperatures for the conversion of SAGE II data in more detail.

### 2.1.3  Other ozone records

We also present results for the shorter (October 1991 to November 2005) Upper Atmosphere Research Satellite (UARS) Halogen Occultation Experiment (HALOE) v19 record (Grooß and Russell, 2005; http://haloe.gats-inc.com/download/index.php). Results are also shown for the Binary DataBase of Profiles (BDBP) Tier 0 dataset (Bodeker et al., 2013) (available from http://www.bodekerscientific.com/data/the-bdbp), which covers 1979–2007 and consists of multiple satellite records, including SAGE I and II (v6.2), HALOE, Polar Ozone and Aerosol Measurement (POAM) II and III and Improved Limb Atmospheric Spectrometer (ILAS) I and II data, as well as ozone sondes and ground-based measurements.

Discussion Paper | Discussion Paper | Discussion Paper | Discussion Paper | Discussion Paper |

**[ACPD](doi:10.5194/acp-2015-882)**

doi:10.5194/acp-2015-882

**Solar cycle signals in stratospheric ozone – Part 1: Satellite observations**

A. Maycock et al.

## 2.2 The multiple linear regression model

Following numerous earlier studies (e.g. Frame and Gray, 2010; Mitchell et al., 2015a), the solar-ozone responses in observations are diagnosed using a multilinear regression (MLR) technique; this enables the signals of different forcings within a single time-series to be separated.

The ozone data are first deseasonalised by removing the long-term monthly mean at each latitude and pressure (or altitude). As in past studies, we then perform an MLR analysis on the timeseries of monthly mean anomalies at each location, $O_3'(t)$, to diagnose the 11 year solar cycle component:

$$O_3'(t) = A \times \text{F10.7}(t) + B \times \text{EESC}(t) + C \times \text{QBO}(t)$$

$$+ D \times \text{QBO}_{\text{orthog}}(t) + E \times \text{AOD}_{\text{volc}}(t) + F \times \text{Nino3.4}(t) + r(t), \tag{1}$$

where $r(t)$ is a residual. The analysis mainly focuses on the annual-mean signals, which are calculated by regressing all months as a single timeseries. The basis functions used in the MLR are: the F10.7 cm radio solar flux (http://lasp.colorado.edu/lisird/tss/noaa_radio_flux.html), equivalent effective stratospheric chlorine (EESC), two orthogonal quasi biennial oscillation (QBO) indices, defined as the first two principal components of ERA-Interim tropical (10° N–10° S) zonal monthly mean winds between 70 and 5 hPa (Dee et al., 2011), the global aerosol optical depth at 550 nm ($\text{AOD}_{\text{volc}}$) updated from Sato et al. (1993), and the Nino 3.4 index derived from the Extended Reconstructed Sea Surface Temperature (ERSST) v3b dataset (http://www.esrl.noaa.gov/psd/data/gridded/data.noaa.ersst.html). Figure 1 shows example timeseries of these indices from 1984–2004 in arbitrary units. The coefficients A–F are calculated using linear least squares regression. We use the F10.7 cm flux to represent solar activity because it is a more appropriate proxy for UV radiation, the key driver of the stratospheric ozone response, than other indices such as sunspot number or total solar irradiance (Gray et al., 2010). The results presented in Sect. 3 assume a difference of 130 solar

Discussion Paper | Discussion Paper | Discussion Paper | Discussion Paper | Discussion Paper |

**[ACPD](doi:10.5194/acp-2015-882)**

doi:10.5194/acp-2015-882

**Solar cycle signals in stratospheric ozone – Part 1: Satellite observations**

A. Maycock et al.

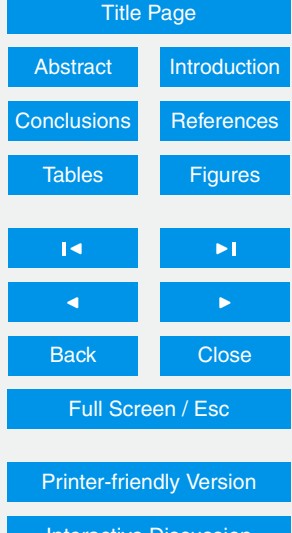

flux units ($1\,\mathrm{SFU} = 10^{-22}\,\mathrm{Wm^{-2}\,Hz^{-1}}$) to represent the difference between the 11 year solar cycle maximum and minimum.

One important issue for MLR analysis is the handling of possible autocorrelation in the regression residuals and its effects on the estimation of statistical uncertainties. A Durbin–Watson test does not reveal significant autocorrelation in the residuals at most locations; however, this is likely to be because most locations have a considerable fraction of missing data points. In the regression analysis of model data in Part 2, for which all locations have complete sampling, a Durbin–Watson test reveals significant serial correlation in the regression residuals in many locations for lags of one and two months, particularly in the lower stratosphere and mesosphere. This serial correlation can lead to spurious overestimation of the statistical significance of the regression coefficients and we therefore include an autoregressive term in the model. Given the significant serial correlations in some regions up to a lag of two months, a second order autoregressive noise process (AR2) is used, which assumes the residuals $r(t)$ have the form:

$$r(t) = ar(t-1) + br(t-2) + w(t), \tag{2}$$

where $a$ and $b$ are constants and $w(t)$ is a white noise process; this is the same approach employed in the recent SPARC SI$^2$N analysis of ozone trends (Tummon et al., 2015; Harris et al., 2015). The inclusion of this term has a very minor effect on the results for the observational datasets in Part 1, but has a greater effect for the model results in Part 2. We therefore include it here for consistency between both parts of the study.

Discussion Paper | Discussion Paper | Discussion Paper | Discussion Paper | Discussion Paper |

**ACPD**

doi:10.5194/acp-2015-882

**Solar cycle signals in stratospheric ozone – Part 1: Satellite observations**

A. Maycock et al.

## 3 Results

### 3.1 The SAGE II record

Figure 2 shows timeseries of monthly tropical (30° S–30° N) mean percent ozone anomalies from 1984 to 2004 at select pressure (or approximately equivalent altitude) levels (1, 3, 5, 10, 30 hPa) for SAGE II versions 6.2 and 7.0 in units of mixing ratio (on pressure surfaces) and number density (on altitude surfaces). The lowest panel shows the monthly mean F10.7 cm solar flux for reference.

The number density data (blue and green lines) are in close agreement for the two versions of SAGE II in the mid stratosphere (24, 31 and 36 km) both in terms of high frequency fluctuations and long-term changes. At 31 km, there are variations which are consistent with a QBO influence, but there are no clear quasi-decadal fluctuations in phase with the solar cycle. At 36 and 40 km, an apparent solar cycle signal becomes more evident, with relatively low ozone values from 1994 to 1998 during an 11 year solar cycle minimum, and higher values from 1989 to 1992 at solar maximum. However, the data in the early and later parts of the records are noisier and clear variations in phase with the solar signal are not immediately evident.

The two SAGE II ozone mixing ratio datasets (black and red lines) are also in reasonable agreement for long-term changes in the mid stratosphere, although there are differences in the interannual variations. However, in the upper stratosphere (1 and 3 hPa) there are substantial differences in both the long and short-term variations. At these levels, SAGE II v6.2 (black) shows persistent negative anomalies in the early part of the record which are not evident in v7.0 (red). These coincide with the 11 year solar cycle minimum from 1985 to 1988. Furthermore, in the latter part of the record, v6.2 shows relatively large amplitude fluctuations with a persistent positive mean anomaly from 2002 to 2004 which coincides with the peak and subsequent downward phase of solar cycle 23. Thus, there are differences between the two SAGE II vmr datasets in the quasi-decadal evolution of ozone in the tropical upper stratosphere. Overall, the

**ACPD**

doi:10.5194/acp-2015-882

**Solar cycle signals in stratospheric ozone – Part 1: Satellite observations**

A. Maycock et al.

two versions of SAGE II number density data are in much closer agreement than the two vmr datasets.

Figure 3a and b shows latitude–altitude plots of the percentage differences in ozone number density between solar maximum and minimum for SAGE II v6.2 and v7.0, respectively. The solar-ozone signals in the two datasets are highly consistent and show an increase of 2–4 % across the tropical and subtropical stratosphere, except for a small ($< 1$ %) decrease at 30 km in the tropics. There is a relative maximum of 3–4 % in the tropics at 50 km, and two off equatorial peaks of a similar magnitude at $\sim 40$ km and $\pm 35°$. These findings are consistent with Remsberg and Lingenfelser (2010) and Remsberg (2014), who found similar 11 year solar-like signals in tropical upper stratospheric ozone number density in the v6.2 and v7.0 datasets.

Figure 3c and d shows equivalent plots to Fig. 3a and b for SAGE II in units of mixing ratios on pressure surfaces. The ozone responses between $\sim 50$–$10$ hPa are very similar in the two versions, and strongly resemble Fig. 3a and b, with an increase in the tropical lower stratosphere of $\sim 1$–$2$ %. The structure of the ozone signals between 20 and 5 hPa are also similar in the two versions, with subtropical maxima of 1–2 % and a distinct equatorial minimum. However, the ozone signals in the upper stratosphere are markedly different. Polewards of $\pm 20°$ the structures are similar in both datasets, but the magnitude is $\sim 1$ % larger in v6.2 compared to 7.0. In the tropics, the v6.2 data show a large maximum in the uppermost stratosphere of up to 5 %, whereas the v7.0 data show a smaller increase of 1 % in this region. This is broadly consistent with the evolution of the ozone timeseries at 1 hPa in Fig. 2 and the presence or lack of a quasi-decadal variations that are approximately in phase with the solar cycle.

### 3.1.1 Sensitivity to temperature record

Since the ozone signals in the two versions of SAGE II are comparable in number density units, the differences between Fig. 3c and d must be related to the conversion of SAGE II to mixing ratios. As described in Sect. 2.1, v6.2 employed NMC/NCEP temperature data for this conversion, but this was changed to MERRA reanalysis data for

Discussion Paper | Discussion Paper | Discussion Paper | Discussion Paper |

**ACPD**

doi:10.5194/acp-2015-882

**Solar cycle signals in stratospheric ozone – Part 1: Satellite observations**

A. Maycock et al.

v7.0 (see Damadeo et al., 2013 for details). The differences in the solar-ozone signals in the upper stratosphere must therefore be related to the use of different temperature analyses in the conversion. It is known that the evolution of upper stratospheric temperatures in some reanalyses show unphysical trends (Mitchell et al., 2015a), and these have been corrected for in some solar–climate studies (e.g. Frame and Gray, 2010; Hood et al., 2015). Spurious variations in upper stratospheric temperatures in meteorological analyses and reanalyses, which are introduced through changes in the observing system over time, may therefore mask or enhance the signal of the 11 year solar cycle in the SAGE II record.

Figure 4 shows timeseries of annual and tropical mean temperature anomalies at select pressure levels (1, 2, 5, 10, 30 hPa) for the NMC/NCEP and MERRA datasets. At 30 hPa, the evolution of the two records is nearly identical, with a long-term cooling trend of $\sim 0.6\,\mathrm{K\,decade^{-1}}$ and a warming of $\sim 0.5\,\mathrm{K}$ around the time of the Mount Pinatubo volcanic eruption in 1991. However, at pressures less than 30 hPa there are substantial differences between the records. In the upper stratosphere at 1 hPa, the NMC/NCEP data show a long-term warming trend of $1.6\,\mathrm{K\,decade^{-1}}$, which is in contrast to the cooling trend of $1\,\mathrm{K\,decade^{-1}}$ in MERRA. The MERRA data also show an exceptional cooling of $\sim 3\,\mathrm{K}$ over a 3 year period at the end of the timeseries during the downward phase of solar cycle 23, while the NMC/NCEP data show a marked warming in this period. There are therefore substantial differences in both the long- and short-term variations of tropical temperatures throughout the middle and upper stratosphere between the two records.

The evolution of temperature will affect the altitude of a given pressure surface, as well as the conversion from number density to mixing ratio. It is well known that a long-term cooling will lower the altitude of pressure surfaces, a so-called "atmospheric shrinking" effect. Therefore the presence of cooling near the stratopause in MERRA would tend to lead to a greater atmospheric shrinking than for the NMC/NCEP temperatures. Furthermore, the conversion from number density to mixing ratio is proportional to temperature, so a positive correlation between number density and temperature over

**ACPD**

doi:10.5194/acp-2015-882

**Solar cycle signals in stratospheric ozone – Part 1: Satellite observations**

A. Maycock et al.

Discussion Paper | Discussion Paper | Discussion Paper | Discussion Paper

the solar cycle would tend to increase the mixing ratio signal on a given pressure surface. Figure 5 shows the solar cycle signals in stratospheric temperatures derived for (a) NMC/NCEP and (b) MERRA. Although the broad structure of the temperature signals are largely consistent, the maximum warming in the tropics occurs at 4 hPa in

MERRA as compared to 2 hPa in NMC/NCEP. The peak warming is also around 25 % smaller in MERRA compared to in NMC/NCEP.

A valid question is thus which representation of past stratospheric temperatures is likely to be most realistic. Mitchell et al. (2015a) compared MERRA temperatures to Stratospheric Sounding Unit (SSU) data in the upper stratosphere and found consider-

10 able differences in the long-term and decadal variations between the records. However, the NMC/NCEP data show a long-term warming in the upper stratosphere, which is in contrast to the cooling expected from increasing atmospheric $CO_2$ and declining ozone abundances over this period. Nevertheless, there remain uncertainties in the observed evolution of upper stratospheric temperatures over the reanalysis era (Thompson et al.,

2012), which makes it more challenging to evaluate which temperature dataset, if any, is likely to be realistic.

In light of these uncertainties, we conduct our own conversion of SAGE II from number densities to mixing ratios to test the impact of the NMC/NCEP and MERRA temperature fields on the solar-ozone signal. Each monthly and zonal mean ozone profile

is first converted to number density on pressure levels, using the hydrostatic relation, and then to mixing ratios using the ideal gas law. The MLR in Eq. (1) is then applied to the converted ozone mixing ratio timeseries to derive a solar-ozone signal that can be compared to the original SAGE II datasets.

As a first test, we convert the SAGE II v6.2 number density profiles using the full

timeseries of temperatures from NMC/NCEP and MERRA to test how our post-hoc converted data compares to the original records. These are shown in Fig. 6a and b for NMC/NCEP and MERRA, respectively, which can be compared to Fig. 3c and d. We stress that differences are to be expected, since in the original datasets each profile

Discussion Paper | Discussion Paper | Discussion Paper | Discussion Paper | Discussion Paper |

**ACPD**

doi:10.5194/acp-2015-882

**Solar cycle signals in stratospheric ozone – Part 1: Satellite observations**

A. Maycock et al.

is converted separately before averaging is performed, whereas here we convert the monthly, zonal and latitudinally averaged profiles.

The solar-ozone signal in the post-hoc converted data using NMC/NCEP show a qualitatively similar structure to Fig. 3c, but the magnitude of the signal is under-estimated by around 2 % in the tropics. The signal in the tropical lower stratosphere is also overestimated, which is an issue in all of the post-hoc converted fields. Although it is not clear what aspect of the conversion method leads to this overestimation (e.g. the use of monthly and zonal mean fields), we note that this corresponds to smaller absolute changes in ozone than the signals in the mid and upper stratosphere. Figure 6a indicates that our conversion method is reasonable for exploring the sensitivity of the SAGE II data to stratospheric temperatures in the mid and upper stratosphere, but is probably not suitable for comparison in the lower stratosphere. The data converted using MERRA temperatures (Fig. 6b) also agrees well with the original dataset in the mid and upper stratosphere (Fig. 3d). In particular, the reduced magnitude of the solar-ozone signal in the tropical upper stratosphere is captured, which allows us to explore which properties of the temperature records contribute to the differences in this region.

We now explore the impact of different properties of the temperature records on the solar-ozone signals. As was discussed above, the NMC/NCEP and MERRA datasets show markedly different long-term trends and some differences in their solar cycle variability in temperatures. The impact of these differences on the solar-ozone signal is tested by converting the SAGE II v6.2 number density data to mixing ratios using a monthly temperature climatology added to a linear trend and solar signal component, which are extracted from each dataset as a function of latitude and pressure. The solar-ozone signals for these trend + solar converted datasets are shown in Fig. 6c and d. The data converted using NMC/NCEP temperatures show a larger solar-ozone signal in the tropical upper stratosphere by 1–2 % compared to that using MERRA. Further tests (not shown), show that this is not affected by differences in the temperature

**[ACPD](doi:10.5194/acp-2015-882)**

doi:10.5194/acp-2015-882

**Solar cycle signals in stratospheric ozone – Part 1: Satellite observations**

A. Maycock et al.

climatologies between the datasets, but is rather associated with the time-dependent trend and solar cycle components.

The remaining panels in Fig. 6 show equivalent results to Fig. 6c and d, but for data converted using either the trend (Fig. 6e and f) or solar (Fig. 6g and h) component of the temperature datasets. In both cases, the data converted using NMC/NCEP show a larger solar-ozone signal in the tropical upper stratosphere compared to that using MERRA. This indicates that both the long-term trend and solar cycle variations in temperature contribute to the differences between the solar-ozone signals in Fig. 6a and b.

In conclusion, the solar-ozone signals in SAGE II are highly consistent across the two versions in terms of number densities on altitude surfaces. However, the upper stratospheric signals in mixing ratio units are extremely sensitive to the temperature records used for conversion. We have shown that much of this sensitivity is related to differences in the long-term trends and solar cycle variability in the temperature records. The long-term warming trend in the upper stratosphere in NMC/NCEP data is at odds with our understanding of recent changes in atmospheric composition and their impact on stratospheric temperatures; however, the peak of the solar cycle signal in stratospheric temperatures is at lower altitude in MERRA than predicted from theory and models. Recent analysis suggests that the anticipated photochemical relationship between ozone and temperature in the upper stratosphere is more realistic for the v7.0 mixing ratio data than for v6.2 (Dhomse et al., 2015). These substantial open questions around the SAGE II mixing ratio datasets raise questions around the approach adopted for representing the solar-ozone signal in the SPARC ozone dataset for CMIP5, which largely relied upon SAGE II v6.2 vmr data. This will be discussed further in Part 2 of this study (Maycock et al., 2015).

## 3.2 Recent extensions to SAGE

Figure 7 shows the solar-ozone signals for 1984–2011 in the five extended SAGE II datasets described in Sect. 2.1.1. The importance of how the separate satellite

Discussion Paper | Discussion Paper | Discussion Paper | Discussion Paper |

**ACPD**

doi:10.5194/acp-2015-882

**Solar cycle signals in stratospheric ozone – Part 1: Satellite observations**

A. Maycock et al.

records are merged for the solar-ozone signal is immediately apparent when comparing Fig. 7a and b, which show SAGE-GOMOS 1 and SAGE-GOMOS 2, respectively. SAGE-GOMOS 1 shows a generally smoother spatial structure as compared to SAGE-GOMOS 2, but the magnitudes a quite similar overall. The differences of the two
merging procedures are summarised by Tummon et al. (2015), and are described in more detail by Kyrölä et al. (2015) and Penckwitt et al. (2015). However, it is difficult to identify which factors in the merging procedure are likely to be most important for the differences in the spatial structures of the solar-ozone responses. Analysis of the datasets over the SAGE II period alone reveals similar differences (not shown), which
suggests that the use of SAGE II or GOMOS as a reference, to which the other record is adjusted, is a key factor which can alter the SAGE II signal itself.

Figure 7c shows the merged SAGE II OSIRIS dataset. These data mostly show significant increases in ozone at solar maximum in the southern subtropics and in the tropics. This is similar to the results of Bourassa et al. (2014), but they also find that
the increase in the northern midlatitudes is statistically significant for the period 1985 to near present day (see their Fig. 9). The absence of a significant change in ozone in the Northern Hemisphere in the SAGE II OSIRIS dataset is also in contrast to the two SAGE-GOMOS datasets, which both show significant increases in ozone in this region at pressures less than $\sim 10$ hPa. Hubert et al. (2015) identified a significant positive
drift of 5–8 % decade$^{-1}$ in OSIRIS data above 35 km compared to ozonesondes and lidar measurements; this may contribute to the differences in the solar-ozone signal between SAGE II v7.0 and SAGE-OSIRIS.

The SWOOSH record (Fig. 7d) shows a much smoother and continuous increase in ozone of 1–3 % across the tropics and subtropics between 2–5 hPa. Since the SAGE II
v7.0 vmr data do not show a significant increase in ozone at these levels between ±10° (see Fig. 3d), this part of the signal must arise from the other data included in SWOOSH over this period. The response in SWOOSH is most similar to that in SAGE-GOMOS 1.

Discussion Paper | Discussion Paper | Discussion Paper | Discussion Paper | Discussion Paper |

**ACPD**

doi:10.5194/acp-2015-882

**Solar cycle signals in stratospheric ozone – Part 1: Satellite observations**

A. Maycock et al.

Finally, GOZCARDS shows a fairly smooth increase in ozone across the tropics, which maximises with a magnitude of 3 % at ∼ 3 hPa. The signal at these levels extends to ±50° and appears to be consistent with the larger increase in ozone at these levels in the SAGE II v6.2 data compared to v7.0, which is used in the other four extended records. There is also a strong and statistically significant increase in ozone in the lower tropical stratosphere of up to 5 % at 50 hPa. The signal in GOZCARDS has a similar structure to that in SWOOSH, but is around 1 % larger.

There are several common features in the solar-ozone signals across the five extended datasets. These include a statistically significant increase in ozone in the mid and upper stratosphere, and an absence of ozone changes in the tropical mid stratosphere at ∼ 10 hPa. Most of the extended SAGE II datasets also show significant increases in ozone in the tropical lower stratosphere of a few percent. It has been hypothesised that positive ozone anomalies could occur in this region as a result of changes in the large-scale stratospheric circulation during the solar cycle (Kuroda and Kodera, 2002).

Although there are some similarities between the five extended datasets there are also marked differences. This is despite the fact that four of the five datasets use the same version of SAGE II as a basis, indicating that the procedures for combining records have an important role in determining the differences. In some key regions, such as the mid and upper stratosphere where ozone heating plays a major role in the radiative budget of the stratosphere, the magnitudes of the solar-ozone signals differ by up to a factor of 3–4 (see e.g. Fig. 4 showing SAGE II vn6.2 and vn7.0). These differences will have implications for the contribution of the solar-ozone response to stratospheric heating. They may therefore be important for understanding the climate response to solar forcing, including the contribution of the "top-down" pathway to the surface climate response. From the results in this section, we conclude that whilst longer ozone records can be obtained by merging multiple datasets, this does not necessarily reduce the uncertainty in the solar-ozone response owing to the dependence of the signals on data selection and merging procedures.

Discussion Paper | Discussion Paper | Discussion Paper | Discussion Paper |

**ACPD**

doi:10.5194/acp-2015-882

**Solar cycle signals in stratospheric ozone – Part 1: Satellite observations**

A. Maycock et al.

Discussion Paper | Discussion Paper | Discussion Paper | Discussion Paper | Discussion Paper |

**[ACPD](doi:10.5194/acp-2015-882)**

doi:10.5194/acp-2015-882

**Solar cycle signals in stratospheric ozone – Part 1: Satellite observations**

A. Maycock et al.

### 3.3 The SBUV record

In addition to SAGE, the other long-term satellite record for stratospheric ozone is the SBUV dataset, which extends from 1970 to near the present day. Unlike for SAGE II, which represents measurements from a single instrument in continuous orbit, the SBUV record is formed from multiple instruments which have been launched on various satellite platforms. Thus, whilst the nadir-viewing SBUV instruments provide greater spatial and temporal sampling than the limb-viewing SAGE II, there are issues around inter-instrument calibration and merging the separate records. Owing to its viewing geometry, the vertical resolution of SBUV below ~ 15 hPa is much poorer than for SAGE II making it more challenging to extract information about ozone in the middle and lower stratosphere.

Figure 8 shows timeseries of monthly percent ozone anomalies at select pressure levels (as in Fig. 2) for the SBUV VN8.0 (black), SBUVMOD VN8.6 (red), and SBUV Merged Cohesive VN8.6 (blue) datasets. The differences between the SBUV records tend to increase with increasing pressure. At 1 hPa, the three datasets are in reasonable agreement between 1984–1994, but start to diverge after this. The most notable difference is that SBUV VN8.0 shows a larger positive trend from the mid-1990s to the mid-2000s than in the two SBUV VN8.6 records; this partly coincides with the ascending phase of solar cycle 23. At 5 hPa, the SBUV Merged Cohesive VN8.6 record shows a larger negative trend from the mid-1980s to the late-1990s than in the other datasets, with the SBUVMOD VN8.6 and SBUV VN8.0 records being more consistent with one another. At 30 hPa, all of the records show similar temporal variations in the 1980s, but with different offsets which come from the more divergent long-term behaviours in the 1990s and 2000s. At this pressure level, the SBUV VN8.0 shows a larger negative trend over the entire period than in the SBUV VN8.6 records.

Figure 9 shows zonal and annual mean cross sections of the solar-ozone responses in the (a) SBUV VN8.0, (b) SBUVMOD VN8.6, and (c) SBUV Merged Cohesive VN8.6 datasets. The SBUV VN8.0 and SBUV Merged Cohesive VN8.6 records show larger

increases in ozone at solar maximum of up to 2–3 % in the upper stratosphere peaking at ∼ 3 hPa. In contrast, SBUVMOD VN8.6 only shows statistically significant increases of 1–2 % in the subtropics at around ±30°. All of the records show an equatorial minimum in the mid stratosphere, although this is at higher pressure (5–10 hPa) in SBUV VN8.0 compared to the SBUV VN8.6 records (3–5 hPa). Between 10–50 hPa, the SBUV VN8.6 based records show generally smaller positive anomalies (1–2 %) in the southern subtropics than in SBUV VN8.0 (2–3 %). There is also an increase in ozone in the northern extratropics in SBUV Merged Cohesive VN8.6, which is present to a lesser extent in the SBUVMOD VN8.6 dataset, but which is absent in SBUV VN8.0. However, we note that the poor vertical resolution (∼ 10 km) of the SBUV instruments at pressures greater than ∼ 15 hPa means that there are large uncertainties in these features.

A comparison of the previous SBUV VN8.0 and the latest SBUVMOD VN8.6 records from NASA reveals a decrease in the magnitude of the solar-ozone signal in the upper stratosphere in the newer dataset. The modifications to the data processing algorithm between the two versions are documented by Bhartia et al. (2013); these include the use of new ozone absorption cross-sections, a new a priori ozone climatology, and a new cloud-height climatology. In addition, changes were also made to the inter-instrument calibration, which is now achieved at the radiance level during periods of overlap between the SBUV instruments (DeLand et al., 2012; Bhartia et al., 2013). It seems likely that the calibration changes would have the greatest impact on the diagnosis of quasi-decadal variability in ozone, and it seems possible that the new procedures may have smoothed out the solar-ozone signal in the SBUVMOD VN8.6.

With regard to the comparison of the two SBUV VN8.6 datasets, Fig. 9 highlights that a key issue for isolating the solar-ozone response relates to how the different records are merged to create a coherent timeseries. The selection of data included in a given time window, along with treatment of inter-instrument offsets and drifts, can have a substantial impact on the apparent variability and trends in ozone. Thus despite the same input data being used by the two SBUV VN8.6 records, they show markedly

**ACPD**

doi:10.5194/acp-2015-882

**Solar cycle signals in stratospheric ozone – Part 1: Satellite observations**

A. Maycock et al.

different solar-ozone signals in the upper stratosphere. Tummon et al. (2015) found that the SBUV Merged Cohesive VN8.6 dataset shows substantially different long-term ozone trends compared to a range of other satellite records. For example, in the tropical upper stratosphere SBUV Merged Cohesive VN8.6 showed a positive ozone trend over 1984–1997, whereas almost all other datasets analysed showed a substantial decline of several percent per decade. In constrast, they found that the trends in SBUVMOD VN8.6 were more consistent with the other records. However, it is not necessarily the case that the long-term behaviour will characterise the variability on quasi-decadal timescales. Further work is therefore required to assess the impact of the data selection and merging procedures on quasi-decadal variability in the two SBUV VN8.6 datasets.

## 3.4   Other satellite-based records

We now evaluate the representation of the solar-ozone response in other datasets which either combine information from multiple satellites or are shorter records than SAGE and SBUV. Figure 10 shows the annual mean solar-ozone responses for the (a) SAGE corrected SBUV VN8.0 dataset from McLinden et al. (2009), (b) BDBP Tier 0, which is the raw measurement component of the BDBP dataset (see Bodeker et al., 2013), and (c) HALOE v19 data (Grooß and Russell, 2005). Figure 10a–b shows the period 1984–2004, while Fig. 10c shows 1991–2004. For comparison with HALOE, Fig. 10d shows the SAGE II v7.0 vmr data analysed for the same 1991–2004 period.

The McLinden et al. (2009) dataset in Fig. 10a shows broadly similar changes to the SBUV VN8.0 dataset (Fig. 9a) at pressures less than 5 hPa. However, in the Northern Hemisphere there are significant increases in ozone of 1–3 % between 5–40 hPa which are not present in the original SBUV VN8.0 data. Both the McLinden et al. (2009) and SBUV VN8.0 datasets show insignificant changes in ozone in the tropics at 10 hPa with an increase below between 20–50 hPa, although the vertical resolution of SBUV at these levels is relatively poor. Since McLinden et al. (2009) used a different temperature record to convert the SAGE II v6.2 to mixing ratio units (see discussion in Sect. 2.1.1),

**ACPD**

doi:10.5194/acp-2015-882

**Solar cycle signals in stratospheric ozone – Part 1: Satellite observations**

A. Maycock et al.

Discussion Paper | Discussion Paper | Discussion Paper | Discussion Paper |

**ACPD**

doi:10.5194/acp-2015-882

**Solar cycle signals in stratospheric ozone – Part 1: Satellite observations**

A. Maycock et al.

a direct comparison between Fig. 10a and SAGE II v6.2 vmr data in Fig. 3c is not possible.

The BDBP dataset in Fig. 10b shows significant increases in ozone across the tropical upper stratosphere. The latitudinal structure is relatively noisy compared to most of the other datasets in this study; this is likely to be because the offsets between the individual records that make up the dataset have not been accounted for (B. Hassler, personal communication, 2015). Nevertheless, there is a general resemblance to the SAGE II v6.2 vmr data, which is consistent with this data forming a large portion of the long-term tropical upper stratospheric ozone measurements in the BDBP dataset.

The solar-ozone response in HALOE (Fig. 10c) is distinct from most of the other datasets in this study. It is one of the only datasets that does not show a significant increase in ozone anywhere in the tropical upper stratosphere, the other exception being the combined SAGE II OSIRIS dataset. Instead, the main features in HALOE consist of increases in ozone in the subtropics of 3 % maximising at 5 hPa in the Southern Hemisphere and 15 hPa in the Northern Hemisphere. This structure would be difficult to rationalise in the context of a photochemical equilibrium response to an increase in UV radiation. Nevertheless, the SAGE II v7.0 vmr shows a broadly similar structure when analysed over the same time period (Fig. 10d), but with a larger magnitude by 1–2 %. The SAGE II v6.2 vmr and SBUVMOD VN8.6 datasets also show a comparable structure to HALOE over this period (not shown). Therefore the differences between HALOE and the other records may be related to differences in sampling period rather than fundamental differences in instrument sampling properties.

### 3.4.1 Sensitivity to time period

To further test the effect of sampling period on the diagnosis of the solar-ozone signal, we conduct MLR analysis on 21 year sub-periods taken from a single satellite record. The SBUVMOD VN8.6 dataset is chosen for this analysis because it extends for more than 40 years, and shows long-term ozone trends that compare more closely with other independent satellite records (Tummon et al., 2015).

**ACPD**

doi:10.5194/acp-2015-882

**Solar cycle signals in stratospheric ozone – Part 1: Satellite observations**

A. Maycock et al.

Figure 11 shows the annual mean solar-ozone signal in the SBUVMOD VN8.6 dataset for successive 21 year periods separated by 2 year intervals covering 1979–2009. Figure 11g shows the full 1979–2009 period for comparison. The signals diagnosed in the earlier part of the record show larger increases in ozone in the upper stratosphere and tropical lower stratosphere than are found in the later periods. The magnitude of the signal extracted for the full 31 year period lies in between these two representations. All of the six sub-periods shown in Fig. 11a–f include the two major tropical volcanic eruptions that have occurred in the past 35 years (El Chichón in April 1982 and Mt Pinatubo in June 1991), so it is unlikely that the differences amongst them are related to volcanic effects. The MLR in Eq. (1) does not include a linear trend term to represent $CO_2$ because for much of the period being considered EESC is also increasing. However, the decline in EESC since the mid-1990s (see Fig. 1c), which has occurred alongside a continued increase in $CO_2$, could affect the results. To test this, we add a linear trend term into the MLR; however, this does not strongly affect the results as compared to Fig. 11 (not shown). Therefore the differences between the six sub-periods must arise from other time-dependent factors, such as inter-instrument calibration and merging, or indeed time-dependence of the solar-ozone signal itself. These results highlight the challenges associated with extracting a quasi-decadal signal from a relatively short observational record.

## 3.5 Seasonality

The analysis so far has focused on the annual mean solar-ozone signals in satellite observations. However, there are likely to be seasonally-dependent variations in the signal that will be influenced by, and could feedback onto, the high latitude dynamical response to solar variability (Hood et al., 2015). Constraining the solar-ozone response on seasonal timescales requires high spatial and temporal coverage; this is to ensure that any seasonal component of the signal can be resolved, but also to increase the number of degrees of freedom (i.e. the number of datapoints) available for the regression. Such coverage is not adequately provided by limb-viewing instruments, such as

Discussion Paper | Discussion Paper | Discussion Paper | Discussion Paper |

SAGE II, which only have relatively sparse and infrequent sampling. The coverage is considerably better for nadir-viewing instruments, such as the SBUV; however, their vertical resolution is much poorer in the middle and lower stratosphere than for limb-viewers. There is therefore a trade-off between the information that can be usefully extracted from different data sources.

In light of the much improve sampling of SBUV compared to SAGE II, we focus here on the SBUVMOD VN8.6 dataset to examine the seasonality of the solar-ozone signal in observations. Figure 12 shows the monthly mean solar-ozone signals in the SBUVMOD VN8.6 dataset for the period 1984–2004. These have been calculated by regressing individual months separately, and therefore no autocorrelation term has been included in the MLR, since separate months are found to be approximately uncorrelated from year-to-year. We note that the precise magnitudes of the localised ozone changes are somewhat sensitive to the sampling period (see e.g. discussion in Sect. 3.4.1), but the broadscale structures are generally consistent. The key point to take from Fig. 12 is that there are much larger localised changes in ozone of up to $\sim 10\%$ in the upper stratosphere in individual months, which are associated with substantially enhanced meridional and vertical gradients compared to the annual mean signal for SBUVMOD VN8.6 shown in Fig. 9b. This point was also noted by Hood et al. (2015).

If the ozone response was determined by photochemical processes alone, one would expect a seasonal component associated with the annual cycle in solar zenith angle, with the largest anomalies expected in the summer hemisphere (Haigh, 1994). However, away from regions in approximate photochemical steady-state localised ozone anomalies are also intimately tied to stratospheric dynamical variability, particularly in the winter hemisphere where intraseasonal variability in the polar vortex is a key driver. Given the hypothesis that solar variability can modify the strength of the polar vortex (Kuroda and Kodera, 2002), it follows that there may be a dynamical signature in the ozone changes, particularly in the winter hemisphere. This is evident in Fig. 12 where there are particularly large gradients in ozone across the extratropics in July in the Southern Hemisphere and in March in the Northern Hemisphere.

**ACPD**

doi:10.5194/acp-2015-882

**Solar cycle signals in stratospheric ozone – Part 1: Satellite observations**

A. Maycock et al.

Discussion Paper | Discussion Paper | Discussion Paper | Discussion Paper

Although much of the localised changes in ozone are clearly driven by dynamical processes, it is also possible that they could feedback onto circulation through their impact on stratospheric heating rates and temperatures. Hood et al. (2015) concluded that the three chemistry–climate models from CMIP5 that simulated strong gradients in ozone in the winter upper stratosphere, which more closely resembled observations, tended to have high latitude dynamical responses that compared more favourably with reanalysis data. It may therefore be important for such seasonal aspects of the solar-ozone response to be included in model simulations that lack interactive chemistry. However, given the tight coupling between ozone and dynamics, attribution of the importance of such radiative feedbacks is particularly challenging. To our knowledge the importance of this two-way coupling for the solar–climate response has not been explicitly tested. This is important to clarify for modeling the impact of solar variability on climate because it is not known whether it is sufficient to simply prescribe a seasonally-varying solar-ozone signal, or whether a chemistry–climate model is required which can capture the coupling and feedbacks between composition, radiation and dynamics. The representation of the solar-ozone response in global atmospheric models is discussed in more detail in Part 2 of this study (Maycock et al., 2015).

## 4   Discussion

The representation of the annual mean solar-ozone response has been analysed in many of the available satellite ozone datasets. Despite there being considerable differences between individual instruments and the techniques adopted to merge multiple records, there are some consistent features in all of the datasets. Every dataset shows a statistically significant increase in ozone at solar maximum somewhere in the region 1–50 hPa, ±60°. The magnitude of the peak increase in ozone ranges from ∼ 1–5 %. An increase in ozone upon an increase in solar ultraviolet radiation is consistent with our understanding of photochemical processes in the stratosphere and results from chemistry–climate models (Haigh, 1994; Austin et al., 2008).

**ACPD**

doi:10.5194/acp-2015-882

**Solar cycle signals in stratospheric ozone – Part 1: Satellite observations**

A. Maycock et al.

Discussion Paper | Discussion Paper | Discussion Paper | Discussion Paper

However, despite all the datasets showing an increase in ozone there are marked differences in the vertical and horizontal structures of the signals. Some of the differences have been shown to be particularly sensitive to the post-processing of data; for example, the importance of the stratospheric temperature record for the conversion of SAGE II from number density to mixing ratio (see Sect. 3.1), or the method employed for merging SBUV records to create a consistent timeseries (see Sect. 3.3). More recent combined ozone datasets, which append other records to the SAGE II timeseries, can provide a longer record which is useful for constraining quasi-decadal signals. However, the results from these records are also sensitive to the methods for combining independent records, as demonstrated by the differences between two versions of the SAGE-GOMOS dataset.

The differences in the magnitude and structure of the solar-ozone response across datasets have important implications for understanding the climate response to solar variability. The impact of changes in irradiance and ozone on stratospheric heating rates, and therefore on stratospheric temperatures, are strongly height-dependent. Therefore the different solar-ozone signals would lead to different solar cycle signatures in stratospheric temperatures (e.g. Shibata and Kodera, 2005; Gray et al., 2009). Since one of the leading "top-down" mechanisms for solar–climate coupling is related to radiatively-driven changes in meridional temperature gradients in the upper stratosphere (e.g. Gray et al., 2010), it is important to constrain the contribution of ozone to this anomalous heating.

Soukharev and Hood (2006) concluded from their MLR analysis of 3 month mean ozone data that the solar signals in SAGE II v6.2 vmr, SBUV VN8.0 and HALOE were comparable enough to create a multi-instrument mean response; this was subsequently used to evaluate the CCMVal-1 models (Austin et al., 2008). However, the analysis presented here shows that the differences between individual records, which have recently been reprocessed in different ways, are often as large as the mean response, and we conclude that this precludes the formulation of a multi-instrument mean signal. We note that the relative differences in the solar-ozone responses between the

# ACPD

doi:10.5194/acp-2015-882

## Solar cycle signals in stratospheric ozone – Part 1: Satellite observations

A. Maycock et al.

Discussion Paper | Discussion Paper | Discussion Paper | Discussion Paper |

multiple records are of a similar magnitude to the differences between estimated linear trends in stratospheric ozone for the recent past (Harris et al., 2015). However, these are both considerably smaller than typical differences in measurements of climatological ozone, which can amount to $\sim$ 5–10 % in the regions of focus in this study
(60° N–60° S, 50–1 hPa; Tegtmeier et al., 2013).

Chemistry–climate models may be useful tools for constraining the solar-ozone response, at least in the annual mean; however, there remain considerable uncertainties in the characteristics of spectral solar irradiance (SSI) variability (Ermolli et al., 2013), which must be prescribed in models, and which will therefore strongly determine
the solar-ozone response (Haigh et al., 2010). Other studies have developed methods aimed at using ozone observations to constrain SSI variability (Ball et al., 2014), but as has been shown here, the differences between individual records are typically too large to provide a stringent constraint.

## 5  Conclusions

This two part study is in support of the WCRP/SPARC SOLARIS-HEPPA task group that has been formed to assist the Chemistry-Climate Model Initiative (CCMI) in the production of an ozone dataset for CMIP6 (see also Maycock et al., 2015; Hegglin et al., 2015). This first part has focused on evaluating the solar-ozone signal in current satellite observations. Changes in stratospheric ozone are an important component of
the atmospheric response to solar variability. It is therefore important to quantify the solar-ozone signal to improve our understanding and ability to model the influence of solar variability on climate. Many global climate models, such as those participating in CMIP exercises, do not currently represent stratospheric chemical processes. As a result, several ozone databases have been created for long-term climate model studies
(e.g. the IGAC/SPARC dataset for CMIP5 covering 1850–2100; Cionni et al., 2011). To allow for a realistic representation of the impacts of solar variability on climate, these

Discussion Paper | Discussion Paper | Discussion Paper | Discussion Paper | Discussion Paper

**ACPD**

doi:10.5194/acp-2015-882

**Solar cycle signals in stratospheric ozone – Part 1: Satellite observations**

A. Maycock et al.

datasets must include a solar-ozone signal. Such a signal could be derived from observations and/or from chemistry–climate model simulations.

The SAGE II dataset has been widely used for ozone studies because of its long-term stability. The representation of the solar-ozone signal in the SPARC CMIP5 ozone dataset was based on analysis of the SAGE II v6.2 mixing ratio data (along with some data from SAGE I). These data show an increase in stratospheric ozone at solar maximum with a peak of ∼ 5 % near the tropical stratopause. However, the more recent SAGE II v7.0 mixing ratio data show a markedly different solar-ozone signal, with smaller increases at the tropical stratopause of ∼ 1 %. The two versions of SAGE II show much greater consistency in their native number density on altitude coordinates; this demonstrates that the differences must arise from the change in the temperature fields used to convert number densities to mixing ratios from NMC/NCEP at v6.2 to MERRA at v7.0. A post-hoc conversion of monthly and zonal mean SAGE II data from number density to mixing ratios revealed that differences in both the long-term trends and solar cycle variations in temperature between NMC/NCEP and MERRA likely contribute to the differences in solar-ozone signals. It is not currently known which temperature dataset is likely to be most realistic, however, other recent research suggests that the expected relationship between ozone and temperature in the tropical upper stratosphere is more representative in SAGE II v7.0 (Dhomse et al., 2015).

The solar-ozone responses were also analysed for the period 1984–2009 in five combined ozone datasets that have extended SAGE II using more recent satellite measurements from GOMOS, OSIRIS, MLS and ACE-FTS. These datasets all show increases in ozone in the stratosphere at solar maximum. However, the peak magnitudes vary by several percent amongst them. The importance of data selection and merging processes for the solar-ozone response is exemplified by the differences between two SAGE-GOMOS datasets, which have the same underlying data records as their basis, but which have magnitudes that differ by around a factor of two.

Analysis of the recent SBUVMOD VN8.6 data produced by NASA show a smaller increase in upper stratospheric ozone at solar maximum by ∼ 1 % compared to the

Discussion Paper | Discussion Paper | Discussion Paper | Discussion Paper | Discussion Paper |

**[ACPD](doi:10.5194/acp-2015-882)**

doi:10.5194/acp-2015-882

**Solar cycle signals in stratospheric ozone – Part 1: Satellite observations**

A. Maycock et al.



previous VN8.0 data (Soukharev and Hood, 2006). However, the SBUV Merged Co-hesive VN8.6 dataset from NOAA, which takes a different approach for combining in-dividual SBUV records, shows a signal which more closely matches the SBUV VN8.0 data. There is therefore an outstanding question as to which of the available versions of SBUV is most reliable for diagnosing the solar-ozone response. HALOE data show a markedly different structure to most of the records presented here, but this has been shown to be at least partly related to its shorter sampling period.

To better constrain the observed solar-ozone response it is clearly desirable to have as long a timeseries as possible. However, this will almost certainly require combining multiple records, which as shown here can considerably increase uncertainties. We therefore encourage instrument teams to undertake a detailed comparison of instru-ment offsets and drifts on decadal timescales and their importance for diagnosing the solar-ozone response in combined satellite datasets.

The results raise issues for how to best include the effects of solar variability on stratospheric ozone in climate models, since current approaches range from implictly including them as part of chemistry–climate models (Hood et al., 2015), prescribing them as part of an imposed ozone field (Cionni et al., 2011), to excluding them alto-gether (Ineson et al., 2015). It is therefore extremely likely that differences in the imple-mentation of the solar-ozone signal contributed to the spread in stratospheric temper-ature responses across CMIP5 models (Mitchell et al., 2015a). This should therefore be improved in CMIP6. It is also desirable for seasonal effects, which were excluded in models without chemistry in CMIP5 (see Maycock et al., 2015; Hood et al., 2015), and which are evident in the available observational records (Fig. 12), to be incorporated, although the importance of having full coupling between chemistry and dynamics re-mains unclear. We conclude that if a more consistent representation of the solar-ozone response can be achieved in CMIP6 it will aid in understanding the response to solar variability in models.

*Acknowledgements.* A. Maycock acknowledges funding from an AXA Postdoctoral Fellowship and the ERC ACCI grant. A. Maycock also acknowledges funding from the COST action

Discussion Paper | Discussion Paper | Discussion Paper | Discussion Paper |

**[ACPD](doi:10.5194/acp-2015-882)**

doi:10.5194/acp-2015-882

**Solar cycle signals in stratospheric ozone – Part 1: Satellite observations**

A. Maycock et al.

ES1005 Towards a more complete assessment of the impact of solar variability on the Earth's climate (TOSCA) for a Short-term Scientific Mission to GEOMAR in September 2014 which initiated this work. Parts of the work at GEOMAR Helmholtz Centre for Ocean Research Kiel was performed within the Helmholtz-University Young Investigators Group NATHAN, funded by the Helmholtz-Association and GEOMAR. We thank Ray Wang for providing useful information about the SAGE II record and Stacey Frith for providing useful information about the SBUV record. We also thank the many instrument scientists and groups who have contributed to the development of the merged SAGE-GOMOS 1, SAGE-GOMOS 2, SAGE II OSIRIS, SWOOSH and GOZCARDS datasets, and for having made their data available for this study.

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

**Table 1.** Details of the ozone datasets used in this study.

| Dataset | Type | Time period considered | Reference |
|---|---|---|---|
| SAGE II v6.2 | Raw satellite product: solar occultation instrument | 1984–2004 | Wang et al. (2002) |
| GOZCARDS | Combined satellite product, including SAGE II v6.2 | 1984–2011 | Froidevaux et al. (2015) |
| SAGE II v7.0 | Raw satellite product: solar occultation instrument | 1984–2004 | Damadeo et al. (2013) |
| SAGE-GOMOS 1 | Combined satellite product, including SAGE II v7.0 | 1984–2011 | Penckwitt et al. (2015) |
| SAGE-GOMOS 2 | Combined satellite product, including SAGE II v7.0 | 1984–2011 | Kyrölä et al. (2015) |
| SAGE-OSIRIS | Combined satellite product, including SAGE II v7.0 | 1984–2011 | Bourassa et al. (2014) |
| SWOOSH | Combined satellite product, including SAGE II v7.0 | 1984–2011 | Davis et al. (2015) |
| SBUV VN8.0 | Raw satellite product: nadir-viewing instrument | 1984–2004 | |
| SAGE II v6.2 corrected SBUV VN8.0 | Combined raw satellite product | 1984–2004 | McLinden et al. (2009) |
| SBUVMOD VN8.6 | Raw satellite product: nadir-viewing instrument | 1984–2004 | McPeters et al. (2013), Frith et al. (2014) |
| SBUV Merged Cohesive VN8.6 | Raw satellite product: nadir-viewing instrument | 1984–2004 | Wild and Long (2015) |
| HALOE v19 | Raw satellite product: solar occultation instrument | 1991–2004 | Grooß and Russell (2005) |
| BDBP Tier 0 | Combined raw satellite, ozone sonde and ground based observation product | 1984–2004 | Bodeker et al. (2013) |

Discussion Paper | Discussion Paper | Discussion Paper | Discussion Paper

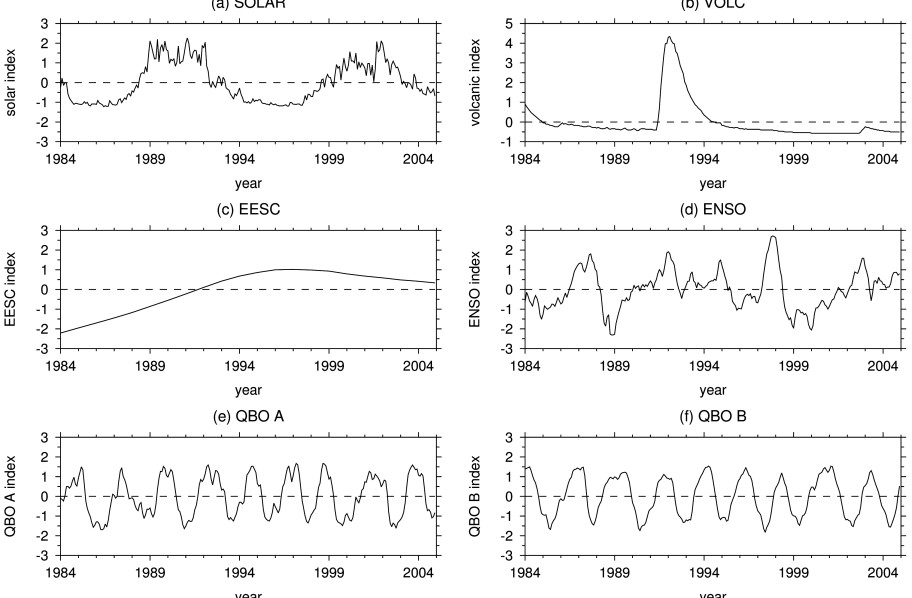

**Figure 1.** Timeseries of the six basis functions used in most of the MLR analysis shown here. **(a)** Solar forcing based on F10.7 cm flux; **(b)** volcanic forcing based on the Sato AOD index; **(c)** equivalent effective stratospheric chlorine; **(d)** ENSO based on ERSST dataset; **(e, f)** two orthogonal QBO indices defined as the first two principal component timeseries of ERA-Interim tropical winds. The timeseries are in units of standard deviation.

## ACPD

doi:10.5194/acp-2015-882

### Solar cycle signals in stratospheric ozone – Part 1: Satellite observations

A. Maycock et al.

Discussion Paper | Discussion Paper | Discussion Paper | Discussion Paper

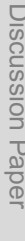

**ACPD**

doi:10.5194/acp-2015-882

**Solar cycle signals in stratospheric ozone – Part 1: Satellite observations**

A. Maycock et al.

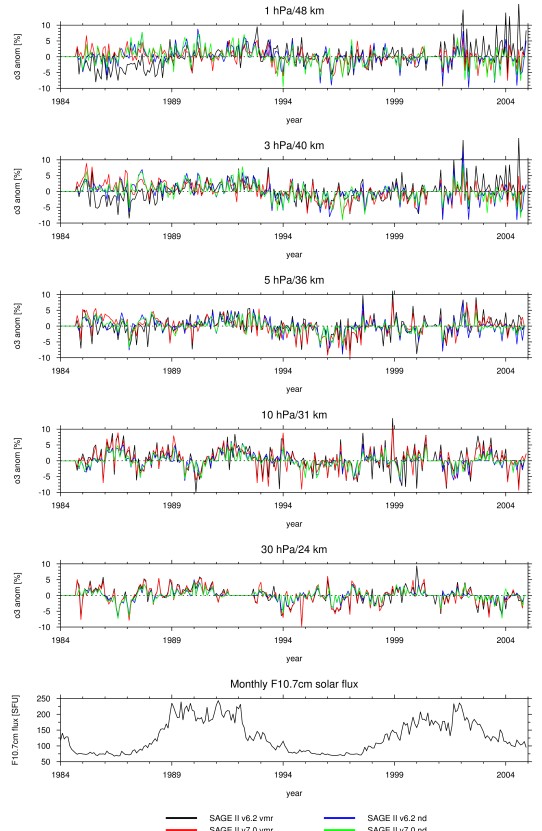

**Figure 2.** Timeseries of percent tropical (30° S–30° N) ozone anomalies for 1984–2004 at **(a)** 1 hPa (48 km), **(b)** 3 hPa (40 km), **(c)** 5 hPa (36 km), **(d)** 10 hPa (31 km), and **(e)** 30 hPa (24 km). Data are shown for SAGE II v6.2 volume mixing ratio (vmr) (black), SAGE II v7.0 vmr (red), SAGE II v6.2 number density (nd) (blue), and SAGE II v7.0 nd (green).

## ACPD

doi:10.5194/acp-2015-882

**Solar cycle signals in stratospheric ozone – Part 1: Satellite observations**

A. Maycock et al.

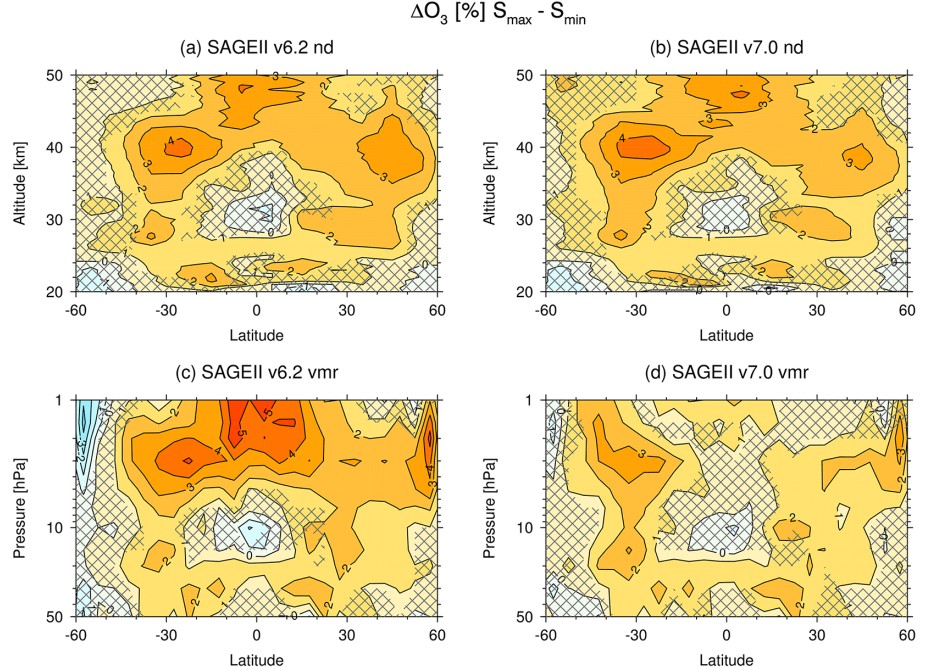

$\Delta O_3$ [%] $S_{max}$ - $S_{min}$

(a) SAGEII v6.2 nd   (b) SAGEII v7.0 nd

(c) SAGEII v6.2 vmr   (d) SAGEII v7.0 vmr

**Figure 3.** The percent (%) differences in ozone per 130 SFU for the **(a, c)** SAGE II v6.2 data and **(b, d)** SAGE II v7.0 data in terms of **(a, b)** number density–altitude units and **(c, d)** volume mixing ratio-pressure units. The contour interval is 1 %. The hatching denotes regions that are not statistically significant at the 95 % confidence level.

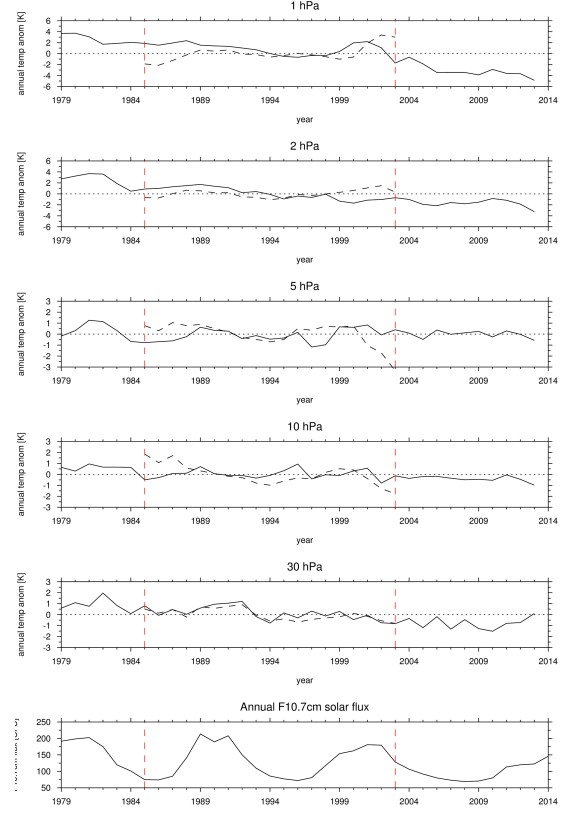

**Figure 4.** Timeseries of tropical mean temperature anomalies from the NMC/NCEP (dashed) and MERRA (solid) datasets for (top-to-bottom) 1, 2, 5, 10, 30 hPa, respectively. The time period is 1979–2014. The dashed red lines denote the period for which the post-hoc conversion of SAGE II from number density to mixing ratio is performed for the results shown in Fig. 6.

Discussion Paper | Discussion Paper | Discussion Paper | Discussion Paper | Discussion Paper |

**ACPD**

doi:10.5194/acp-2015-882

**Solar cycle signals in stratospheric ozone – Part 1: Satellite observations**

A. Maycock et al.

Discussion Paper | Discussion Paper | Discussion Paper | Discussion Paper | Discussion Paper |

**ACPD**

doi:10.5194/acp-2015-882

**Solar cycle signals in stratospheric ozone – Part 1: Satellite observations**

A. Maycock et al.

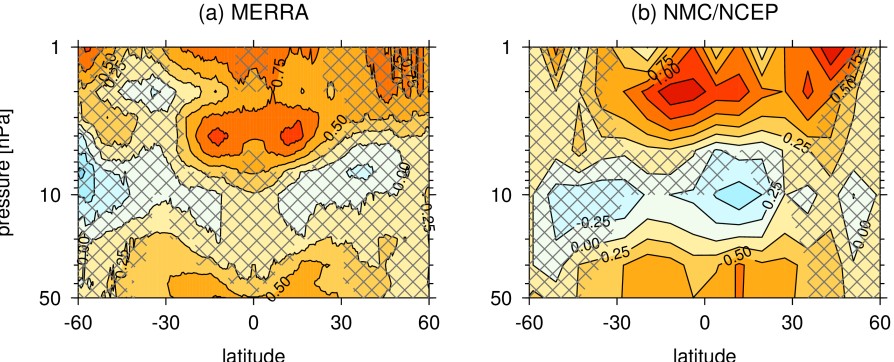

**Figure 5.** The annual mean 11 year solar cycle signals in temperature [K] from the **(a)** MERRA and **(b)** NMC/NCEP datasets. Shading denotes regions where the coefficients are not statistically significant at the 95 % confidence level. The contour interval is 0.25 K. These fields are used in the independent conversion of SAGE II v6.2 from number density to mixing ratio (see Sect. 3.1.1 for details).

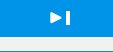
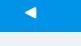
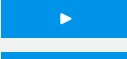


**ACPD**

doi:10.5194/acp-2015-882

**Solar cycle signals in stratospheric ozone – Part 1: Satellite observations**

A. Maycock et al.

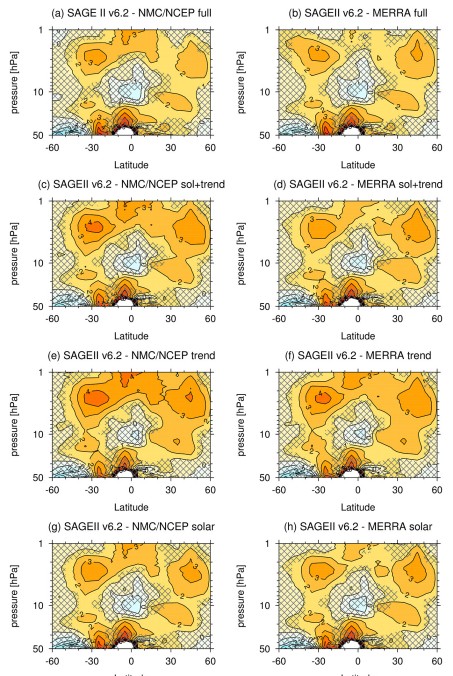

**Figure 6.** The annual percent (%) differences in ozone per 130 SFU in SAGE II v6.2 data using the post-hoc conversion from number density to mixing ratio for the period 1985–2003. The conversions are conducted using full time-dependent monthly **(a)** NMC/NCEP and **(b)** MERRA temperatures. A comparison of these with Fig. 3a and b gives an indication of how the post-hoc conversion method described in Sect. 3.1 performs. Panels **(c, d)** show the same data converted using a monthly temperature climatology from MERRA added to a linear trend and solar signal in stratospheric temperatures extracted from **(c)** NMC/NCEP and **(d)** MERRA. The remaining pairs of panels show the same as **(c, d)** but for the SAGE II conversion using the **(e, f)** linear trend or **(g, h)** solar cycle components of the temperature datasets alone. The shading is as is Fig. 3.

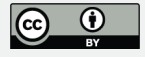

## ACPD

doi:10.5194/acp-2015-882

**Solar cycle signals in stratospheric ozone – Part 1: Satellite observations**

A. Maycock et al.

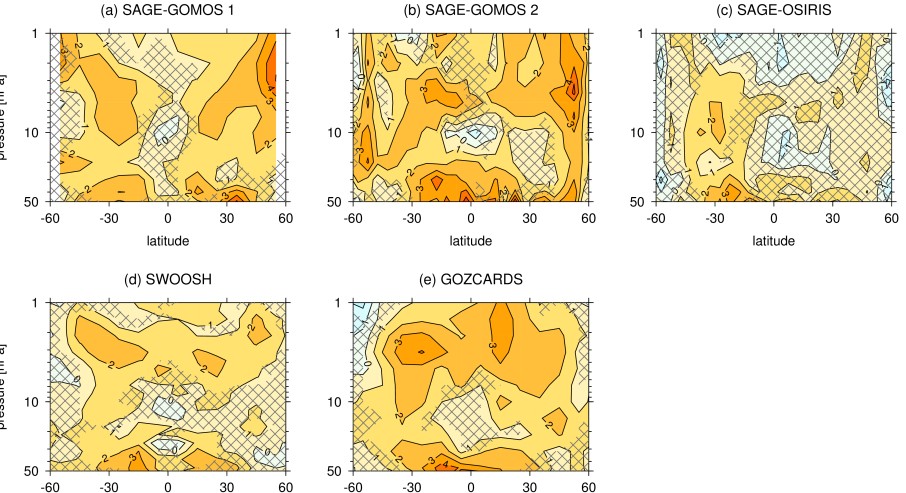

**Figure 7.** The annual percent (%) differences in ozone per 130 SFU for the **(a)** SAGE-GOMOS 1, **(b)** SAGE-GOMOS 2, **(c)** SAGE OSIRIS, **(d)** SWOOSH, and **(e)** GOZCARDS datasets. Signals are derived from a multiple linear regression analysis for the period 1984–2011 inclusive. The contour interval is 1 %. The hatching denotes regions that are not statistically significant at the 95 % confidence level.

Discussion Paper | Discussion Paper | Discussion Paper | Discussion Paper | Discussion Paper |

**ACPD**

doi:10.5194/acp-2015-882

**Solar cycle signals in stratospheric ozone – Part 1: Satellite observations**

A. Maycock et al.

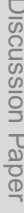
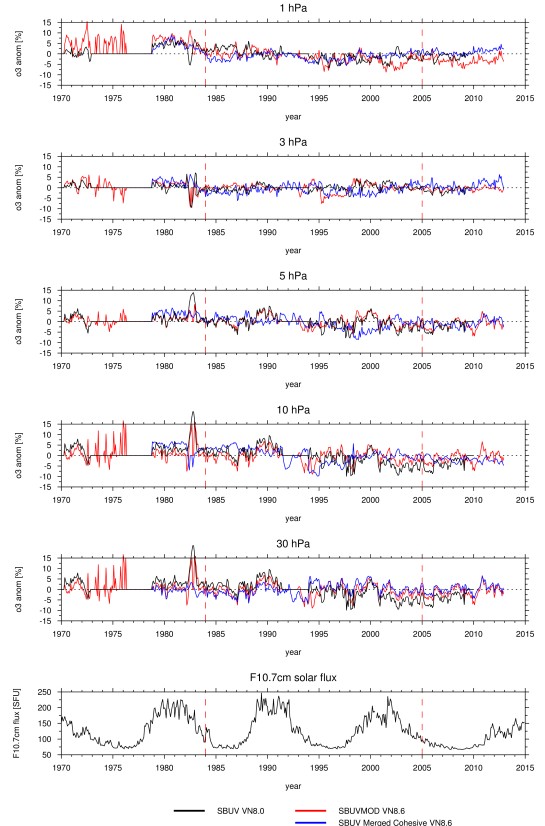
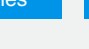
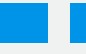
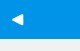
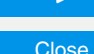

**Figure 8.** As in Fig. 2, but for the SBUV VN8.0 (black), SBUVMOD VN8.6 (red), and SBUV Merged Cohesive VN8.6 (blue) datasets. Note the time period is 1970–2015. The dashed red lines denote the period for which the MLR analysis is performed to obtain the results shown in Fig. 9.

**ACPD**

doi:10.5194/acp-2015-882

**Solar cycle signals in stratospheric ozone – Part 1: Satellite observations**

A. Maycock et al.

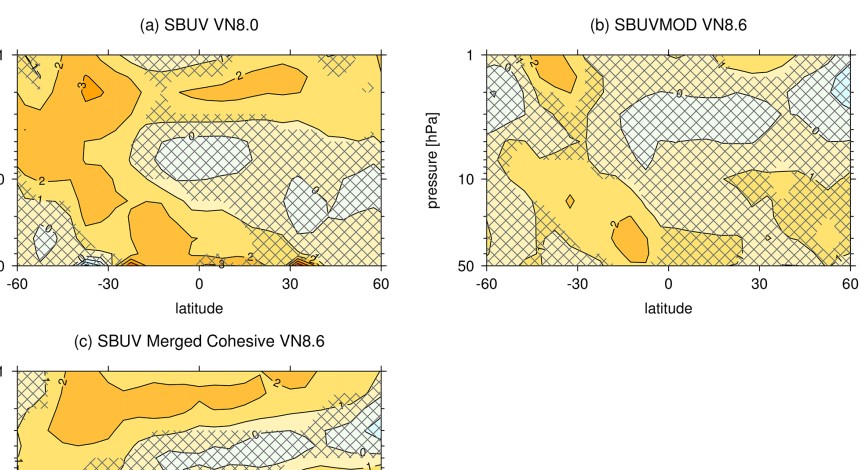

**Figure 9.** The annual percent (%) differences in ozone per 130 SFU for the **(a)** SBUV VN8.0 (McPeters et al., 1994), **(b)** SBUVMOD VN8.6 dataset (McPeters et al., 2013; Frith et al., 2014), and **(c)** SBUV Merged Cohesive VN8.6 datasets (Wild and Long, 2015). Signals are derived for the period 1984–2004 inclusive. The contour interval is 1 %. The hatching denotes regions that are not statistically significant at the 95 % confidence level.

# ACPD

doi:10.5194/acp-2015-882

**Solar cycle signals in stratospheric ozone – Part 1: Satellite observations**

A. Maycock et al.

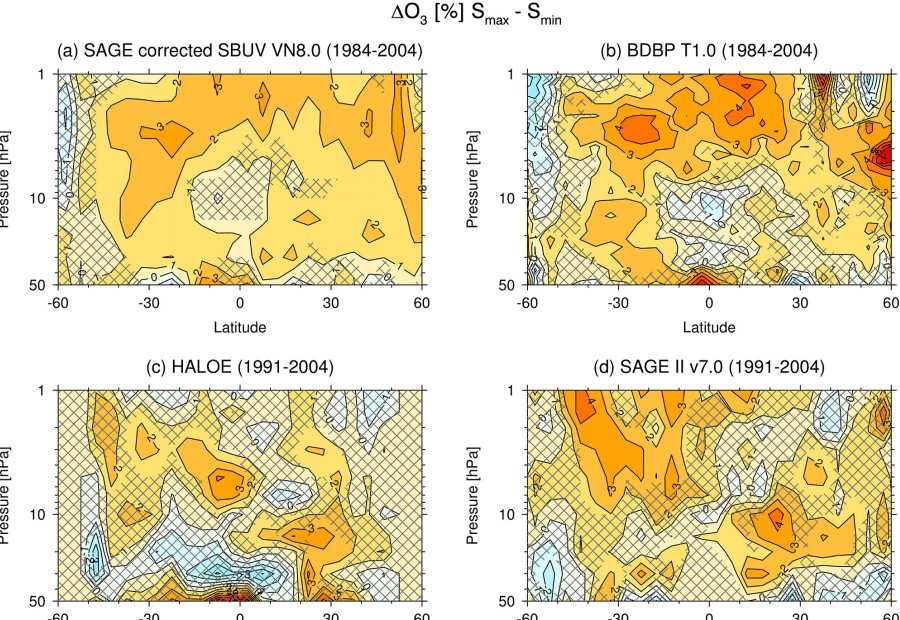

**Figure 10.** The percent (%) differences in ozone per 130 SFU for the **(a)** SAGE II corrected SBUV(/2) data (McLinden et al., 2009), **(b)** Binary Data Base of Profiles (BDBP) Tier 1.0 (Bodeker et al., 2013), **(c)** Halogen Occultation Experiment (HALOE) v19 (Grooß and Russell, 2005), and **(d)** SAGE II v7.0 vmr dataset. Signals are derived for the period 1984–2004 inclusive, except in **(c, d)** which are for 1991–2004. The contour interval is 1 %. The hatching denotes regions that are not statistically significant at the 95 % confidence level.



# ACPD

doi:10.5194/acp-2015-882

**Solar cycle signals in stratospheric ozone – Part 1: Satellite observations**

A. Maycock et al.

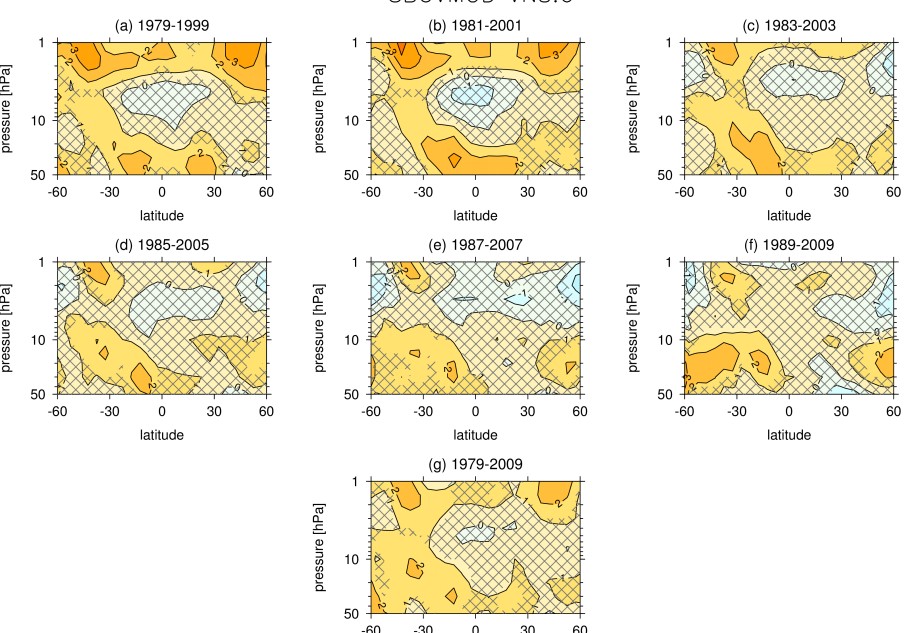

**Figure 11. (a–f)** The percent (%) differences in ozone per 130 SFU in the SBUVMOD VN8.6 dataset for 21 year periods separated by 2 year intervals covering 1979–2009. Panel **(g)** shows the result for the full 1979–2009 period. The contour interval is 1 %. The hatching denotes regions that are not statistically significant at the 95 % confidence level.

# ACPD

doi:10.5194/acp-2015-882

**Solar cycle signals in stratospheric ozone – Part 1: Satellite observations**

A. Maycock et al.

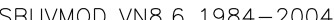

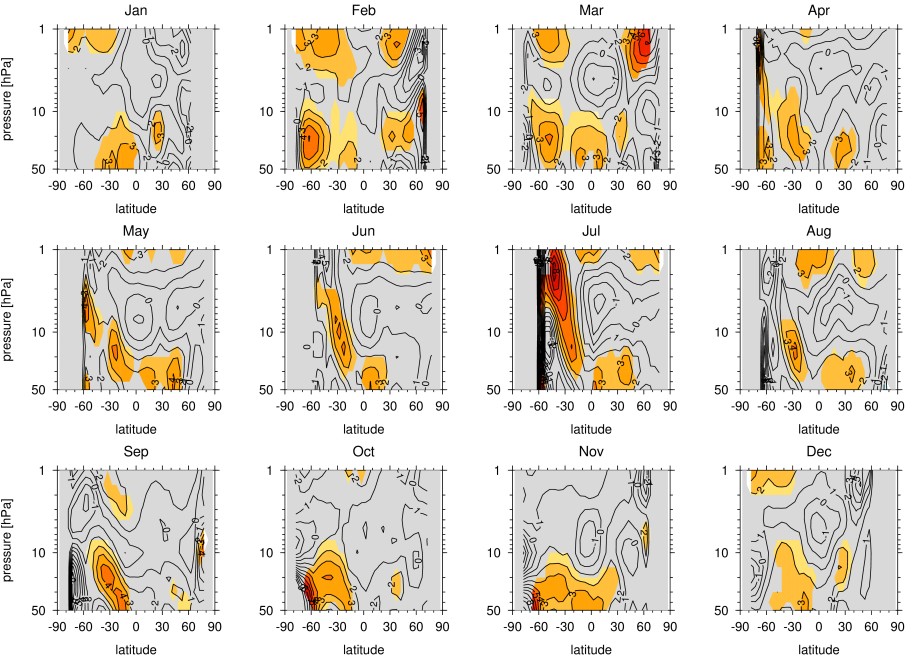

**Figure 12.** The monthly percent (%) differences in ozone per 130 SFU in the SBUVMOD VN8.6 dataset for the period 1984–2004. The contour interval is 1 %. The grey shading denotes regions that are not statistically significant at the 95 % confidence level.