# Peer review of "The representation of solar cycle signals in stratospheric ozone – Part 1: A comparison of satellite observations"

_Atmospheric Chemistry and Physics, 2015_

## Referee Comment (RC1) · Anonymous Referee #3 · 10 Feb 2016

General Comments:

The paper seeks to explore the response of stratospheric ozone to solar cycle changes as observed in several global ozone data sets through a regression analysis. The observed response is presented, discussed, and contrasted between the various data sets. There appears to be as many differences as there are similarities in the observed responses. This paper is part one of a two-part series, with the second part said to focus more on atmospheric modeling.

While there are major, but correctable, flaws in the analysis of the data sets, the general qualitative conclusions of the paper are likely robust. The quantitative results, however, will need correction for issues listed below in the Specific Comments.
[Figure]

Overall this is an important and timely study that will demonstrate the limitations of the information content in the existing historical stratospheric ozone record, as well as, the dependence of these records on the quality of ancillary data.

The title of the paper is appropriate and the abstract is a complete summary of the paper's current content. References included in the paper strike a nice balance in both quality and quantity.

Specific Comments:

The primary concern with the analysis methodology used in this paper has to do with the regression model - specifically, the lack of a diurnal term. The sparse spatial and temporal sampling provided by an occultation measurement system presents unique challenges. As has been done for decades now, the data are often reduced into monthly zonal mean time series with each mean treated as though it is representative of both the latitude and month of the center of the monthly zonal bin. While this is not usually too problematic, it is also common practice to assume that both the local sunrise and sunset sampling is unbiased in each mean so that the diurnal variability can be ignored.

While many published papers have already ignored the diurnal sampling issue, and undoubtedly many more will be submitted, it is now time to address this problem and develop a suitable approach for mitigating its impact. In order to demonstrate the presence of the issue in this work, look at figure 2. After the interruption of the SAGE II data record in late 2000 due to an instrument problem, the measurements resumed at a 50% duty cycle - alternating between sunset only and sunrise only periods of approximately 1 month duration each. This is seen in the deseasonalized monthly anomalies plotted in figure 2 as an abrupt increase in variance after 2000. The authors simply call this increased noise. Closer inspection of the figure shows that this "noise" in the equatorial-zone anomaly increases markedly with altitude. This variance is not simply noise, but rather the direct effect of a biased sampling of the known diurnal variability
in stratospheric ozone. More subtle variances in diurnal sampling occur throughout the record and other latitudes all of which contributes to the apparent "noise".

It is not immediately obvious how this additional unmodeled variance effects the regression results, but it is likely that it will correlate with some of the regression terms and produce biased results. Many, but not all (SAGE-GOMOS being an exception), of the extended time series that add other data sets on to the SAGE II record have ignored the diurnal issue in the normalization process and are therefore highly suspect. More on these extended data sets momentarily.

Computing monthly zonal means is not difficult and it should be possible to create time series where the local sunrise and local sunset measurements are kept separate. The regression would have to include a new "diurnal comb" term, but the number of data points in the regression would nearly double. The AR analysis would have to make sensible assumptions before application. Alternatively, it may be possible to keep the existing time series and add a new term that accounts for the relative sunrise to sunset proportions of events contributing to the mean. While also adding a term to the regression, this would reduce the degrees of freedom since no new data points are added to the regression. In this case, the AR analysis would probably apply as it is currently done. Either approach may require additional terms to concurrently deseasonalize the time series, since the current approach of subtracting the mean monthly mean would no longer work.

Exactly which results from this Part I are being carried forward into the Part II paper or will emerge as relevant in the combined whole is not readily apparent. If, however, the quantitative results presented here are important, this regression analysis will need to be completely redone.

Another concern, which is already discussed to some extent in the paper, has to do with the impact of the relative drift between data sets in time series comprised of data from more than one measurement system. It is reasonably clear that, given the timing

of the end of the SAGE II data set, any such relative drift will bias primarily into the EESC term. The solar cycle term will also be biased, however, as there is some correlation between the EESC and F10.7 terms. What is not discussed is that the amplitude assigned to the volcanic term will change in response to varying degrees of drift between measurement systems via its correlation with the F10.7 term. This would seem to be an important diagnostic (especially in the upper stratosphere where the expected volcanic influence may, arguably, be small), but in reality it only serves to reinforce the conclusion that the currently available time series are too short to provide sufficient orthogonality between terms with predominantly low frequency content. It would seem prudent to add a figure showing the lat-alt distribution of the amplitude of the volcanic term. Attribution of the actual response of ozone to solar cycle variations solely to the amplitude of the F10.7 term is highly problematic with these extended data sets. Drift corrected composite time series, to the extent that they can be created, would seem to be required. The associated correlations and resultant coupled uncertainties should be more thoroughly discussed.

On a positive note, the analysis done to attribute the SAGE II v6.2-v7 differences to either algorithm changes or Met data source selection are enlightening.

A few less critical Specific Comments:

Page 7 lines 224-225: Would it be possible to illustrate the effect of the AR2 vs AR1?

In several places, the authors "blame" NMC/NCEP for the poor quality of the SAGE II mixing ratio conversion when, in reality, the method by which the SAGE II team extends the NMC/NCEP profile to high altitude may be the culprit. The details of this process are discussed in a paper this work already references.

Lines 526-527: Is the amplitude of the volcanic response invariant as the time subperiod is changed?

Minor Technical Comments:

[Figure]

Overall the paper is well written and the figures are both appropriate and quite clear.

The URL link on line 111 does not appear to work as expected

The time series show in figure 1 should be extended to match the longest time period to which they are applied.

Why does panel (b) in figure 5 appear to contain much coarser latitudinal structure than seen in panel (a)?

Line 500 and elsewhere, this reviewer had some difficulty determining whether the term "increase" referred to trends or the solar cycle response. The authors may wish to introduce and use an acronym (e.g., SCR - Solar Cycle Response) to help clarify the topic under discussion rather than use a generic term such as "increase".

Line 548: "improve" should be "improved".

———————————————————

---

## Referee Comment (RC2) · Anonymous Referee #4 · 11 Feb 2016

Summary

To correctly investigate the impacts of solar variability on the climate using models, it is imperative that models correctly simulate the stratospheric ozone response to the solar cycle that may lead to an impact upon surface climate. The ozone response and feedback may be as important as the direct heating effect from solar variability. Thus, for models that do not calculate ozone variability online, a realistic representation is vital to correctly simulate this solar pathway to impact the climate. This requires using either modelled ozone responses, or those taken from multiple satellite sources. The difficulty is that different satellite data can tell different stories of how the stratospheric ozone has varied over time. Thus quantifying the true behavior, and extracting the solar

component as an input into models is difficult.

This paper, as the first of two parts, aims to investigate the behavior of several (SI2N) merged datasets that are relatively new and many of which are either based upon, or have a large component from, the long SAGE-II record. The authors present a comprehensive comparison of the behavior of the extracted solar signal in all seven SI2N ozone datasets. They also seek to quantify and understand why two versions of the same data in units of volume mixing ratio, v6.2 and v7.0, differ so much. Identifying the source of the difference due to the temperature data used to convert from number density, the authors apply their own conversion to investigate the differences in the two versions of SAGE and then expand their investigation and discussion to the other datasets based on SAGE.

While this paper does not lead to a better understanding of why the solar signal extracted from the SI2N data differ so much with each other (except versions of SAGE II), or hint which one is likely the best to use in future studies, this is an important contribution to the field. The knowledge of how and where datasets differ will provide a step towards, not only, understanding the datasets, but potentially improving them in future work. The work done to understand why SAGE II v6.2 and 7.0 differ was an interesting, revealing and useful analysis. The results are generally clear and well communicated. In context of the two part study this analysis aims to, and presumably will, inform in the production of an input ozone data set for the CMIP6 modelling runs. From the view of this reviewer, following a point that needs addressing, and some clarifications, the paper fulfils its aims and will be ready for publication.

Specific comment:

Page 25, lines 7-10: The point is made that for each of the sub-periods considered in the MLR in Figures 11a-f, both El Chichon and Mt Pinatubo are included. With both eruptions included, and yet the upper stratosphere spatial pattern changing with each sub-period, the authors suggest that this implies differences are unlikely to be volcanic.

[Figure]

As stated by the authors, the eruption of El Chichon occurred in April 1982, so the effects of the eruption on the stratosphere would be expected to have gone by early 1985, and be less pronounced in 1984 than 1983 and 1982. However, while the authors are correct that both eruptions are included in 11a-b, and any effects likely present in 11c, El Chichon is not included in panels 11d-f, which begin in 1985, almost three years after the eruption. While temperature responses to volcanic eruptions are stronger in the lower stratosphere, and less, if any in the mid-stratosphere, there are hints that the mesosphere sees a response to volcanic eruptions (e.g. Beig et al., 2003), so there may indeed be an aliasing with volcanoes that decreases (as seen in 11c-f relative to 11a-b) when El Chichon's effect is removed by the sub-period chosen. Further to this, there is a very large anomaly present in both SBUVMOD and SBUVN8.0 around the time of the 1982 eruption (Fig 8) that lasts for 1-2 years; a similar event is not present following the Pinatubo eruption, and so is likely not of volcanic origin (unless it is related to a change in the atmospheric viewing of the instrument for a reason unique to El Chichon). Such a large anomaly may have an influence on the MLR leading to the change in the spatial patterns plotted in Figure 11, and then the authors may indeed be correct that it is not volcanic in origin. Perhaps it would be worth applying SBUV-Merged Cohesive to test this, as the anomaly is not visible in that time series in Fig. 8.

The following are suggestions for the authors to clarify or reword:

Page 13; lines 20-26: Indeed, there also appears to be a larger positive anomaly in 1992-1994, a period of maximum and high activity, so it is possible this may also contribute to the enhanced signal seen in Fig. 3c.

Page 15; line 13: While records indeed show there was a warming of $\sim$0.5-1.0 K following the Pinatubo eruption, and perhaps there is a small increase in 1991/1992 in Fig. 4, it appears that at 30 hPa a warming began in 1989, followed by the $\sim$2 K decrease after 1992. The eruption does not appear to stand out in this time series, so perhaps the authors may wish to revise the focus of their comment here.
Page 15, lines 18-19: Two points of clarity here. It would better to reformulate to discuss NCEP first, as the MERRA data do not show the decline in the last three years, but in the last three years of NCEP. Note also that the solar cycle decline began in mid to late 2002, so this three year period is mainly during the maximum period. This of course does not change the point being made, and it is well worth highlighting also that this odd behavior in NCEP also occurs (though inversely) in this same, three-year period, at 5 and 10 hPa, hinting at an issue with NCEP.

Page 20, lines 19-22: The authors state that many of the SAGE-II based datasets have differences that are likely the result of merging procedures. Do the authors include in this comment also that SAGE-GOMOS 1 &2 and SAGE-OSIRIS have less data (or more data gaps) in the equatorial region than SWOOSH and GOZCARDS? Or if not, might this additionally decrease the significance of the signal in the tropics and lead to the less 'smooth' appearance of the spatial patterns? The reviewer is also aware that Aura MLS v2.2 used in GOZCARDS, and v3.3 in SWOOSH have different short term variability (larger in GOZCARDS). This might be worth checking/considering.

Page 22, line 29: The two SBUV records have almost the same datasets used, except the Merged Cohesive uses a little over a year from NOAA9 (see Tummon et al., 2015, Fig 1.).

Figure 6 might benefit with a third column of difference plots, since the differences are well discussed, though specific altitudes and latitudes are usually not mentioned. Thus the difference plots might make it easier for the reader to locate what the authors are referring to. This is at the authors discretion. However, for the point made about the solar signal in Figs 6g and 6h, relative to 6a and 6b, that the signals are larger in NCEP than MERRA, while in absolute values this is the case, I wonder how significantly different, statistically these signals are? Figures 6g and h seem more similar than 6a and b do in the upper stratosphere; this may be helped with a difference plot with significance, as mentioned.
[Figure]

Technical corrections:

Page 19, line 4: "but the magnitudes [are] quite similar"

---

## Referee Comment (RC3) · J. Lean (Referee) · 12 Feb 2016

*Review of* **"The representation of solar cycle signals in stratospheric ozone. Part 1: A comparison of satellite observations"** by A. Maycock et al

**General Comments**

The task of this paper is, according to the title "The representation of solar cycle signals in ozone". The present paper is Part I of two papers, this first focusing on using observations to determine the solar cycle representation, the second using models. The overall goal of the work is to prescribe the ozone changes during the solar cycle for use as input to climate models that do not themselves calculate the ozone responses directly, thereby allowing these models to include indirect effects of solar forcing.

The manuscript states that the "goal is to synthesize current knowledge to inform a recommendation for including the solar-ozone signal in the prescribed ozone dataset being created for CMIP6". It does not accomplish this task. A representation of the solar cycle signals based on a synthesis of observations of stratospheric ozone is not recommended. Instead, the paper concludes that it is "unlikely that satellite ozone measurements alone can be applied to estimate the necessary solar cycle ozone component of the prescribed ozone database for future coupled model intercomparisons".

These aspects of the paper preclude publication in its present form. Unless, or until the authors can/do produce the product that they set out to produce then it would seem that a paper about not achieving their goal is unwarranted. Their conclusion that satellite data are not able to inform such a product is incorrect. Ozone has been measured in one form or another by multiple instruments for at least three solar cycles. Others have demonstrated that there is sufficient information available to quantify the solar cycle in ozone using these data, albeit with uncertainties (perhaps even large ones). Bodeker et al. (Earth System Sci Data 2013) produce just such a product. Their Tier 1.4 database is the natural component (solar plus volcanic) of vertical ozone profile variability extracted from observations by linear regression, from which the solar component can be further extracted. As well the manuscript cites various other such products reported previously e.g., by Randell and Wu, Hood and coworkers etc. So its not that this task can't be done, its that the present manuscript doesn't do it.

In its present form, the manuscript focuses on comparing multiple (nine plus) ozone datasets, each produced from a variety of data reduction techniques and assumptions, combined in various ways and covering different time periods and different lengths of time. For the most part, these datasets have already been described and compared in detail, including using multiple linear regression analysis. So the dataset comparisons themselves are not new material.

The authors proceed to extract solar signals from all nine plus of these datasets mainly over a common 21-year period 1984 to 2004; that there are notable differences leads them to conclude that the observations are not adequate to extract the solar cycle signal. But just because some datasets over this 21-year time period are not suitable for

this task, does not mean that some longer datasets are also not; more importantly a few of the datasets extend over the period 1979 to 2015 – 37 years – and these (much) longer time series are more suitable and more likely to give meaningful results than those analyzed for just 21 years. A key characteristic of a database suitable for extracting a decadal solar cycle signal is that the record be as long as is possible. Even 37 years is just 3 solar cycles. This is crucial so that the regression analysis can properly separate the solar cycle from other influences on decadal time scales, namely EESC and volcanic activity, each of which has decadal scale variability. It is not surprising that datasets like HALOE that cover only the period 1991 to 2005, or other datasets analyzed over only 14 years (barely one solar cycle) do not yield statistically meaningful results or that results among them differ (e.g. Figure 10).

To achieve the stated goal of the paper – namely the representation the solar signal in ozone using observations - the authors should use the longest available time series of ozone profiles that exists so as to minimize correlation among the predictors and decrease regression model coefficient uncertainties– this is from 1979 to the present 2015. A dataset of 37 years is far superior (even with other extenuating instrumental limitations) to one of 21 years for extracting the decadal solar cycle. Yet nowhere in the paper do the authors ever use the entire dataset available.

A truly observational representation of the solar cycle in ozone, which this paper seeks to achieve, is important for input to physical model simulations and as independent validation of those simulations. In it present form, the manuscript does not provide useful material for climate models to use. One assumes therefore, that they will recommend in Paper II that a representation of the solar cycle signal in ozone be based on model simulations. Since the present manuscript does not address such models - or assess their limitations - this conclusion cannot be justified here, especially given the many known uncertainties and limitations of models. Models require observations for validation and so in order to be suitable for publication, this paper needs to produce this observational product - with as many caveats as needed and with realistic (possibley large) uncertainties– or be withdrawn.

Fortunately, from their exhaustive (albeit non-selective, somewhat unfocused and inconclusive) analysis of multiple datasets, the authors have the material available from which to extract the needed product and to avoid the obfuscation that results from reporting the details of multiple (less worthy) ozone datasets.

One way for the authors to proceed to revise their manuscript and achieve their goal of producing an observational representation of ozone's response to the solar cycle (with uncertainties) might be as follows:

1) Select (on the basis of prior work and current analysis) the longest, most reliable SAGE-extended dataset.

2) Select (on the basis of prior work and their own analysis) the longest most reliable SBUV datasets (possible NOAA Cohesive Merged Dataset – see below).

3) Develop a robust multiple linear regression methodology for extracting solar cycle signals separately from these two independent datasets on their native altitude and spatial grids (for this purpose the current MLR model needs to be expanded and tested – see below).

4) Merge, or otherwise combine/integrate- the solar representations from the two different approaches and propagate uncertainties from the statistical regression coefficients of the two representations.

5) Constrain as needed the ozone profile solar cycle changes to be consistent with that of total ozone (which the present analysis doesn't utilize at all).

6) Compare with Bodeker et al Tier 1.4 representation of vertical ozone profile responses to solar cycle or some other published representation (e.g. that used in CMIP5); discuss and quantify limitations and uncertainties in the final observational product.

**Specific Comments**

*Requirements of Physical Models*

Nowhere does the manuscript define requirements for the task they are undertaking, namely the details of the inputs that physical models need for the ozone representation. What is the altitude grid (and resolution), latitude and longitude grid (and resolution) for which the models need these inputs? Lacking these stated definitions, the authors are without robust criteria upon which to build their product, or to assess how well the final product meets the requirements. Although the SBUV observations have poorer vertical resolution than do the SAGE observations, is this actually an issue for the climate models? What vertical (and spatial) resolution is actually needed? How does the magnitude of the uncertainties affect the usefulness of the observational representation; the authors show regions where significance exceeds 95% but it is quite likely that the patterns of the response, even including regions that are less statistically significant, can still provide useful model input and validation.

Is the ozone product to be represented by absolute values or anomalies – if the latter than is a reference distribution also needed?

*Dataset Comparison and Selection*

As the authors explain, multiple ozone datasets now exist, in basically two categories – those based on the SAGE observations and those based on the SBUV observations. Much of the material in Section 2 – currently called "Methods" - is really about the datasets and might be renamed "Datasets" (with the MLR presented in its own separate and expanded section). A number of other papers already describe these various datasets in great detail - including their multiple linear regression analysis - and to some extent the current paper repeats more of these results than might be necessary, in that this paper is about the solar signal in the datasets, not the datasets per se.

Most importantly, the authors should describe the selection of, and reasons for their selection, of the most suitable (longest) dataset in each category then extract the solar signals that will inform their product for climate model use from just these two datasets. In an unfocused and round about way, the present manuscript actually does this to some extent, but it doesn't use the longest available datasets and it fails to synthesize the many derived representations into a final product.

The SBUV record exemplifies this more streamlined approach. In the present manuscript the authors calculate and examine the solar signals in all three SBUV records when a judicious assessment of prior work would likely lead them to decide, right from the beginning of their study, that the best record for the present purpose is likely the NOAA Merged Cohesive Data from 1979 to 2015 - for a number of reasons (which the authors should lay out clearly and concisely). In a revision of the paper along these lines, the entire Section 3.3 about the SBUV record could appear much earlier in the paper, in a section about Datasets. The general (valid) conclusion presented already in this paper is that there are distinct instrumental differences between the NASA MOD V8.0 and MOD V8.6 datasets – a conclusion that Lean (JAS, 2014) also reached and discussed in detail, and which the authors might simply reference to justify why they probably shouldn't use MOD V8.6. So then, the authors need to obtain the solar signal in just one SBUV-based dataset. By elimination this would seem to be the NOAA Merged Cohesive Dataset which the authors show has a solar response like that of the (shorter) NASA MOD V8.0 dataset. They could then test and evaluate more comprehensively the MLR results for this one dataset (e.g., effects of lags, cross-correlation of predictors, addition of trend term, stability of model coefficients with length of database etc).

A similar selection could be made of the SAGE-based datasets – utilizing the material about these datasets already published and whatever additional tests the authors undertake to clarify their selection. A distinction among the SAGE datasets is that different versions use different temperature databases which the current manuscript expends much effort in analyzing and comparing. Another few pages of the manuscript (pages 11 and 12) compare various characteristics of the datasets such as their trends. But a comprehensive regression analysis should capture and extract these different trends without the trends necessarily affecting the separate (independent) solar component. And as with the SBUV dataset selection, this material about the SAGE-based datasets could be included in a section on Datasets – then in a subsequent section on Results, the authors need only extract and examine the solar signals from one, preferred, SAGE-based dataset  - the longest one.

What about total ozone? Nowhere in the paper is use made of total ozone datasets, even though whatever altitude profiles are specified must be consistent with the total ozone amount. Total ozone datasets have, arguably, greater long term stability than the profile datasets, so that solar signals can - and have been – extracted somewhat reliability; differences among the different datasets can therefore guide the selection of the profile datasets, as suggested above for SBUV datasets. Analysis of the SBUV MOD V8.6 total ozone compared with TOMS MOD V8 strongly suggests instrument effects in SBUV MOD V8.6 around the 1996 time frame that cause a smaller solar response in

MOD V8.6 than in MOD V8.0.

*The Regression Model Formulation and Results*

Multiple linear regression (MLR) is the method that the authors use to extract the sought-for representation of the ozone's response to the solar cycle. Presenting the direct time series together with the solar index (Figure 2), and discussing (page 8) how a solar signal is not directly evident is not all that useful – it is not even evident in total ozone because of the various other influences.

The MLR analysis of the type that the authors use has been used extensively to model ozone variability statistically in terms of individual influences. But the model that the authors use doesn't take advantage of the understanding achieved from those prior studies. As a result it likely doesn't provide the best representation of the various predictors, especially EESC but also ENSO and trends.

EESC: As described for example in Bodeker et al (2013) and Lean (2014), among others, the EESC depends on the age of air and on bromine. It therefore has latitude – and possible altitude – dependencies. The peak of the EESC temporal structure shifts accordingly, which affects how the MLR apportions variance among EESC and other influences. The authors don't say which EESC profile they use but it seems that they use only one profile. However, there is no one EESC profile that is appropriate for all latitudes. Rather a suite of profiles is needed corresponding to different ages of air (and bromine); these can be obtained from GSFC's on-line capability.

ENSO:  the ENSO signal in ozone may lag the MEI index by a few months – while this is not an issue for mid and high latitudes, where ENSO signal is small, it could affect tropical signal. Is zero lag of the MEI index the most appropriate lag?

Solar Irradiance: Why do the authors use the 10.7 cm flux and not modeled UV irradiance or the Mg index, which provide better representations of the true solar changes? While it is likely that for monthly means these differences wont be large, the authors nevertheless need to acknowledge possible limitations of the use of the 10.7 cm index in representing solar UV irradiance.

Trend:  The authors state that they include a trend in some instances but that it makes little difference to their "results" by which they presumably mean the MLR solar cycle coefficients (see Technical comment below about confusion due to the authors' generic use of "results"). This is possibly because the 21-year time period of their analysis is too short. Using the longest available datasets, the authors would be able to – and would need to - separate the EESC and trend components. The manuscript discusses in quite a lot of detail the different trends among the different datasets with the inference that this provides a measure of the dataset quality. But as long as the trend and solar cycle terms are sufficiently orthogonal (low correlation coefficient) then a linear trend shouldn't influence the extraction of the solar cycle representation. The difficulty is that the long-term drifts in the ozone records aren't actually linear trends and that further more, the combined EESC and GHG influences aren't linear either. Other authors have used piece wise linear regression to account for the fact that the actual trend in ozone is

some combination of that due to EESC and due to GHGs. Modelling the longest available ozone dataset to achieve the most reliable solar signal possible will very likely require the inclusion of a realistic trend term.

Seasonality: The present paper cites the need for specifying the seasonality of the ozone response to the solar cycle, and the authors explore this (Figure 12) but without coming to concrete conclusions. Bodeker et al (2013) report that the seasonality is not pronounced for solar cycle variations in their regression model analysis. The authors could investigate this more robustly in a number of ways. One approach would be to add additional cosine and sine terms to modulate the solar index in the regression model. Another would be to perform the regression using 3 months to define the four primary seasons, which would likely be more representative (less noise?) than the figures shown for each sequential month in Figure 12. Either way, some statement is needed about the magnitude of this effect – is it important or not? (in a statistical sense). Will the solar signal representation still be useful as input to the physical model simulations without the seasonal modulation (see above comments about specifying requirements)?

Once a properly formulated MLR is applied to extract the solar cycle signal in each of the two basic (SAGE-based and SBUV-based) datasets, the extracted solar signals then provide baseline solar cycle representations, with associated statistical uncertainties. Post processing of these two solar representation could include (uncertainty weighted) averaging and/or merging and error propagation to achieve a coherent synthesis and final product that the present manuscript lacks.

*Time Span of the Regression Analysis*

For extracting the most reliable decadal solar signal the database should be as long as possible to minimize cross projection of the solar and other influences. As noted above, by limiting their analysis of ozone's response to solar variability to data mainly from 1984 to 2004 the authors are not utilizing the longest datasets available (by far!). Ozone data extend from 1979 to 2015 – during the additional 10 years from 2004 to 2014 the solar cycle has increased to another maximum, volcanic aerosols have been minimal, EESCs have continued to decline, and greenhouse gases to increase.

The authors investigate the effect of time spans on their results but their approach is not systematic and they conclude only that different time spans give different results. Their Figure 11 shows the patterns of ozone's solar cycle response derived from 21-year datasets over different epochs. It is probably not surprising that using different 21-year epochs gives different statistical models since, for one thing, the correlation among the various predictor times series likely differs for each 21-year period and this may affect the derive coefficients. This could occur even if the ozone time series were "perfect" so it is not necessarily true, as the authors conclude, that the differences in the ozone representations obtained from different 21-year epochs is due entirely to uncertainties in the database.

Examination of the stability of the statistical model coefficients is indeed important but

the approach of using separate 21-year epochs is perhaps not the best way to establish this. As a dataset lengthens, the magnitude of the coefficient of a given predictor should converge to a "stable" value. The usual way to evaluate the stability of the coefficients is to start with a core dataset of maximum length and then reevaluate the model coefficients using successively shorter lengths of the primary datasets. In revising their manuscript, the authors might consider developing a more quantitative metric (solar signal in total ozone derived by integrating the vertical profiles?) to establish the stability of the model coefficients as a function of length of the dataset. When accomplished using the longest possible dataset, such a metric will automatically provide feedback about limitations using shorter datasets (such as HALOE).

*Derived Product –Quantitative Assessment, Uncertainties, Comparison with Similar Products.*

The authors make numerous comparisons and describe differences among multiple ozone (and temperature) datasets but they do not synthesize their results or reach quantitative conclusions about the solar-ozone representation for input to climate model simulations. Having determined a best possible solar cycle representation of ozone and corresponding uncertainties, they could asses how this compares with independent representations and whether on not the uncertainties make the derived solar signal representation useful for input to the model calculations. The answer to this will depend on the magnitude of the uncertainties in conjunction with the requirements of the models (as noted above). Even if the uncertainty in the solar cycle signal is 50%, this may still be useful if, say, the models differ by 100% among themselves.

Independent validation is possible by comparison with Tier1.4 or other solar cycle representation of ozone published previously. This is different from the analysis that the authors have made to extract the solar signal, using their own regression model, from the Tier 0 database. Rather, comparing their solar cycle representation of ozone with that of the Bodeker et al. and others takes into account the different regression model formulation as well as the different datasets. Additionally, since it is likely that climate models will use the Bodeker et al. ozone profile dataset (it was developed expressly for that purpose) it would be good to compare the solar cycle representation in that dataset with the one developed from this work.

**Technical Comments:**

The text of this manuscript is not as precise as it might be and the authors should provide additional clarity when revising the manuscript. It can be quite difficult to discern the actual analyses being described. The discussion of Figure 3 is an example of this. The text states that "Figures 3(a) and (b) show latitude-altitude plots of the percentage differences in ozone number density between solar maximum and minimum for SAGE II v6.2 and v7.0, respectively". What is not at all clear is how these percentage differences were obtained. On first reading they seemed to be differences of the direct time series themselves (shown in Figure 2) during solar maximum and minimum conditions. If so, then because of the other influences on ozone, such direct differences

are not reliable indicators of the solar cycle signal. Are they, instead, derived using the MLR? The text doesn't say.

Another example of this general lack of precision of the text is frequent generic reference to "results" (as noted above). An example is on page 16: "to test this, we add a linear trend term into the MLR; however, this does not strongly affect the results as compared to Figure 11 (not shown)."  By results they presumably mean the coefficients of the solar predictor in the statistical model – so this is what they should say in the text. More generally, the paper presents many "results" about many things and each one should be properly articulated. Even if this lengthens the manuscript, it makes the message far more clear for the reader.

Line 20 ff: the authors refer throughout to the SBUV VN8.0 and 8.6 datasets – do they mean SBUV MOD8.0 and 8.6 – where MOD is Merged Ozone Data?

Line 29-30: The authors make the mistake of confusing the absolute energy changes in the total and UV spectrum with their relative changes and suggesting that TSI changes are somehow much less than UV changes. The change in TSI from solar max to min is larger in absolute energy units than is the change in the UV spectrum. The manuscript needs to clarify this.

Lines 454-477: These differences have been reported and interpreted previously in the corresponding total ozone datasets of the two versions. Analysis of the solar (and other signals) in the Ozone MOD V8.0 and MOD V8.6 total ozone datasets (Lean, JAS, 2014) shows the regional and global differences for the solar cycle signal in the two different datasets, finding a smaller solar cycle amplitude in the MOD V8.6 of 3DU versus 5DU on Mod V8. That paper discussed these differences, including the calibration issues.

Page 20, lines 670-673. The authors should remove their "encouragement" of instrument teams to better analyze their data! The instruments teams are already undertaking a very challenging and difficult task of space-based metrology and are, without doubt, (more than) fully aware of the need for properly specifying instrumental effects in their datasets as well as they can!

A more helpful conclusion would be for the authors to generate an actual quantitative product – namely the solar representation of the solar cycle in ozone that is the stated goal of their paper -  and assess the future needs of ozone observations in the context of the associated uncertainties of that product.

---

## Referee Comment (RC4) · Anonymous Referee #1 · 29 Feb 2016

The paper compares the solar cycle signal in several stratospheric ozone data sets. It is obviously the first part of a two part paper that forms the basis of a model assessment study (to follow). It is also related to the upcoming CMIP6 ozone forcing paper. The paper goes relatively deep into the comparison of data products, and I must stress that I am not an expert on these products. In my view, the comparison of different product or product versions with respect to solar cycle signals is a worthwhile task and the conclusions are strong but interesting. One focus of the paper is on the seasonality, which (according to the authors) should be captured by any potential climate model forcing data set, but also the general altitude-latitude structure. The authors conclude that there are considerable differences between data sets in this respect. The paper
is well written, albeit quite technical at times. It is scientifically sound. In my view the paper is publishable after minor revisions.

1. My main comments concerns how the outline of the paper and its position amongst the other paper (Part 2, CMIP6): If the assessment of data sets for use as forcings in CMIP6 is the main topic, then perhaps it should be made more clear at the beginning, which properties such a data set should have and which not and how such a data set is (or might be) generated. Will some smoothed satellite-based data set be used and then extended forward and backward? Will an existing model simulation or ensemble be used? Or will an ozone data set be generated purely statistically? Perhaps a few words on that would help, otherwise the paper is in danger of being misunderstood. The starting point (and recommendation) of the paper is that in any case a realistic solar-cycle imprint should be in, and the conclusion (in the abstract) is that satellitebased ozone data sets alone will not be good enough to get that signal. I find that interesting and well demonstrated in the analysis. However, I would like to know a little bit more about other effects, although this is not the topic of the paper: Obviously ozone depleting substances should be in such a data set. It is also clear that climatic influences such as SSTs should be excluded because the coupled models will generate their own SSTs. What about the QBO, should it be in or out? How will volcanic eruptions be specified in CMIP6? This sounds a bit off topic, but it might help the reader to position the paper amongst the other two upcoming papers before the focus then goes entirely towards the many data sets and the solar cycle imprint. Also, it would help to assess the relevance of the uncertainties found against other uncertainties. Once the position of the paper is made clear, I can agree with most of the paper.

2. Temperature used for conversion: It is often not clear how temperatures were used to convert number densities to mixing ratio. In which cases were daily profiles used, in which monthly, or even just a climatology? In which cases were zonally averaged temperatures used, in which the full 3D fields? Perhaps add a table.

3. Regression models: Perhaps it is common practice to write the model in this way. I
am still surprised that no lags are used. Also, the volcanic term is basically Pinatubo, so perhaps it might be better just to cut that period out.

---

## Author Comment (AC2) · 10 Jun 2016

The comment was uploaded in the form of a supplement:
http://www.atmos-chem-phys-discuss.net/acp-2015-882/acp-2015-882-AC2-supplement.pdf

---

## Author Comment (AC6) · 10 Jun 2016

The comment was uploaded in the form of a supplement:
http://www.atmos-chem-phys-discuss.net/acp-2015-882/acp-2015-882-AC6-supplement.pdf

---

## Author Response (AR1)

**Summary**

The paper compares the solar cycle signal in several stratospheric ozone data sets. It is obviously the first part of a two-part paper that forms the basis of a model assessment study (to follow). It is also related to the upcoming CMIP6 ozone forcing paper. The paper goes relatively deep into the comparison of data products, and I must stress that I am not an expert on these products. In my view, the comparison of different product or product versions with respect to solar cycle signals is a worthwhile task and the conclusions are strong but interesting. One focus of the paper is on the seasonality, which (according to the authors) should be captured by any potential climate model forcing data set, but also the general altitude-latitude structure. The authors conclude that there are considerable differences between data sets in this respect. The paper is well written, albeit quite technical at times. It is scientifically sound. In my view the paper is publishable after minor revisions.

*We thank the reviewer for reading the manuscript and providing their helpful comments and suggestions. We address their specific issues in turn below.*

**Specific Comments**

1. My main comment concerns how the outline of the paper and its position amongst the other paper (Part 2, CMIP6): If the assessment of data sets for use as forcings in CMIP6 is the main topic, then perhaps it should be made more clear at the beginning, which properties such a data set should have and which not and how such a data set is (or might be) generated. Will some smoothed satellite-based data set be used and then extended forward and backward? Will an existing model simulation or ensemble be used? Or will an ozone data set be generated purely statistically? Perhaps a few words on that would help, otherwise the paper is in danger of being misunderstood. The starting point (and recommendation) of the paper is that in any case a realistic solar-cycle imprint should be in, and the conclusion (in the abstract) is that satellite-based ozone data sets alone will not be good enough to get that signal. I find that interesting and well demonstrated in the analysis. However, I would like to know a little bit more about other effects, although this is not the topic of the paper: Obviously ozone depleting substances should be in such a data set. It is also clear that climatic influences such as SSTs should be excluded because the coupled models will generate their own SSTs. What about the QBO, should it be in or out? How will volcanic eruptions be specified in CMIP6? This sounds a bit off topic, but it might help the reader to position the paper amongst the other two upcoming papers before the focus then goes entirely towards the many data sets and the solar cycle imprint. Also, it would help to assess the relevance of the uncertainties found against other uncertainties. Once the position of the paper is made clear, I can agree with most of the paper.

*The reviewer raises many important points regarding the ozone database for CMIP6. As part of the CMIP6 special issue in Geoscientific Model Development Discussions (GMDD) there will be a paper that describes the CMIP6 ozone dataset in full (cited as Hegglin et al. (in prep.) in the manuscript) and a paper that describes the solar forcing for CMIP6*

*(Matthes et al (2016)). The ozone database will be based on chemistry-climate model simulations from the WCRP/SPARC Chemistry Climate Model Initiative (CCMI), and will include effects of ozone depleting substances, greenhouse gases and the solar cycle. The authors are not sure whether it will also include the effects of QBO and volcanic eruptions, since these are represented differently across models.*

*Because the CMIP6 ozone dataset has not yet been finalized and will be fully documented in the GMDD special issue, we are cautious of adding additional information to the current manuscript. However, the revised manuscript actually places much less emphasis on the assessment of satellite ozone datasets for CMIP6, and instead focuses more on the interpretation of the datasets themselves. Part II will provide more detail about a solar-ozone response recommendation for CMIP6. However, we have added some more detailed text at the end of Section 1 that describes the main properties that an ozone dataset for climate models must possess which helps to put the analysis into a modeling context.*

2. Temperature used for conversion: It is often not clear how temperatures were used to convert number densities to mixing ratio. In which cases were daily profiles used, in which monthly, or even just a climatology? In which cases were zonally averaged temperatures used, in which the full 3D fields? Perhaps add a table.

*All of the conversions presented in Section 3.1.1 use zonal and monthly mean temperature profiles to convert the zonal and monthly mean SAGE II profiles. We have expanded the text in this section to give a more detailed description of the temperature fields used to conduct the conversions to mixing ratios. We hope that the reviewer finds this explanation clearer.*

3. Regression models: Perhaps it is common practice to write the model in this way. I am still surprised that no lags are used. Also, the volcanic term is basically Pinatubo, so perhaps it might be better just to cut that period out.

*Near identical multiple linear regression models have been recently applied to satellite ozone datasets (see e.g. SI²N papers by Tummon et al., 2015; Harris et al., 2015 in ACP). The main term in the regression model for which a lag might be appropriate is ENSO. We have tested the sensitivity of the results to lags in the ENSO index in the range 0-12 months, but find no significant effects on the diagnosed solar-ozone response. We therefore do not include any lags in the regression model. Text has been added to the Methods section of the revised manuscript to explain this.*

*At the suggestion of the reviewer, we now exclude the periods following major volcanic eruptions from the analysis and no longer include the volcanic regressor in the MLR model. However, these changes have little impact on the results.*

**Review by Judith Lean**

**General Comments**

*As a result of the criticisms made by the reviewer we have made substantial modifications to the manuscript, as described below and in the 'General response to all reviewers'. While we accept some of the criticisms raised by the reviewer, and have modified the manuscript accordingly, we maintain that documenting and comparing solar-ozone signals in different satellite ozone datasets is a valuable contribution to the scientific literature. This is particularly motivated by the fact that there have been recent updates to the two main long-term satellite records, SBUV and SAGE II. Given the length and degree of detail of many of the reviewer's comments below, we have only attempted to respond to the main points raised. However, we emphasise that the revised manuscript has been substantially modified compared to the original, and we therefore encourage the reviewer to re-read the entire manuscript.*

The task of this paper is, according to the title "The representation of solar cycle signals in ozone". The present paper is Part I of two papers, this first focusing on using observations to determine the solar cycle representation, the second using models. The overall goal of the work is to prescribe the ozone changes during the solar cycle for use as input to climate models that do not themselves calculate the ozone responses directly, thereby allowing these models to include indirect effects of solar forcing.

The manuscript states that the "goal is to synthesize current knowledge to inform a recommendation for including the solar-ozone signal in the prescribed ozone dataset being created for CMIP6". It does not accomplish this task. A representation of the solar cycle signals based on a synthesis of observations of stratospheric ozone is not recommended. Instead, the paper concludes that it is "unlikely that satellite ozone measurements alone can be applied to estimate the necessary solar cycle ozone component of the prescribed ozone database for future coupled model intercomparisons".

These aspects of the paper preclude publication in its present form. Unless, or until the authors can/do produce the product that they set out to produce then it would seem that a paper about not achieving their goal is unwarranted. Their conclusion that satellite data are not able to inform such a product is incorrect. Ozone has been measured in one form or another by multiple instruments for at least three solar cycles. Others have demonstrated that there is sufficient information available to quantify the solar cycle in ozone using these data, albeit with uncertainties (perhaps even large ones). Bodeker et al. (Earth System Sci Data 2013) produce just such a product. Their Tier 1.4 database is the natural component (solar plus volcanic) of vertical ozone profile variability extracted from observations by linear regression, from which the solar component can be further extracted. As well the manuscript cites various other such products reported previously e.g., by Randell and Wu, Hood and coworkers etc. So it's not that this task can't be done, it's that the present manuscript doesn't do it.

*We have reframed the objectives of our study in the revised manuscript. We no longer state the aim of "synthesizing current knowledge to inform a recommendation for*

**Review by Judith Lean**

**General Comments**

*As a result of the criticisms made by the reviewer we have made substantial modifications to the manuscript, as described below and in the 'General response to all reviewers'. While we accept some of the criticisms raised by the reviewer, and have modified the manuscript accordingly, we maintain that documenting and comparing solar-ozone signals in different satellite ozone datasets is a valuable contribution to the scientific literature. This is particularly motivated by the fact that there have been recent updates to the two main long-term satellite records, SBUV and SAGE II. Given the length and degree of detail of many of the reviewer's comments below, we have only attempted to respond to the main points raised. However, we emphasise that the revised manuscript has been substantially modified compared to the original, and we therefore encourage the reviewer to re-read the entire manuscript.*

The task of this paper is, according to the title "The representation of solar cycle signals in ozone". The present paper is Part I of two papers, this first focusing on using observations to determine the solar cycle representation, the second using models. The overall goal of the work is to prescribe the ozone changes during the solar cycle for use as input to climate models that do not themselves calculate the ozone responses directly, thereby allowing these models to include indirect effects of solar forcing.

The manuscript states that the "goal is to synthesize current knowledge to inform a recommendation for including the solar-ozone signal in the prescribed ozone dataset being created for CMIP6". It does not accomplish this task. A representation of the solar cycle signals based on a synthesis of observations of stratospheric ozone is not recommended. Instead, the paper concludes that it is "unlikely that satellite ozone measurements alone can be applied to estimate the necessary solar cycle ozone component of the prescribed ozone database for future coupled model intercomparisons".

These aspects of the paper preclude publication in its present form. Unless, or until the authors can/do produce the product that they set out to produce then it would seem that a paper about not achieving their goal is unwarranted. Their conclusion that satellite data are not able to inform such a product is incorrect. Ozone has been measured in one form or another by multiple instruments for at least three solar cycles. Others have demonstrated that there is sufficient information available to quantify the solar cycle in ozone using these data, albeit with uncertainties (perhaps even large ones). Bodeker et al. (Earth System Sci Data 2013) produce just such a product. Their Tier 1.4 database is the natural component (solar plus volcanic) of vertical ozone profile variability extracted from observations by linear regression, from which the solar component can be further extracted. As well the manuscript cites various other such products reported previously e.g., by Randell and Wu, Hood and coworkers etc. So it's not that this task can't be done, it's that the present manuscript doesn't do it.

*We have reframed the objectives of our study in the revised manuscript. We no longer state the aim of "synthesizing current knowledge to inform a recommendation for*

*including the solar-ozone signal in the prescribed ozone dataset being created for CMIP6".
Instead we focus on comparing the solar-ozone responses in recently updated long-term
satellite datasets (SAGE II, SBUV) with their predecessors. The study documents these
differences and where possible gives insights into why these occur. These comparisons are
important to document because many previous studies of the solar-ozone response have
been published using the older datasets, but many fewer with the newer datasets, and in
some cases substantial differences are found between them. This will provide a valuable
point of reference for other researchers who are studying solar cycle effects on climate.*

In its present form, the manuscript focuses on comparing multiple (nine plus) ozone
datasets, each produced from a variety of data reduction techniques and assumptions,
combined in various ways and covering different time periods and different lengths of
time. For the most part, these datasets have already been described and compared in
detail, including using multiple linear regression analysis. So the dataset comparisons
themselves are not new material.

*While multiple linear regression analysis has been applied to many of the datasets used in
this study (e.g. Tummon et al., 2015), the main goal of this other work has been to
document linear ozone trends, and most of them therefore do not discuss the solar-ozone
response whatsoever. There has been no recent comparison of the latitude-height
structures of solar-ozone responses across multiple satellite datasets (see e.g. the last
major effort by Soukharev and Hood (2006)). We therefore disagree with the reviewer
that these comparisons do not present new material. To give just one example from our
study, the comparison of the solar-ozone responses in SAGE II v6.2 and v7.0 is new and
provides useful and important insight.*

**Time Span of the Regression Analysis**

For extracting the most reliable decadal solar signal the database should be as long as
possible to minimize cross projection of the solar and other influences. As noted above,
by limiting their analysis of ozone's response to solar variability to data mainly from
1984 to 2004 the authors are not utilizing the longest datasets available (by far!). Ozone
data extend from 1979 to 2015 – during the additional 10 years from 2004 to 2014 the
solar cycle has increased to another maximum, volcanic aerosols have been minimal,
EESCs have continued to decline, and greenhouse gases to increase.

The authors investigate the effect of time spans on their results but their approach is
not systematic and they conclude only that different time spans give different results.
Their Figure 11 shows the patterns of ozone's solar cycle response derived from 21-
year datasets over different epochs. It is probably not surprising that using different 21-
year epochs gives different statistical models since, for one thing, the correlation among
the various predictor times series likely differs for each 21-year period and this may
affect the derive coefficients. This could occur even if the ozone time series were
"perfect" so it is not necessarily true, as the authors conclude, that the differences in the
ozone representations obtained from different 21-year epochs is due entirely to
uncertainties in the database.

Examination of the stability of the statistical model coefficients is indeed important but
the approach of using separate 21-year epochs is perhaps not the best way to establish

this. As a dataset lengthens, the magnitude of the coefficient of a given predictor should converge to a "stable" value. The usual way to evaluate the stability of the coefficients is to start with a core dataset of maximum length and then reevaluate the model coefficients using successively shorter lengths of the primary datasets. In revising their manuscript, the authors might consider developing a more quantitative metric (solar signal in total ozone derived by integrating the vertical profiles?) to establish the stability of the model coefficients as a function of length of the dataset. When accomplished using the longest possible dataset, such a metric will automatically provide feedback about limitations using shorter datasets (such as HALOE).

The authors proceed to extract solar signals from all nine plus of these datasets mainly over a common 21-year period 1984 to 2004; that there are notable differences leads them to conclude that the observations are not adequate to extract the solar cycle signal. But just because some datasets over this 21-year time period are not suitable for this task, does not mean that some longer datasets are also not; more importantly a few of the datasets extend over the period 1979 to 2015 – 37 years – and these (much) longer time series are more suitable and more likely to give meaningful results than those analyzed for just 21 years. A key characteristic of a database suitable for extracting a decadal solar cycle signal is that the record be as long as is possible. Even 37 years is just 3 solar cycles. This is crucial so that the regression analysis can properly separate the solar cycle from other influences on decadal time scales, namely EESC and volcanic activity, each of which has decadal scale variability. It is not surprising that datasets like HALOE that cover only the period 1991 to 2005, or other datasets analyzed over only 14 years (barely one solar cycle) do not yield statistically meaningful results or that results among them differ (e.g. Figure 10). To achieve the stated goal of the paper – namely the representation the solar signal in ozone using observations - the authors should use the longest available time series of ozone profiles that exists so as to minimize correlation among the predictors and decrease regression model coefficient uncertainties– this is from 1979 to the present 2015. A dataset of 37 years is far superior (even with other extenuating instrumental limitations) to one of 21 years for extracting the decadal solar cycle. Yet nowhere in the paper do the authors ever use the entire dataset available.

*As requested by the reviewer, in the revised manuscript all ozone datasets are analysed for their entire lengths. We have removed the section discussing the stability of the regression coefficients. Furthermore, the revised manuscript is more selective about which datasets are included, and results from short records, such as HALOE, are no longer presented. Instead we focus on comparing SAGE II v6.2 and v7.0, a subset of SI2N extended SAGE records, and SBUV VN8.0 and VN8.6.*

A truly observational representation of the solar cycle in ozone, which this paper seeks to achieve, is important for input to physical model simulations and as independent validation of those simulations. In it present form, the manuscript does not provide useful material for climate models to use. One assumes therefore, that they will recommend in Paper II that a representation of the solar cycle signal in ozone be based on model simulations. Since the present manuscript does not address such models - or assess their limitations - this conclusion cannot be justified here, especially given the many known uncertainties and limitations of models. Models require observations for validation and so in order to be suitable for publication, this paper needs to produce

this observational product - with as many caveats as needed and with realistic (possibly large) uncertainties– or be withdrawn.

*In the revised manuscript, we no longer make reference to a recommendation for CMIP6. Furthermore, we make clearer recommendations for which SAGE II and SBUV datasets are likely to be most reliable for diagnosing the solar-ozone response.*

Fortunately, from their exhaustive (albeit non-selective, somewhat unfocused and inconclusive) analysis of multiple datasets, the authors have the material available from which to extract the needed product and to avoid the obfuscation that results from reporting the details of multiple (less worthy) ozone datasets.

*The revised manuscript has narrower aims and presents results from only a subset of the ozone datasets in the original manuscript. We believe that this results in a more selective, focused, and conclusive study.*

One way for the authors to proceed to revise their manuscript and achieve their goal of producing an observational representation of ozone's response to the solar cycle (with uncertainties) might be as follows:

1) Select (on the basis of prior work and current analysis) the longest, most reliable SAGE-extended dataset.

2) Select (on the basis of prior work and their own analysis) the longest most reliable SBUV datasets (possibly NOAA Cohesive Merged Dataset – see below).

3) Develop a robust multiple linear regression methodology for extracting solar cycle signals separately from these two independent datasets on their native altitude and spatial grids (for this purpose the current MLR model needs to be expanded and tested – see below).

4) Merge, or otherwise combine/integrate- the solar representations from the two different approaches and propagate uncertainties from the statistical regression coefficients of the two representations.

5) Constrain as needed the ozone profile solar cycle changes to be consistent with that of total ozone (which the present analysis doesn't utilize at all).

6) Compare with Bodeker et al Tier 1.4 representation of vertical ozone profile responses to solar cycle or some other published representation (e.g. that used in CMIP5); discuss and quantify limitations and uncertainties in the final observational product.

*Since the objectives of the manuscript have altered compared to the original submission, we have incorporated some but not all of the reviewer's recommendations (see below).*

**Specific Comments**

*Requirements of Physical Models*

Nowhere does the manuscript define requirements for the task they are undertaking, namely the details of the inputs that physical models need for the ozone representation. What is the altitude grid (and resolution), latitude and longitude grid (and resolution) for which the models need these inputs? Lacking these stated definitions, the authors are without robust criteria upon which to build their product, or to assess how well the final product meets the requirements. Although the SBUV observations have poorer vertical resolution than do the SAGE observations, is this actually an issue for the climate models? What vertical (and spatial) resolution is actually needed? How does the magnitude of the uncertainties affect the usefulness of the observational representation; the authors show regions where significance exceeds 95% but it is quite likely that the patterns of the response, even including regions that are less statistically significant, can still provide useful model input and validation.

Is the ozone product to be represented by absolute values or anomalies – if the latter than is a reference distribution also needed?

*The revised manuscript places much less emphasis on the goal of creating an ozone dataset for models, and thus some of the comments raised above are no longer applicable. Nevertheless, we have added a paragraph at the end of Section 1 that states "Given the potential application of the results described below for use in climate model simulations, it is prudent to briefly review the typical requirements of an ozone database for models by describing the CMIP5 dataset as a representative example (Cionni et al., 2011) (see also Bodeker et al. (2013)). The CMIP5 ozone database provided monthly mean ozone mixing ratios on a regular latitude/pressure grid at a horizontal resolution of 5◦×5◦ (lon/lat) on 24 pressure levels covering 1000-1 hPa for the period 1850-2100. Data were provided on the following pressure levels: 1000, 850, 700, 600, 500, 400, 300, 250, 200, 150, 100, 80, 70, 50, 30, 20, 15, 10, 7, 5, 3, 2, 1.5, 1 hPa. Stratospheric ozone data (at p≤300 hPa) were given as zonal mean values. Therefore for any de- scription of the SOR must fulfil these (or similar) criteria to be viable for use in climate models (i.e. global coverage at monthly mean resolution and with sufficient vertical and horizontal resolution throughout the stratosphere)."*

*Dataset Comparison and Selection*

As the authors explain, multiple ozone datasets now exist, in basically two categories – those based on the SAGE observations and those based on the SBUV observations. Much of the material in Section 2 – currently called "Methods" - is really about the datasets and might be renamed "Datasets" (with the MLR presented in its own separate and expanded section). A number of other papers already describe these various datasets in great detail - including their multiple linear regression analysis - and to some extent the current paper repeats more of these results than might be necessary, in that this paper is about the solar signal in the datasets, not the datasets per se.

*We have renamed Section 2 "Ozone datasets". As noted above, although MLR analysis has been applied to ozone datasets in the past, there has been no recent effort to analyse and compare the structures of the solar-ozone signals in different records (see e.g. Soukharev and Hood (2006)). This is certainly the case for SAGE II v6.2 and v7.0, as well as the SI²N datasets.*

Most importantly, the authors should describe the selection of, and reasons for their selection, of the most suitable (longest) dataset in each category then extract the solar signals that will inform their product for climate model use from just these two datasets. In an unfocused and round about way, the present manuscript actually does this to some extent, but it doesn't use the longest available datasets and it fails to synthesize the many derived representations into a final product.

The SBUV record exemplifies this more streamlined approach. In the present manuscript the authors calculate and examine the solar signals in all three SBUV records when a judicious assessment of prior work would likely lead them to decide, right from the beginning of their study, that the best record for the present purpose is likely the NOAA Merged Cohesive Data from 1979 to 2015 - for a number of reasons (which the authors should lay out clearly and concisely). In a revision of the paper along these lines, the entire Section 3.3 about the SBUV record could appear much earlier in the paper, in a section about Datasets. The general (valid) conclusion presented already in this paper is that there are distinct instrumental differences between the NASA MOD V8.0 and MOD V8.6 datasets – a conclusion that Lean (JAS, 2014) also reached and discussed in detail, and which the authors might simply reference to justify why they probably shouldn't use MOD V8.6. So then, the authors need to obtain the solar signal in just one SBUV- based dataset. By elimination this would seem to be the NOAA Merged Cohesive Dataset which the authors show has a solar response like that of the (shorter) NASA MOD V8.0 dataset. They could then test and evaluate more comprehensively the MLR results for this one dataset (e.g., effects of lags, cross-correlation of predictors, addition of trend term, stability of model coefficients with length of database etc).

A similar selection could be made of the SAGE-based datasets – utilizing the material about these datasets already published and whatever additional tests the authors undertake to clarify their selection. A distinction among the SAGE datasets is that different versions use different temperature databases which the current manuscript expends much effort in analyzing and comparing. Another few pages of the manuscript (pages 11 and 12) compare various characteristics of the datasets such as their trends. But a comprehensive regression analysis should capture and extract these different trends without the trends necessarily affecting the separate (independent) solar component. And as with the SBUV dataset selection, this material about the SAGE-based datasets could be included in a section on Datasets – then in a subsequent section on Results, the authors need only extract and examine the solar signals from one, preferred, SAGE-based dataset - the longest one.

*The revised manuscript presents results for a smaller number of ozone datasets (8 instead of 13) and analyses them all over their full lengths. The selection of datasets is justified by the revised goals of the study to compare solar-ozone signals in recently updated datasets (SAGE II v6.2 vs. v7.0 and SBUV VN8.0 vs. VN8.6), and to evaluate recent extended SAGE II datasets. Based on the comparisons amongst these versions, and consideration of the main sources of uncertainty, we now include recommendations for which are the most reliable datasets to use for diagnosing the solar-ozone response. We do not agree with the reviewer that the SBUV Merged Cohesive Data VN8.6 dataset is necessarily better than SBUVMOD VN8.6 for analyzing the solar-ozone response. Tummon et al. (2015) highlight a number of issues with the representation of long-term trends in the Merged Cohesive*

*Dataset, which may also affect the solar-ozone response. We therefore present both SBUVMOD VN8.6 and Merged Cohesive VN8.6 and discuss their respective pros and cons.*

What about total ozone? Nowhere in the paper is use made of total ozone datasets, even though whatever altitude profiles are specified must be consistent with the total ozone amount. Total ozone datasets have, arguably, greater long term stability than the profile datasets, so that solar signals can - and have been – extracted somewhat reliability; differences among the different datasets can therefore guide the selection of the profile datasets, as suggested above for SBUV datasets. Analysis of the SBUV MOD V8.6 total ozone compared with TOMS MOD V8 strongly suggests instrument effects in SBUV MOD V8.6 around the 1996 time frame that cause a smaller solar response in MOD V8.6 than in MOD V8.0.

*We now include a paragraph that discusses literature on the solar-ozone response in TCO. However, most of the results in our study are related to differences in the solar-ozone response in the upper stratosphere, which make only a small contribution to the TCO signal (e.g. Hood (1997)). Therefore to keep the study focused on the latitude-height structures of the solar-ozone response we do not include new analysis of total column ozone. As noted by the reviewer, the solar-ozone response in some of these column ozone datasets has already been analysed and described elsewhere (e.g. Lean 2014), and we now include text referencing these results.*

**The Regression Model Formulation and Results**

Multiple linear regression (MLR) is the method that the authors use to extract the sought-for representation of the ozone's response to the solar cycle. Presenting the direct time series together with the solar index (Figure 2), and discussing (page 8) how a solar signal is not directly evident is not all that useful – it is not even evident in total ozone because of the various other influences.

The MLR analysis of the type that the authors use has been used extensively to model ozone variability statistically in terms of individual influences. But the model that the authors use doesn't take advantage of the understanding achieved from those prior studies. As a result it likely doesn't provide the best representation of the various predictors, especially EESC but also ENSO and trends.

EESC: As described for example in Bodeker et al (2013) and Lean (2014), among others, the EESC depends on the age of air and on bromine. It therefore has latitude – and possible altitude – dependencies. The peak of the EESC temporal structure shifts accordingly, which affects how the MLR apportions variance among EESC and other influences. The authors don't say which EESC profile they use but it seems that they use only one profile. However, there is no one EESC profile that is appropriate for all latitudes. Rather a suite of profiles is needed corresponding to different ages of air (and bromine); these can be obtained from GSFC's on-line capability.

*The reviewer is correct that we have used a single EESC index for all latitudes/altitudes, which neglects the effects of age of air on EESC. We have tested the sensitivity of the MLR results to this assumption by using latitude-height dependent EESC timeseries taken from a chemistry-climate model (UM-UKCA) simulation (REF-C1 CCMI integration). This model*

*simulation implicitly includes effects of variations in stratospheric age of air on EESC. However, using this more sophisticated EESC index does not affect the diagnosed solar-ozone response in the datasets, and we therefore retain the use of a single EESC index for simplicity. A line has been added to the text that describes this sensitivity test to justify our choice.*

ENSO: the ENSO signal in ozone may lag the MEI index by a few months – while this is not an issue for mid and high latitudes, where ENSO signal is small, it could affect tropical signal. Is zero lag of the MEI index the most appropriate lag?

*We have tested the effect of lagging the Nino 3.4 index by 0-12 months and find that this does not affect the diagnosed solar-ozone response. We now state this in the revised manuscript. We also note that other recent MLR studies of long-term trends in satellite ozone datasets have also not used a lag for ENSO (e.g. Tummon et al., 2015; Harris et al., 2015).*

Solar Irradiance: Why do the authors use the 10.7 cm flux and not modeled UV irradiance or the Mg index, which provide better representations of the true solar changes? While it is likely that for monthly means these differences wont be large, the authors nevertheless need to acknowledge possible limitations of the use of the 10.7 cm index in representing solar UV irradiance.

*We have added a sentence "We adopt the widely used F10.7cm solar flux as a proxy for solar activity in the MLR model. This is a more appropriate measure for variations in the UV spectral region, the key driver of the stratospheric ozone response, than other indices such as total solar irradiance (Gray et al., 2010); however, it should be noted that the F10.7cm flux is not a direct measurement of UV variability, but rather is a proxy for variations at these wavelengths."*

Trend: The authors state that they include a trend in some instances but that it makes little difference to their "results" by which they presumably mean the MLR solar cycle coefficients (see Technical comment below about confusion due to the authors' generic use of "results"). This is possibly because the 21-year time period of their analysis is too short. Using the longest available datasets, the authors would be able to – and would need to - separate the EESC and trend components. The manuscript discusses in quite a lot of detail the different trends among the different datasets with the inference that this provides a measure of the dataset quality. But as long as the trend and solar cycle terms are sufficiently orthogonal (low correlation coefficient) then a linear trend shouldn't influence the extraction of the solar cycle representation. The difficulty is that the long-term drifts in the ozone records aren't actually linear trends and that further more, the combined EESC and GHG influences aren't linear either. Other authors have used piece wise linear regression to account for the fact that the actual trend in ozone is some combination of that due to EESC and due to GHGs. Modelling the longest available ozone dataset to achieve the most reliable solar signal possible will very likely require the inclusion of a realistic trend term.

*We now include a $CO_2$ term in the MLR model. However, it does not have a large effect on the diagnosed solar-ozone response.*

Seasonality: The present paper cites the need for specifying the seasonality of the ozone response to the solar cycle, and the authors explore this (Figure 12) but without coming to concrete conclusions. Bodeker et al (2013) report that the seasonality is not pronounced for solar cycle variations in their regression model analysis. The authors could investigate this more robustly in a number of ways. One approach would be to add additional cosine and sine terms to modulate the solar index in the regression model. Another would be to perform the regression using 3 months to define the four primary seasons, which would likely be more representative (less noise?) than the figures shown for each sequential month in Figure 12. Either way, some statement is needed about the magnitude of this effect – is it important or not? (in a statistical sense). Will the solar signal representation still be useful as input to the physical model simulations without the seasonal modulation (see above comments about specifying requirements)?

*Part of the motivation for including a section on "Seasonality in the solar-ozone response" in the manuscript is that this is a relatively unexplored area of research, and one that the authors believe warrants further investigation. Specifically, there is no scientific basis to answer the reviewer's important question: "Will the solar signal representation still be useful as input to the physical model simulations without the seasonal modulation?" because a comparison of e.g. the climate response to solar forcing in a model with and without a seasonal modulation of the SOR has not been performed.*

*It is therefore difficult within the scope of this study (i.e. without performing the above model calculations) to quantify how important this effect is for climate simulations. Instead, we include a short section that explains why a seasonal component to the SOR is to be expected from photochemical theory, and possibly also from coupling between ozone transport and dynamics (see e.g. Hood et al. (2015)). In a similar manner to Hood et al (2015), we then attempt to extract this seasonal dependence from an observational dataset in spite of several challenges described in the manuscript.*

*There is motivation to discuss seasonality in the SOR on monthly timescales for a number of reasons:*

- *There may be intraseasonal variations in the SOR that would be smoothed out by taking a seasonal mean: e.g. Hood et al. (2015) emphasise the importance of solar-induced ozone anomalies at high latitudes in early winter.*
- *Climate model ozone datasets are typically produced at monthly resolution.*

*Upon performing the analysis, we find regions where the magnitude of the SOR on monthly timescales is considerably different from the annual mean. We then provide a discussion of the implications of these findings that concludes this is an under researched area and new studies are required to establish whether or not such seasonal fluctuations in the SOR are important for models.*

Once a properly formulated MLR is applied to extract the solar cycle signal in each of the two basic (SAGE-based and SBUV-based) datasets, the extracted solar signals then provide baseline solar cycle representations, with associated statistical uncertainties. Post processing of these two solar representation could include (uncertainty weighted)

averaging and/or merging and error propagation to achieve a coherent synthesis and final product that the present manuscript lacks.

*One of our main conclusions is that the SAGE data are most reliable for assessing the solar-ozone response in number density units. We have therefore not averaged the recommended SAGE and SBUV datasets to produce a final merged product because SBUV is provided as mixing ratios. Instead we suggest that chemistry-climate models be compared to both datasets in each of their native coordinates.*

**Derived Product –Quantitative Assessment, Uncertainties, Comparison with Similar Products.**

The authors make numerous comparisons and describe differences among multiple ozone (and temperature) datasets but they do not synthesize their results or reach quantitative conclusions about the solar-ozone representation for input to climate model simulations. Having determined a best possible solar cycle representation of ozone and corresponding uncertainties, they could assess how this compares with independent representations and whether on not the uncertainties make the derived solar signal representation useful for input to the model calculations. The answer to this will depend on the magnitude of the uncertainties in conjunction with the requirements of the models (as noted above). Even if the uncertainty in the solar cycle signal is 50%, this may still be useful if, say, the models differ by 100% among themselves.

Independent validation is possible by comparison with Tier1.4 or other solar cycle representation of ozone published previously. This is different from the analysis that the authors have made to extract the solar signal, using their own regression model, from the Tier 0 database. Rather, comparing their solar cycle representation of ozone with that of the Bodeker et al. and others takes into account the different regression model formulation as well as the different datasets. Additionally, since it is likely that climate models will use the Bodeker et al. ozone profile dataset (it was developed expressly for that purpose) it would be good to compare the solar cycle representation in that dataset with the one developed from this work.

*The solar-ozone response in the Bodeker et al dataset will be strongly affected by the version of SAGE II that it includes (i.e. SAGE II v6.2). Therefore the comparison of SAGE II v6.2 and v7.0 for number density and mixing ratios is extremely relevant for understanding the solar-ozone response in ozone datasets that are expressly produced for models such as Bodeker et al. and Cionni et al.*

*Since the scope of the manuscript has been revised, and the analysis is no longer orientated around CMIP6, we will instead include analysis of the solar-ozone response in ozone datasets for models (Cionni et al; Bodeker et al) in Part II of the study, which focuses on models. The revised manuscript includes more information about the statistical uncertainties in the solar regression coefficients extracted from the MLR which will enable the observed uncertainties to be more readily compared to uncertainties amongst models in Part II.*

**Technical Comments:**

The text of this manuscript is not as precise as it might be and the authors should provide additional clarity when revising the manuscript. It can be quite difficult to discern the actual analyses being described. The discussion of Figure 3 is an example of this. The text states that "Figures 3(a) and (b) show latitude-altitude plots of the percentage differences in ozone number density between solar maximum and minimum for SAGE II v6.2 and v7.0, respectively". What is not at all clear is how these percentage differences were obtained. On first reading they seemed to be differences of the direct time series themselves (shown in Figure 2) during solar maximum and minimum conditions. If so, then because of the other influences on ozone, such direct differences are not reliable indicators of the solar cycle signal. Are they, instead, derived using the MLR? The text doesn't say.

Another example of this general lack of precision of the text is frequent generic reference to "results" (as noted above). An example is on page 16: "to test this, we add a linear trend term into the MLR; however, this does not strongly affect the results as compared to Figure 11 (not shown)." By results they presumably mean the coefficients of the solar predictor in the statistical model – so this is what they should say in the text. More generally, the paper presents many "results" about many things and each one should be properly articulated. Even if this lengthens the manuscript, it makes the message far more clear for the reader.

*The acronym solar-ozone response (SOR) has been introduced throughout the text to clarify various references to "results". We have also made textual changes throughout the manuscript to improve the overall clarity and precision.*

Line 20 ff: the authors refer throughout to the SBUV VN8.0 and 8.6 datasets – do they mean SBUV MOD8.0 and 8.6 – where MOD is Merged Ozone Data?

*Changed to SBUVMOD VN8.0 throughout.*

Line 29-30: The authors make the mistake of confusing the absolute energy changes in the total and UV spectrum with their relative changes and suggesting that TSI changes are somehow much less than UV changes. The change in TSI from solar max to min is larger in absolute energy units than is the change in the UV spectrum. The manuscript needs to clarify this.

*Changed to: "Whilst fractional changes in total solar irradiance (TSI) between the maximum and minimum phases of the approximately 11 year solar cycle are known to be small (<0.1%), there is enhanced fractional variability in the ultraviolet (UV) spectral region (>6%) (e.g. Ermolli et al. (2013))."*

Lines 454-477: These differences have been reported and interpreted previously in the corresponding total ozone datasets of the two versions. Analysis of the solar (and other signals) in the Ozone MOD V8.0 and MOD V8.6 total ozone datasets (Lean, JAS, 2014) shows the regional and global differences for the solar cycle signal in the two different datasets, finding a smaller solar cycle amplitude in the MOD V8.6 of 3DU versus 5DU on Mod V8. That paper discussed these differences, including the calibration issues.

*We have added a citation to Lean (2014) and discuss their mains findings.*

Page 20, lines 670-673. The authors should remove their "encouragement" of instrument teams to better analyze their data! The instruments teams are already undertaking a very challenging and difficult task of space-based metrology and are, without doubt, (more than) fully aware of the need for properly specifying instrumental effects in their datasets as well as they can!

*Removed.*

A more helpful conclusion would be for the authors to generate an actual quantitative product – namely the solar representation of the solar cycle in ozone that is the stated goal of their paper - and assess the future needs of ozone observations in the context of the associated uncertainties of that product.

*As noted above, the goals of the study have been reframed in the revised manuscript to focus on comparisons of the solar-ozone response between recently updated and previous versions of the main long-term satellite ozone datasets (SBUV and SAGE II). The conclusions have been amended accordingly.*

**Anonymous Referee #3**

**General Comments**
The paper seeks to explore the response of stratospheric ozone to solar cycle changes as observed in several global ozone data sets through a regression analysis. The observed response is presented, discussed, and contrasted between the various data sets. There appears to be as many differences as there are similarities in the observed responses. This paper is part one of a two-part series, with the second part said to focus more on atmospheric modeling.

While there are major, but correctable, flaws in the analysis of the data sets, the general qualitative conclusions of the paper are likely robust. The quantitative results, however, will need correction for issues listed below. Overall this is an important and timely study that will demonstrate the limitations of the information content in the existing historical stratospheric ozone record, as well as, the dependence of these records on the quality of ancillary data. The title of the paper is appropriate and the abstract is a complete summary of the paper's current content. References included in the paper strike a nice balance in both quality and quantity.

*We thank the reviewer for reading the manuscript and providing their helpful comments and suggestions. We address their specific issues in turn below.*

**Specific Comments**

The primary concern with the analysis methodology used in this paper has to do with the regression model - specifically, the lack of a diurnal term. The sparse spatial and temporal sampling provided by an occultation measurement system presents unique challenges. As has been done for decades now, the data are often reduced into monthly zonal mean time series with each mean treated as though it is representative of both the latitude and month of the center of the monthly zonal bin. While this is not usually too problematic, it is also common practice to assume that both the local sunrise and sunset sampling is unbiased in each mean so that the diurnal variability can be ignored.
While many published papers have already ignored the diurnal sampling issue, and undoubtedly many more will be submitted, it is now time to address this problem and develop a suitable approach for mitigating its impact. In order to demonstrate the presence of the issue in this work, look at figure 2. After the interruption of the SAGE II data record in late 2000 due to an instrument problem, the measurements resumed at a 50% duty cycle - alternating between sunset only and sunrise only periods of approximately 1 month duration each. This is seen in the deseasonalized monthly anomalies plotted in figure 2 as an abrupt increase in variance after 2000. The authors simply call this increased noise. Closer inspection of the figure shows that this "noise" in the equatorial-zone anomaly increases markedly with altitude. This variance is not simply noise, but rather the direct effect of a biased sampling of the known diurnal variability in stratospheric ozone. More subtle variances in diurnal sampling occur throughout the record and other latitudes all of which contributes to the apparent "noise". It is not immediately obvious how this additional unmodeled variance affects the regression results, but it is likely that it will correlate with some of the regression terms and produce biased results. Many, but not all (SAGE-GOMOS being an exception),

of the extended time series that add other data sets on to the SAGE II record have ignored the diurnal issue in the normalization process and are therefore highly suspect. More on these extended data sets momentarily. Computing monthly zonal means is not difficult and it should be possible to create time series where the local sunrise and local sunset measurements are kept separate. The regression would have to include a new "diurnal comb" term, but the number of data points in the regression would nearly double. The AR analysis would have to make sensible assumptions before application. Alternatively, it may be possible to keep the existing time series and add a new term that accounts for the relative sunrise to sunset proportions of events contributing to the mean. While also adding a term to the regression, this would reduce the degrees of freedom since no new data points are added to the regression. In this case, the AR analysis would probably apply as it is currently done. Either approach may require additional terms to concurrently deseasonalize the time series, since the current approach of subtracting the mean monthly mean would no longer work. Exactly which results from this Part I are being carried forward into the Part II paper or will emerge as relevant in the combined whole is not readily apparent. If, however, the quantitative results presented here are important, this regression analysis will need to be completely redone.

*At the suggestion of the reviewer, we have added in a new term to the MLR model for the SAGE II datasets that quantifies the fraction of sunrise to total (sunrise + sunset) retrievals that are used to produce each monthly and zonally averaged profile. An example of this term for SAGE II v7.0 at 1hPa over the tropics is shown in Figure 2 of the revised manuscript. As the reviewer highlights, this index shows strong time dependency that is likely linked to the increasing variance of the SAGE II ozone anomalies with altitude in Figure 3. This variance should project strongly onto the SR/(SR+SS) term. The results for the SAGE II solar-ozone response in Figures 4 and 5 of the revised manuscript now include the effects of the SR/(SR+SS) term in the MLR. The inclusion of this term does not fundamentally change the diagnosed solar-ozone response, but it does slightly reduce the peak magnitude in the tropical upper stratosphere.*

*Because the extended SAGE II datasets take varying approaches for dealing with diurnal sampling, as pointed out by the reviewer, we have not included a similar diurnal sampling term in the MLR model applied to those datasets. However, the results for the SAGE II data suggest that this should not have a large effect on the estimation of the solar-ozone response in those SI$^2$N datasets.*

Another concern, which is already discussed to some extent in the paper, has to do with the impact of the relative drift between data sets in time series comprised of data from more than one measurement system. It is reasonably clear that, given the timing of the end of the SAGE II data set, any such relative drift will bias primarily into the EESC term. The solar cycle term will also be biased, however, as there is some correlation between the EESC and F10.7 terms. What is not discussed is that the amplitude assigned to the volcanic term will change in response to varying degrees of drift between measurement systems via its correlation with the F10.7 term. This would seem to be an important diagnostic (especially in the upper stratosphere where the expected volcanic influence may, arguably, be small), but in reality it only serves to reinforce the conclusion that the currently available time series are too short to provide sufficient orthogonality between terms with predominantly low frequency content. It would seem prudent to add a figure

showing the lat-alt distribution of the amplitude of the volcanic term. Attribution of the actual response of ozone to solar cycle variations solely to the amplitude of the F10.7 term is highly problematic with these extended data sets. Drift corrected composite time series, to the extent that they can be created, would seem to be required. The associated correlations and resultant coupled uncertainties should be more thoroughly discussed. On a positive note, the analysis done to attribute the SAGE II v6.2-v7 differences to either algorithm changes or Met data source selection are enlightening.

*We agree with the reviewer that for relatively short time periods it is difficult to separate possible effects of multiple external drivers that may be partly correlated with each other. In the revised manuscript we analyse all datasets over their entire lengths, rather than focusing on a shorter common analysis period. Although this by no means removes all the issues highlighted by the reviewer, it does help with increasing the degrees of freedom available to separate individual drivers.*

*As a result of comments from the reviewers, we treat volcanic effects differently in the revised manuscript. Rather than including a volcanic term in the MLR, we instead exclude data from the analysis in the 2 year periods following the El Chichon and Mt Pinatubo eruptions. We hope that this addresses the reviewer's concern about aliasing between the solar cycle and volcanic signals.*

*With respect to drifts between individual datasets, this is most relevant for the extended SAGE II datasets and the SBUV records. We now show timeseries of ozone anomalies for the extended SAGE II datasets, which makes it possible to see behaviours of the combined records. For example, we now point out that OSIRIS shows persistent positive tropical ozone anomalies during the solar cycle 23 minimum and this might contribute to the reduced magnitude of the solar-ozone response in the SAGE-OSIRIS dataset compared to the SAGE II data.*

*For SBUV records, the process of data selection, calibration and merging is important and this is stated in the manuscript. We show how this can affect the solar-ozone response by comparing two SBUV VN8.6 datasets and describe how the differences in methodologies of these datasets affect the diagnosed solar signal.*

**A few less critical Specific Comments**

Page 7 lines 224-225: Would it be possible to illustrate the effect of the AR2 vs AR1?
*As a result of the reviewer comments, the Methodology section has now been significantly expanded and we feel that including a detailed comparison of the use of an AR1/AR2 model for the residuals would make this section even more dense when the effect on the results is minor. We have therefore not included this in the revised manuscript.*

In several places, the authors "blame" NMC/NCEP for the poor quality of the SAGE II mixing ratio conversion when, in reality, the method by which the SAGE II team extends the NMC/NCEP profile to high altitude may be the culprit. The details of this process are discussed in a paper this work already references.
*Our intention was not to "blame" either the NMC/NCEP or MERRA temperature datasets, but rather to point out that there are uncertainties in the evolution of stratospheric temperatures over the recent past (both in reanalyses and satellite observations). We have*

*amended the text to read: "The NMC/NCEP data show exceptional behaviour between 2000-03. At 1hPa, there is a warming of more than 3K over this short period, which is coincident with a warming of ~1K at 2hPa. In contrast, at 5 and 10hPa there is a cooling of more than 4 and 2K, respectively, over this period. The magnitude and vertical structure of these changes in the NMC/NCEP record seems inexplicable as to be related to any physical process, particularly when compared to the variations found in the remainder of the record. Some of these issues may be related to the method used to construct the NMC/NCEP temperature record itself. NCEP reanalysis data were only available for pressures greater than 10hPa, requiring the addition of operational analyses to extend the data to the stratopause. Data from an atmospheric model was used to further extend the temperature data to the mesosphere, but these levels are not considered here (see e.g. Damadeo et al (2013) for more details). The NMC/NCEP temperature record used to convert SAGE II is therefore constructed from several component datasets. Regardless of the exact cause, it seems likely that some of the temperature variations in the NMC/NCEP record are spurious and this may impact on the diagnosed SOR in the SAGE II v6.2 mixing ratio data."*

Lines 526-527: Is the amplitude of the volcanic response invariant as the time sub-period is changed?
*As noted above, to avoid possible aliasing between the solar cycle and volcanic signals the periods following major tropical volcanic eruptions are now excluded from the MLR analysis and we therefore no longer diagnose a volcanic response.*

**Minor Technical Comments**

The URL link on line 111 does not appear to work as expected
*URL link has been updated.*

The time series show in Figure 1 should be extended to match the longest time period to which they are applied.
*Change has been made.*

Why does panel (b) in figure 5 appear to contain much coarser latitudinal structure than seen in panel (a)?
*MERRA data have now been regridded to the same resolution as NMC/NCEP in this Figure.*

Line 500 and elsewhere, this reviewer had some difficulty determining whether the term "increase" referred to trends or the solar cycle response. The authors may wish to introduce and use an acronym (e.g., SCR - Solar Cycle Response) to help clarify the topic under discussion rather than use a generic term such as "increase".
*We have attempted to clarify the language to avoid the use of general statements such as "increase" and have introduced an acronym for the Solar-Ozone Response (SOR), as suggested by the reviewer.*

Line 548: "improve" should be "improved".
*Change has been made.*

**Anonymous Referee #4**

**Summary**

To correctly investigate the impacts of solar variability on the climate using models, it is imperative that models correctly simulate the stratospheric ozone response to the solar cycle that may lead to an impact upon surface climate. The ozone response and feedback may be as important as the direct heating effect from solar variability. Thus, for models that do not calculate ozone variability online, a realistic representation is vital to correctly simulate this solar pathway to impact the climate. This requires using either modelled ozone responses, or those taken from multiple satellite sources. The difficulty is that different satellite data can tell different stories of how the stratospheric ozone has varied over time. Thus quantifying the true behavior, and extracting the solar component as an input into models is difficult. This paper, as the first of two parts, aims to investigate the behavior of several (SI$^2$N) merged datasets that are relatively new and many of which are either based upon, or have a large component from, the long SAGE-II record. The authors present a comprehensive comparison of the behavior of the extracted solar signal in all seven SI$^2$N ozone datasets. They also seek to quantify and understand why two versions of the same data in units of volume mixing ratio, v6.2 and v7.0, differ so much. Identifying the source of the difference due to the temperature data used to convert from number density, the authors apply their own conversion to investigate the differences in the two versions of SAGE and then expand their investigation and discussion to the other datasets based on SAGE. While this paper does not lead to a better understanding of why the solar signal extracted from the SI2N data differ so much with each other (except versions of SAGE II), or hint which one is likely the best to use in future studies, this is an important contribution to the field. The knowledge of how and where datasets differ will provide a step towards, not only, understanding the datasets, but potentially improving them in future work. The work done to understand why SAGE II v6.2 and 7.0 differ was an interesting, revealing and useful analysis. The results are generally clear and well communicated. In context of the two-part study this analysis aims to, and presumably will, inform in the production of an input ozone data set for the CMIP6 modelling runs. From the view of this reviewer, following a point that needs addressing, and some clarifications, the paper fulfills its aims and will be ready for publication.

*We thank the reviewer for reading the manuscript and providing their helpful comments and suggestions. We address their specific issues in turn below.*

**Specific comments**

Page 25, lines 7-10: The point is made that for each of the sub-periods considered in the MLR in Figures 11a-f, both El Chichon and Mt Pinatubo are included. With both eruptions included, and yet the upper stratosphere spatial pattern changing with each sub-period, the authors suggest that this implies differences are unlikely to be volcanic. As stated by the authors, the eruption of El Chichon occurred in April 1982, so the effects of the eruption on the stratosphere would be expected to have gone by early 1985, and be less pronounced in 1984 than 1983 and 1982. However, while the authors are correct that both eruptions are included in 11a-b, and any effects likely present in

11c, El Chichon is not included in panels 11d-f, which begin in 1985, almost three years after the eruption. While temperature responses to volcanic eruptions are stronger in the lower stratosphere, and less, if any in the mid-stratosphere, there are hints that the mesosphere sees a response to volcanic eruptions (e.g. Beig et al., 2003), so there may indeed be an aliasing with volcanoes that decreases (as seen in 11c-f relative to 11a-b) when El Chichon's effect is removed by the sub-period chosen. Further to this, there is a very large anomaly present in both SBUVMOD and SBUVN8.0 around the time of the 1982 eruption (Fig 8) that lasts for 1-2 years; a similar event is not present following the Pinatubo eruption, and so is likely not of volcanic origin (unless it is related to a change in the atmospheric viewing of the instrument for a reason unique to El Chichon). Such a large anomaly may have an influence on the MLR leading to the change in the spatial patterns plotted in Figure 11, and then the authors may indeed be correct that it is not volcanic in origin. Perhaps it would be worth applying SBUV-Merged Cohesive to test this, as the anomaly is not visible in that time series in Fig. 8.

*Figure 11 has now been removed from the revised manuscript as a result of comments from another referee. Furthermore, the periods following the two major volcanic eruptions (El Chichon and Mt Pinatbuo) are now excluded from the MLR analysis to reduce the potential for aliasing between the solar cycle and volcanic signals.*

**The following are suggestions for the authors to clarify or reword:**

Page 13; lines 20-26: Indeed, there also appears to be a larger positive anomaly in 1992-1994, a period of maximum and high activity, so it is possible this may also contribute to the enhanced signal seen in Fig. 3c.
*As noted above, the period June 1991-May 1993 has now been excluded from the analysis to remove the effects of the Mt Pinatubo eruption. This change has little impact on the results in Figure 3(c) (now Figure 4(c)).*

Page 15; line 13: While records indeed show there was a warming of ~0.5-1.0 K following the Pinatubo eruption, and perhaps there is a small increase in 1991/1992 in Fig. 4, it appears that at 30 hPa a warming began in 1989, followed by the ~2 K decrease after 1992. The eruption does not appear to stand out in this time series, so perhaps the authors may wish to revise the focus of their comment here.
*We have amended the text to read: "At 30hPa, the evolution of the two temperature records is nearly identical during the period of overlap, with a long-term cooling trend of ~0.6 K decade$^{-1}$ that is strongly connected to an apparent step-wise cooling of ~2K between 1992 and 1994."*

Page 15, lines 18-19: Two points of clarity here. It would better to reformulate to discuss NCEP first, as the MERRA data do not show the decline in the last three years, but in the last three years of NCEP. Note also that the solar cycle decline began in mid to late 2002, so this three year period is mainly during the maximum period. This of course does not change the point being made, and it is well worth highlighting also that this odd behavior in NCEP also occurs (though inversely) in this same, three-year period, at 5 and 10 hPa, hinting at an issue with NCEP.
*We have reformulated and expanded these sentences as suggested by the reviewer (see revised manuscript).*

Page 20, lines 19-22: The authors state that many of the SAGE-II based datasets have differences that are likely the result of merging procedures. Do the authors include in this comment also that SAGE-GOMOS 1&2 and SAGE-OSIRIS have less data (or more data gaps) in the equatorial region than SWOOSH and GOZCARDS? Or if not, might this additionally decrease the significance of the signal in the tropics and lead to the less 'smooth' appearance of the spatial patterns? The reviewer is also aware that Aura MLS v2.2 used in GOZCARDS, and v3.3 in SWOOSH have different short-term variability (larger in GOZCARDS). This might be worth checking/considering.

*In the revised manuscript, only 3 extended SAGE II datasets are analysed which provide ozone number densities rather than mixing ratios (SAGE-GOMOS 1/2 and SAGE-OSIRIS). Therefore SWOOSH and GOZCARDS no longer feature in the revised manuscript.*

*We now include figures in the Supplementary Information that show the number of data points used to diagnose the solar-ozone response as a function of latitude and height. This enables a comparison of the sampling by the original SAGE II data and the extended SAGE II datasets with the SBUV records. These differences in sampling between the datasets are briefly referred to in the main text.*

Page 22, line 29: The two SBUV records have almost the same datasets used, except the Merged Cohesive uses a little over a year from NOAA9 (see Tummon et al., 2015, Fig 1.).
*We have amended the text to read: "The two SBUV VN8.6 datasets contain some differences in the data that is included from different instruments within a particular period (see Figure 1 in Tummon et al (2015)), and in the methods for averaging and merging these data. SBUV Merged Cohesive VN8.6 uses data from a single instrument in any time period; the individual records are then bias-corrected to produce a continuous record (Wild and Long, 2015). In contrast, SBUVMOD VN8.6 is constructed by averaging all available data within a particular time window (Frith et al., 2014). The SBUVMOD datasets extend back to 1970 by including data from the BUV instrument on Nimbus 4 from 1970-1976, whereas the SBUV Merged Cohesive dataset starts from 1978 with the first SBUV instrument on Nimbus 7."*

Figure 6 might benefit with a third column of difference plots, since the differences are well discussed, though specific altitudes and latitudes are usually not mentioned. Thus the difference plots might make it easier for the reader to locate what the authors are referring to. This is at the authors discretion. However, for the point made about the solar signal in Figs 6g and 6h, relative to 6a and 6b, that the signals are larger in NCEP than MERRA, while in absolute values this is the case, I wonder how significantly different, statistically these signals are? Figures 6g and h seem more similar than 6a and b do in the upper stratosphere; this may be helped with a difference plot with significance, as mentioned.
*A third column showing differences has been included in Figure 6 (now Figure 8 in revised manuscript).*

**Technical corrections**
Page 19, line 4: "but the magnitudes [are] quite similar"
*Change has been made.*

Manuscript prepared for Atmos. Chem. Phys.
with version 2015/04/24 7.83 Copernicus papers of the LaTeX class copernicus.cls.
Date: 10 June 2016

**The representation of solar cycle signals in stratospheric ozone. Part I: A comparison of satellite observations**

Amanda Maycock[1,2,3], Katja Matthes[4,5], Susann Tegtmeier[4], Rémi Thiéblemont[4], and Lon Hood[6]

[1]Centre for Atmospheric Science, University of Cambridge, Cambridge, UK.
[2]National Centre for Atmospheric Science, UK.
[3]now at School of Earth and Environment, University of Leeds, Leeds, UK.
[4]GEOMAR Helmholtz for Ocean Research, Kiel, Germany.
[5]Christian-Albrechts Universität zu Kiel, Kiel, Germany.
[6]Lunar and Planetary Laboratory, University of Arizona, Tucson, Arizona, USA.

*Correspondence to:* Amanda C. Maycock (a.c.maycock@leeds.ac.uk)

**Abstract.**  Changes in incoming solar ultraviolet ~~irradiance on stratospheric ozone forms an important part of the climate response to solar variability. To realistically simulate the climate response to solar variability using climate models, a minimum requirement is that they should include a solar cycle ozone component that has a realistic amplitude and structure, and which varies with season. For climate models that do not include interactive ozone chemistry, this component must be derived from observations and/or chemistry-climate model simulations and included in an externally prescribed ozone database that also includes the effects of all major external forcingstototwo part study presents the solar-ozone responsesfor the period 1984-2004, including the Stratospheric Aerosol and Gas Experiment (SAGE) II version 6.2 and version 7.0 data, and the Solar Backscatter Ultraviolet Instrument (SBUV) version 8.0 and version 8.6 data. A number of combined datasets, which have extended SAGE II using more recent satellite measurements, are also analysed for the period 1984-2009. It is shown that SAGE II derived solar-ozone signals are sensitive to the independent temperature measurementsatoozone number~~

density to mixing ratio units. A change in these temperature measurements in the recent SAGE II ozone number densities to mixing ratios. Since these temperature records contain substantial uncertainties, we suggest that datasets based on SAGE II number densities are currently most reliable for evaluating the SOR. We further analyse three extended ozone datasets that combine SAGE II v7.0 data leads to substantial differences in the mixing ratio solar-ozone response number density data with more recent GOMOS or OSIRIS measurements. The extended SAGE-OSIRIS dataset (1984-2013) shows a smaller and less statistically significant SOR across much of the tropical upper stratosphere compared to the previous v6.2, particularly in the tropical upper stratosphere SAGE II data alone. In contrast, the two SAGE-GOMOS datasets (1984-2011) show SORs that compare better with the original SAGE II data and therefore appear to provide a more reliable estimate of the SOR. We also show that alternate satellite ozone datasets have issues (e. g. , analyse the SOR in recent SBUVMOD version 8.6 (VN8.6) (1970-2012) and SBUV Merged Cohesive VN8.6 (1978-2012) datasets and compare them to the previous SBUVMOB VN8.0 (1970-2009). Over their full lengths, the three records generally agree in terms of the broad magnitude and structure of the annual mean SOR. The main difference is that SBUVMOD VN8.6 shows a smaller and less significant SOR in the tropical upper stratosphere, and therefore more closely resembles the SAGE II v7.0 mixing ratio data than does the SBUV Merged Cohesive VN8.6, which has a more continuous SOR of ~2% in this region. The sparse spatial and temporal sampling , low vertical resolution, and shortness of measurement record), and that the methods of accounting for instrument offsets and drifts in merged satellite datasets can have a substantial impact on the solar cycle signal in ozone. For example, the magnitude of the solar-ozone response varies by around a factor of two across different versions of the SBUV of limb satellite measurements prohibits the extraction of sub-annual variations in the SOR from SAGE-based datasets. However, the SBUVMOD VN8.6 record, which appears to be due to the methods used to combine the separate SBUV timeseries. These factors make it difficult to extract more than an annual-mean solar-ozone response from the available satellite observations. It is therefore unlikely that satellite ozone measurements alone can be applied to estimate the necessary solar cycle ozone component of the prescribed ozone database for future coupled model intercomparison projects (e.

g., CMIP6). dataset suggests substantial month-to-month variations in the SOR, particularly in the winter extratropics, which may be important for the proposed high latitude dynamical response to solar variability. Overall, the results highlight substantial uncertainties in the magnitude and structure of the observed SOR from different satellite records. The implications of these uncertainties for understanding and modelling the effects of solar forcing on climate should be explored.

**1 Introduction**

 Whilst fractional changes in total solar irradiance (TSI) between the maximum and minimum phases of the approximately 11 year solar cycle are known to be small (<0.1%), there is enhanced fractional variability in the ultraviolet (UV) spectral region (>6%) (e.g. Ermolli et al. (2013)). An in-
crease in UV irradiance impacts stratospheric heating rates, and thus temperatures, through two main mechanisms: (1) enhanced absorption of radiation by ozone, and (2) enhanced production of ozone through the photolysis of oxygen at wavelengths less than ~242 nm. Consistent with these mechanisms, past studies using observations, reanalysis data and  models have identified an increase in annual mean temperature in the upper stratosphere of up to ~1.5 K between solar maximum and minimum (e.g.  Ramaswamy et al. (2001); Mit and an increase in ozone abundances of a few percent  (Soukharev and Hood, 2006; Haigh, 1994) . These radiatively driven changes modify the meridional temperature gradients in the upper stratosphere, which can lead to a modulation of planetary wave propagation and breaking, and changes in the strength of the stratospheric polar vortex (e.g. Kuroda and Kodera (2002); Matthes et al. (2004, 2006); Gray et al. (2010); Ineson et al. (2011)). Such feedback mechanisms can lead to amplified changes in regional surface climate via stratosphere-troposphere dynamical coupling (e.g. Gray et al. (2010)). Constraining the stratospheric  response to solar forcing is therefore important for understanding solar-climate coupling and potential sources of decadal variability in the climate system (e.g. Thiéblemont et al. (2015)).
 The solar-ozone response (SOR) has been estimated to make a substantial contribution to variations in stratospheric temperatures over the 11-year solar cycle. Gray et al. (2009) used an estimate of the SOR from SAGE II (Stratospheric Aerosol and Gas Experiment II) version 6.2 (v6.2) satellite ozone mixing ratio data and spectral solar irradiance (SSI) variations from Lean (2000) to show that the contribution of the SOR to temperature changes between the maximum and minimum phases of the 11-year solar cycle is around 60% at the  tropical stratopause~~(~1,hPa) ranges from 0.3-1.2and 70-80% between 20-30 K across stratosphere resolving 'high-top' models (Mitchell et al., 2015b)majority of these models used the same spectral solar irradiance dataset (Wang et al., 2005) , the reasons for this spread are likely to include differences in the models' shortwave radiation schemes (Nissen et al., 2007; Forster et al., 2011) , and/or whether they prescribe or interactively simulate stratospheric ozone (Hood et al., 2015) . For models which include interactive chemistry, the details of their photolysis schemes will also be important for simulating the impact of solar variability on ozone. The CMIP5 models also show~~

considerable differences in their high latitude dynamical responses to solar variability (Mitchell et al., 2015b) ,

95 two studies used similar SSI data, this difference must arise from the SOR estimated from SAGE II observations used by Gray et al. (2009) being different from that simulated in the atmospheric chemistry models used by Shibata and Kodera (2005) . It is therefore important to evaluate the SOR and its uncertainties in different observational datasets to understand the climate

100 response to solar variability and to provide an independent means for evaluating the performance of atmospheric chemistry models (e.g. Austin et al. (2008) ; see also Part II).

Whilst past studies have  quantified the SOR in observations (e.g.  Soukharev and Hood (2 there are differences in the magnitudes and structures between individual satellite records. It is not

105 clear whether these are due to inter-instrument differences in observational periods and/or differences in instrument resolution, sampling or drifts. There are also apparent differences in the structure and magnitude of the  SOR between observations and atmospheric  chemistry models (e.g. Haigh (1994); Soukharev and Hood (2006); Austin et al. (2008); Dhomse et al. (2011)). These issues are compounded by current uncertainties in the

110 characteristics of spectral solar irradiance variability (e.g. Ermolli et al. (2013)), which have implications for constraining the magnitude and structure of the  SOR because of its dependence on photochemical processes  (Haigh et al., 2010; Dhomse et al., 2015; Ball et al., These factors present  an additional challenge for understanding and evaluating the overall climate response to solar variability, particularly since dynamical feedbacks may amplify the effects of an

115 initially small forcing (e.g. Matthes et al. (2006)).

The  aim of this two part study (see also Maycock et al., in prep.) is  to evaluate the representation of the  SOR and its uncertainties in satellite observations and global models. The

120   present Part I

125  describes the SOR in the latest version 7.0   (v7.0) of the SAGE II dataset and compares it to the former v6.2, which has been used in several solar-climate studies (e.g. Soukharev and Hood (2006); Gray et al. (2009)) and in several ozone databases developed for climate models without interactive chemistry (Cionni et al., 2011; Bodeker et al., 2013) . A number

130 of merged satellite ozone datasets, which extend SAGE II using more recent measurements,

have also been created and  analysed as part of the  WCRP/SPARC (World Climate research Programme/Stratospheric-tropospheric Processes and their Role in Climate) SI$^2$N ozone trends activity (e.g. Tummon et al. (2015));  we analyse the  SOR in three of these combined satellite ozone datasets. We also  analyse the SOR in two versions of the recently released VN8.6 of the Solar Backscatter Ultraviolet Instrument (SBUV) data and compare  these to the former SBUVMOD VN8.0 data.  Part II of  the study (Maycock et al., in prep.) describes the  SOR in atmospheric chemistry-climate model simulations from the WCRP/SPARC Chemistry-Climate Model  Initiative (CCMI) and compares them to  a subset of the observational records discussed here that are determined to be most reliable for diagnosing the SOR (see below). Part II also discusses the representation of the  SOR  the climate model ozone dataset created for the fifth Coupled Model Intercomparison Project (CMIP5  145 ) (Cionni et al., 2011) . This leads to a discussion of the representation of the SOR in the ozone dataset being created for CMIP6  model simulations (Hegglin et al., in prep.).

Given the potential application of the results described below for use in climate model simulations, it is prudent to briefly review the typical requirements of an ozone database for models by describing the CMIP5 dataset as a representative example (Cionni et al., 2011) (see also Bodeker et al. (2013) ). The CMIP5 ozone database provided monthly mean ozone mixing ratios on a regular latitude/pressure grid at a horizontal resolution of $5° \times 5°$ (lon/lat) on 24 pressure levels covering 1000-1 hPa for the period 1850-2100. Data were provided on the following pressure levels: 1000, 850, 700, 600, 500, 400, 300, 250, 200, 150, 100, 80, 70, 50, 30, 20, 15, 10, 7, 5, 3, 2, 1.5, 1 hPa. Stratospheric ozone data (at p$\leq$300 hPa) were given as zonal mean values. Therefore for any description of the SOR must fulfil these (or similar) criteria to be viable for use in climate models (i.e. global coverage at monthly mean resolution and with sufficient vertical and horizontal resolution throughout the stratosphere).

**2  Ozone datasets**

**2.1**

The satellite ozone datasets examined in this study are summarised in Table 1.  A detailed overview of  their spatial and temporal sampling characteristics and, where appropriate, their merging procedures is provided by  Tummon et al. (2015) and

references therein. Their main properties are briefly summarised below. Since our goal is to extract a signal with power on a quasi-decadal timescale, it is desirable to use the longest available timeseries and we therefore analyse all datasets for their full time periods. For the longest record considered, this amounts to approximately three solar cycles.

**2.0.1**

**2.1 SAGE II based records**

The SAGE II record forms the basis of many long-term ozone datasets (see e.g. Tummon et al. (2015)). As a limb-viewing instrument, the spatial and temporal sampling of SAGE is fairly sparse, with a given latitude measured approximately once per month; however, it is recognised as having good long-term stability and a vertical resolution of $\sim$1 km in the stratosphere, which are  characteristics that are likely to be important for analysing the SOR. We use zonal and monthly mean ozone data from October 1984 to August 2005 provided through the WCRP/SPARC Data Inititive (SDI) (Tegtmeier et al., 2013).

 native retrieval coordinate of SAGE II is  units of ozone number densities on altitude levels;  data are post-processed to volume mixing ratios (vmr) on pressure surfaces

 levels using temperatures from a meteorological reanalysis dataset. The SAGE II retrieval algorithm was recently updated as part of the version 7.0 release (Damadeo et al., 2013) . The SOR in SAGE II  v6.2 data has been discussed in a number of studies: e.g. Randel and Wu (2007); Soukharev and Hood (2006); Gray et al. (2009) for mixing ratios, and Remsberg and Lingenfelser (2010) for number densities.  Here we compare the SOR in the latest v7.0 release to the pre- v6.2 in units of number densities and mixing ratios. It is important to conduct this comparison for both sets of units because the temperature  record used to convert SAGE II  to mixing ratios  was changed between v6.2 and v7.0 from National Meteorological Center/National Center for Environ-tal Prediction (NMC/NCEP) to Modern Era-Retrospective Analysis for Research and Applications version 1 (MERRA-1) reanalysis data. The impact of this change on the SOR has not been previously evaluated and is described in Section 4.1.

As a solar occultation instrument, SAGE II profiles can be categorised as a sunrise (SR) or sunset (SS) measurement. There are known variations in the relative numbers of SR/SS retrievals over the SAGE II record. For example, SAGE II obtained profiles in two narrow latitude bands each day,

15 each at sunrise and sunset, but after November 2000 SAGE II measured only one profile per orbit at either SR or SS. These variations in SR/SS sampling have been shown to affect estimates of climatological ozone values due to diurnal cycle effects (Toohey et al., 2013), but could also affect temporal variability in monthly mean ozone values. To account for the possible effects of 205 these sampling issues on the estimation of the SOR, we add an additional term to the multiple linear regression model for SAGE II data that represents the fraction of SR to total (SR+SS) profiles used to generate each monthly mean data point (see Section 3).

The SAGE II mission stopped measuring in 2005. Since then several  satellite instruments have continued to measure ozone, and there are now a number of combined datasets that have ex-210 tended  SAGE II to near the present day. These datasets were recently analysed as part  the WCRP/SPARC SI$^2$N  activity to evaluate long-term

 ozone trends (see Tummon et al. (2015) and references therein), including SWOOSH (Stratospheric Water and OzOne Satellite Homogenized) (Davis et al., 2016), GOZCARDS (Froidevaux et al., 2015), SAGE-GOMOS (Global Ozone Monitoring by Occultation of Stars) (Kyrölä et al., 2015; Penckwitt et al., 2015), 220 and SAGE-OSIRIS (Optical Spectrograph and Infrared Imager System) (Bourassa et al., 2014). As mentioned above, SAGE II mixing ratios are produced by conversion of number densities using an independent temperature record. The uncertainties in the SOR that result from using different stratospheric temperature records for this conversion are demonstrated in Section 4.1. This leads us to focus our analysis of  SOR on the extended records that provide ozone as number densities and are therefore less dependent on the conversion issues that accompany the choice of a particular temperature record (see Section 4.2). Since SWOOSH and GOZCARDS currently only provide ozone mixing ratios we do not analyse them here.

230 The three extended ozone datasets all include SAGE II v7.0 number densities. Differences in the SOR between the datasets may therefore arise as a result of the more recent measurements used to extend SAGE  II and/or from the methods used to merge the different satellite records.  Two of the datasets extend SAGE II using GOMOS, which flew on the ENVISAT satellite and covers  2002-2011, but take different approaches for combining the two records. Kyrölä et al. (2015) use GOMOS as a reference and adjust SAGE II sunrise

and sunset profiles separately at each latitude and altitude; this dataset will be referred to as SAGE-GOMOS 1. Conversely, Penckwitt et al. (2015) use SAGE II as a reference and adjust GOMOS data using seasonally-varying offsets at each latitude and altitude; this dataset will be referred to as SAGE-GOMOS 2.

Another dataset is analysed which The third dataset analysed extends SAGE II with OSIRIS (Optical Spectrograph and Infrared Imager System) data covering 2001-present using OSIRIS data and covers 1984-2013 (Bourassa et al., 2014; Sioris et al., 2014). Latitude and altitude dependent offsets are calculated for the deseasonalised data during the instrument overlap period (January 2002-August 2005), and the OSIRIS data are adjusted to produce a consistent combined SAGE II and OSIRIS timeseries.

Two datasets that are comprised of more than two satellite records are also analysed. The SWOOSH (Stratospheric Water and OzOne Satellite Homogenized) record includes SAGE II (v7.0), SAGE III (2002-2005), HALOE (1991-2005), UARS MLS (1991-1999), and Aura MLS (2004 onwards), with Aura MLS used as a reference from which offsets for the other records are calculated (Davis et al., 2016). Finally, GOZCARDS includes data from SAGE I (1979-1982), HALOE, UARS MLS, Aura MLS, and ACE-FTS (2003 onwards), in addition to the SAGE II v6.2 data, which is used as a reference to which the other records are adjusted (Froidevaux et al., 2015). The solar-ozone signals in these five extended records are analysed for the period 1984-2009.

**2.2 SBUV based records**

**2.2.1 SBUV based records**

In addition to SAGE II, the other main long-term internally-calibrated satellite ozone dataset is comprised of data from the Backscatter Ultraviolet Radiometer (BUV) and Solar Backscatter Ultraviolet Radiometer (SBUV) instrument on the Nimbus satellite instruments on board Nimbus satellites and the SBUV/2 instruments on various National Oceanic and Atmospheric Administration (NOAA) satellites. Data are available as mixing ratios on pressure levels from January 1970 to near the present day. As nadir-viewing instruments, SBUV has the BUV/SBUV records have more frequent global coverage than the limb-viewing SAGE II, but its their vertical resolution is at least an order of magnitude poorer at pressures greater than $\sim$15 hPa rendering it more difficult to resolve detailed ozone structures in the mid and lower stratosphere. Since SBUV the entire BUV/SBUV record is comprised of multiple separate records from different satellites, inter-instrument biases and drifts must also be accounted for to produce a homogenised record.

We analyse zonal and monthly mean data from the longstanding SBUV SBUV Merged Ozone Dataset (SBUVMOD) version 8.0 (VN8.0) dataset and the latest release SBUV VN8.6 (McPeters et al., 2013; Bhartia et al., 2013), thereby complementing the analysis of SBUV previous analyses of the SOR (e.g. Soukharev and Hood (2006)). SBUVMOD VN8.0 by Soukharev and Hood (2006).

 covers the period 1970-2009 and was downloaded from http://acd-ext.gsfc.nasa.gov/Data_services/merged/data/sbuv.70-09.za.v8_prof.vmr.rev1.txt.

275 Two versions of the SBUV VN8.6 record have been produced so far: the  SBUVMOD VN8.6 dataset from NASA  which covers 1970-2012 (Frith et al., 2014) , and the SBUV Merged Cohesive dataset from NOAA which covers 1978-2012 (Wild and Long, 2015). These are identical to the datasets analysed as part of the SI$^2$N  activity (e.g. Tummon et al. (2015)). The two SBUV VN8.6 datasets

280 contain some differences in the data that is included from different instruments within a particular period (see Figure 1 in Tummon et al. (2015) ), and in the methods for averaging and merging  these data. SBUV Merged Cohesive VN8.6  uses data from a single instrument in any time period; the individual records are then bias-corrected to produce a continuous record (Wild and Long, 2015).

285 In contrast,  SBUVMOD VN8.6  is constructed by averaging all available data

 within a particular time window (Frith et al., 2014) . The SBUVMOD datasets extend back to 1970 by including data from the BUV instrument on Nimbus 4 from 1970-1976, whereas the SBUV Merged Cohesive dataset starts from 1978 with
290 ~~above, we also analyse the ozone dataset from McLinden et al. (2009) , which uses SAGE I and SAGE II v6.2 data to correct for instrument drifts and inter-instrument biases in the SBUV VN8.0 dataset ; this therefore benefits from the improved long-term stability of the SAGE II data, but retains the improved spatial sampling of SBUV. Note that the SAGE II v6.2 data employed by McLinden et al. (2009) differs from the version described in Section 2.1, which has been used in~~
295 ~~most previous solar cycle studies (e.g. Soukharev and Hood (2006); Randel and Wu (2007); Gray et al. (2009); Cionni et al. (2011) ). McLinden et al. (2009) discuss how the temperature profiles provided in the SAGE data files, which as described above came from the NMC/NCEP (re)analysis, contain apparently spurious long-term trends in the upper stratosphere, as compared to observations. McLinden et al. (2009) therefore convert the SAGE II data to mixing ratios using a temperature climatology with an estimate of the long-term~~
300
first SBUV instrument on Nimbus 7.

**2.2.1**

305 ~~Satellite (UARS) Halogen Occultation Experiment (HALOE) v19 record (Grooß and Russell (2005) ; http://haloe.gats-inc.com/download/index.php). Results are also shown for the Binary DataBase of Profiles (BDBP) Tier 0 dataset (Bodeker et al., 2013) (available from http://www.bodekerscientific.com/data/the-bdbp), which covers 1979-2007 and consists of multiple satellite records, including SAGE I and II (v6.2),~~

HALOE, Polar Ozone and Aerosol Measurement (POAM) II and III and Improved Limb Atmospheric Spectrometer (ILAS) I and II data, as well as ozone sondes and ground-based measurements.

**3   The multiple linear regression model**

Following numerous earlier studies (e.g.  Frame and Gray (2010); Soukharev and Ho the  SOR is diagnosed using multiple linear regression (MLR); this technique enables the signals  associated with different forcings within a single timeseries to be separated.

The ozone data are first deseasonalised by removing the long-term monthly mean at each latitude and pressure (or altitude). As in past studies, we then perform an MLR analysis on the timeseries of monthly mean ozone anomalies at each location, $O_3'(t)$, to diagnose the 11 year solar cycle component:

$$O_3'(t) = A \times F10.7(t) + B \times \underline{EESC}CO_2(t) + C \times \underline{QBO}EESC(t)$$

$$+D \times \underline{QBO_{orthog}}ENSO(t) + E \times \underline{AOD_{volc}}QBO_A(t) + F \times \underline{Nino3.4}QBO_B(t) + r(t), \quad (1)$$

where $r(t)$ is a residual. The analysis mainly focuses on  annual-mean signals, which are calculated by regressing all months as a single timeseries.

The monthly basis functions are: the F10.7cm radio solar flux (http://lasp.colorado.edu/lisird/tss/noaa_radio_flux.html), the $CO_2$ concentration at Mauna Loa (http://www.esrl.noaa.gov/gmd/ccgg/trends/data.html), the equivalent effective stratospheric chlorine (EESC), , the global aerosol optical depth at 550nm ($AOD_{volc}$) updated from  the Nino 3.4 index  calculated from the Extended Reconstructed Sea Surface Temperature (ERSST) v3b dataset (http://www.esrl.noaa.gov/psd/data/gridded/data.noaa.ersst.html), and two quasi biennial oscillation (QBO) indices representing tropical zonal winds at 30 and 50 hPa (http://www.cpc.ncep.noaa.gov/data/indices/). Figure 1 shows example timeseries of these indices from  1970-2015 in arbitrary units. The coefficients A-F are calculated using linear least squares regression.

ENSO is the main regressor for which a lagged response in stratospheric ozone might be expected; however, we find that the SOR is not sensitive to lagging the ozone anomalies with respect to the Nino 3.4 index by 0-12 months. We therefore do not include any lags in Equation 1. We have also tested the sensitivity of the diagnosed SOR to the use of a spatially-varying EESC field using

output from the UM-UKCA chemistry-climate model REF-C1 CCMI integration. However, this has virtually no effect on the SOR compared to the use of a single EESC timeseries for all locations, and therefore adopt the latter approach for simplicity.

We do not include a volcanic term in the regression model, but instead choose to exclude data from the 2 year periods following the two major tropical volcanic eruptions during the analysis epoch: El Chićhon (data excluded from April 1982 - March 1984) and Mt. Pinatubo (data excluded from June 1991 - May 1993). These periods are excluded from the analysis for two reasons: firstly, some of the datasets analysed implicitly exclude data in these periods for quality control purposes, whereas others do not. For consistency, we therefore exclude these periods for all datasets. Secondly, removing these periods reduces the likelihood of aliasing between volcanic and solar signals, which can be an issue within relatively short climate data records (Chiodo et al., 2014) .

We adopt the widely used F10.7cm  solar flux as a proxy for solar activity in the MLR model. This is a more appropriate  measure for variations in the UV spectral region, the key driver of the stratospheric ozone response, than other indices such as  total solar irradiance (Gray et al., 2010) ; however, it should be noted that the F10.7cm flux is not a direct measurement of UV variability, but rather is a proxy for variations at these wavelengths. Throughout the manuscript the SOR is expressed as percent ozone change per 130 solar flux units ($1 \ \mathrm{SFU} = 10^{-22} \ \mathrm{Wm}^{-2}\mathrm{Hz}^{-1}$) to represent the difference between the 11 year solar cycle maximum and minimum.

The 95% confidence intervals on the SORs are estimated by:

$$A \pm t_{\alpha/2, n-(k+1)} \sqrt{C_{AA}}, \tag{2}$$

where $A$ is the solar regression coefficient in Equation 1, $t_{\alpha/2, n-(k+1)}$ is the critical t-value at a confidence level, $\alpha$, of 0.05 with degrees of freedom $n - (k + 1)$ where $n$ is the number of data points in the regression, $k$ is the number of regressors, and $C_{AA}$ is the variance of the estimated solar regression coefficient $A$.

As mentioned in Section 2.1, the SAGE II record is affected by irregular SR and SS sampling as a function of time. This could introduce spurious variability in the monthly mean ozone values, particularly in the upper stratosphere, as a result of the diurnal cycle in ozone. However, many previous regression studies of SAGE II data have not accounted for the non-stationarity in SR/SS sampling (e.g. Randel and Wu (2007) ). Here, we account for this by including an additional term in Equation 1 that quantifies the ratio between the number of SR and the total (SR+SS) number of profiles used to produce each monthly mean SAGE II data point; this index can take values between 0 and 1. An example of this index for the SAGE II v7.0 dataset at 1 hPa averaged over the tropics (30°S-30°N) is shown in Figure 2.

One important issue for MLR analysis  is the handling of possible autocorrelation in the regression residuals and its effects on the estimation of statistical uncertainties. A Durbin-Watson
380  test does not reveal significant autocorrelation in the regression residuals at most locations; however, this is likely to be because there is a considerable fraction of missing data points in many of the datasets analysed. In the analysis of chemistry-climate model simulations in Part II of this study, for which there is implicit complete spatial and temporal sampling, a Durbin-Watson test reveals significant serial correlation in the  regression
385  residuals  in many locations for lags of one and two months, particularly in the lower stratosphere and mesosphere. This autocorrelation can lead to spurious overestimation of the statistical significance of the regression coefficients and we therefore include an autoregressive term in the MLR model. Given the significant serial correlation of the residuals in the chemistry-climate models at up to two months lag in some regions, a second order autoregressive noise process (AR2)
390  is used, which assumes the residuals $r(t)$ have the form:

$$r(t) = ar(t-1) + br(t-2) + w(t), \qquad (3)$$

where $a$ and $b$ are constants and $w(t)$ is a white noise process; this is the same approach employed in the recent SPARC SI$^2$N analysis of ozone trends (Tummon et al., 2015; Harris et al., 2015). The inclusion of
395     this term has a very minor effect on the results for the observational datasets in Part I, but has a greater effect for the model results in Part II. We therefore include it in the analysis here for consistency between both parts of the study.

400  **4  Results**

**4.1  The SOR in SAGE II  datasets**

Figure 3 shows timeseries of monthly and tropical (30°S-30°N) mean percent ozone anomalies from 1984 to 2004 at select  stratospheric levels for SAGE II  v6.2 and v7.0 in units of mixing  ratios (on
405  pressure surfaces) and number  densities (on approximately equivalent altitude surfaces). Data are only plotted where at least 1/2 of the points within the tropical band have values in a given month. The lowest panel shows the monthly  F10.7 cm solar flux for reference.
The  anomalies in the two ozone number density datasets (blue and green lines) are in close agreement  in the mid-stratosphere
410  (24, 31 and 36 km) both in terms of high frequency fluctuations and long-term changes. At 31 km, there are  ozone variations that are consistent with a QBO influence

. At 36 and 40 km, there are variations that are visibly in phase with the solar cycle, with relatively  high ozone values from  1989 to  1992 during solar cycle 22 maximum, and lower ozone values from 1994 to 1998 during the cycle minimum. The data show greater variance in the early and later parts of the records  and fluctuations in phase with the solar  cycle are not evident from the timeseries alone.

The two SAGE II ozone mixing ratio datasets (black and red lines) are also in reasonable agreement for long-term changes in the mid-stratosphere (10 and 30 hPa). However, in the upper stratosphere (1 and 3 hPa) there are substantial differences in both short and long-term variations. For example, SAGE II v6.2 (black line) shows persistent negative anomalies in the early part of the record which are not evident in v7.0 (red line). These coincide with the 11 year solar cycle 21 minimum from 1985 to 1988. Furthermore, in the latter part of the record, v6.2 shows relatively large amplitude fluctuations with  mean positive anomalies from 2002 to 2004, which coincide with the peak and subsequent  declining phase of solar cycle 23. Thus, there are differences in the evolution of ozone between the two SAGE II  mixing ratio datasets, particularly in the upper stratosphere. Overall, the two versions of SAGE II number  densities are in closer agreement than the mixing ratio data.

Figures 4(a) and (b) show latitude-altitude plots of the  SOR for SAGE II v6.2 and v7.0 number densities,  95% confidence intervals for the SORs in Figure 4 expressed as percent ozone anomalies are shown in Figure 5. The SORs in Figures 4(a) and (b) are generally consistent for the two datasets, and show positive values of 2-4% across the tropical and subtropical stratosphere, except for a region of small (<1%) negative values at 30 km in the tropics. There is a relative maximum in the SOR of 3-4% in the tropics 45 km, and two off equatorial peaks of a similar magnitude at ∼40 km and ±35°. These findings  consistent with Remsberg and Lingenfelser (2010) and Remsberg (2014), who found similar 11 year solar-like signals in tropical upper stratospheric ozone number densities in SAGE II v6.2 and v7.0

. The confidence intervals for the SORs in Figures 5(a) and 5(b) show the largest uncertainty at the equator at ∼45 km, which is close to a maximum in the SOR. The uncertainties between 35-45 km are slightly larger in the northern subtropics compared to the southern subtropics. The uncertainties in the lower stratosphere between 22-28 km are smaller in magnitude, but this is partly because the SOR is also smaller here (note the confidence intervals are expressed as percent ozone to be directly

comparable to Figure 4). Overall, the 95% confidence intervals are around 30-50% of the magnitude of the 'best estimate' SOR in Figure 4 indicating that there are considerable uncertainties in the SOR in the SAGE II datasets. This has implications for understanding the contribution of the SOR to the climate response to the solar cycle.

Figures 4(c) and 4(d) show equivalent plots to 4(a) and 4(b) for SAGE II in units of mixing ratios on pressure  levels. The SORs between ∼50-10 hPa are very similar in the two versions  and strongly resemble Figures 4(a) and 4(b), with  a positive SOR in the tropical lower stratosphere of ∼1-2%. The  structures of the SOR between 20 and ∼7 hPa are also similar, with subtropical maxima of 1-2% and a distinct equatorial minimum. However, the  SORs in the upper stratosphere are markedly different between v6.2 and v7.0. Polewards of ±20° the  structure of the SORs are similar in both datasets, but the magnitude is ∼1% larger in v6.2. In the tropics, the v6.2 data show a large  peak in the SOR in the uppermost stratosphere of up to 5%, whereas the v7.0 data show a smaller  SOR of 1% in this region.  The confidence intervals for the SAGE II mixing ratio SORs in Figures 5(c) and 5(d) are generally similar to those for number densities, with the exception of the uncertainties being considerably larger in the tropical upper stratosphere in both datasets, but particularly in SAGE II v6.2. The relatively large uncertainties in the 'best estimate' of the SOR would feed through to similar uncertainties in the contribution of the SOR to the atmospheric response to the 11-year solar cycle (Gray et al., 2009; Shibata and Kodera, 2005) . It is therefore important to understand the causes of the differences in SOR between the SAGE II v6.2 and v7.0 datasets, since it presents a limitation for understanding and simulating the climate response to solar forcing (e.g. Ermolli et al. (2013); Mitchell et al. (2015b) ). This is explored in the next section.

**4.1.1   Differences in NMC/NCEP and MERRA-1 stratospheric temperature records**

Since the  two versions of SAGE II show comparable SORs for number densities, the differences between Figures 4(c) and 4(d) must be related to the conversion of SAGE II  data to ozone mixing ratios. As described in Section 2.1, SAGE II v6.2 employed NMC/NCEP temperature data for this conversion, but this was changed to  MERRA-1 for v7.0 (see Damadeo et al. (2013) for details). The differences in the  SOR in the upper stratosphere must therefore be related to the use of different temperature  records in the conversion. It is known that the evolution of  stratospheric temperatures in some reanalyses show unphysical variability and trends (Mitchell et al., 2015a), and these have been corrected for in some solar-climate studies (e.g. Frame and Gray (2010); Hood et al.

485 (2015)).  However, the effect of temperatures on the SOR in SAGE II data has not been considered previously. Indeed, spurious variations in stratospheric temperatures in  reanalyses datasets, which are introduced through changes in the observing system over time,  could mask or enhance the signal of the 11 year solar cycle in  SAGE II ozone mixing ratios.

490 Figure 6 shows timeseries of annual and tropical mean temperature anomalies at select  stratospheric levels (1, 2, 5, 10, 30 hPa) for the NMC/NCEP and  MERRA-1 datasets. The NMC/NCEP temperatures are those provided with the published SAGE II data files and cover 1985-2003. MERRA-1 data were downloaded for 1979-2013 from the NASA GFSC website. At 30 hPa, the evolution of the two temperature records is nearly identical during the 495 period of overlap, with a long-term cooling trend of $\sim$0.6 K decade$^{-1}$  that is strongly connected to an apparent step-wise cooling of $\sim$2 K  between 1992 and 1994. However, at pressures less than 30 hPa there are substantial differences between the records.  The NMC/NCEP data show  exceptional behaviour between 2000-03. At 1 hPa, 500  is a warming of more than 3 K over this short period, which is  coincident with a warming of $\sim$1 K at 2  hPa. In contrast, at 5 and 10 hPa there is a cooling of more than 4 and 2 K, respectively, over this period. The magnitude and vertical structure of these changes in the NMC/NCEP record seems inexplicable as to be related to any physical process, particularly 505 when compared to the variations found in the remainder of the record. Some of these issues may be related to the method used to construct the NMC/NCEP temperature record itself. NCEP reanalysis data were only available for pressures greater than 10 hPa, requiring the addition of operational analyses to extend the data to the stratopause. Data from an atmospheric model was used to further extend the temperature data to the mesosphere, but these levels are not considered here (see e.g. 510 Tomadeo et al. (2013) for more details). The NMC/NCEP temperature record used to convert SAGE II is therefore constructed from several component datasets. Regardless of the exact cause, it seems likely that some of the temperature variations in the NMC/NCEP record are spurious and this may impact on the diagnosed SOR in the SAGE II v6.2 mixing ratio data.

The temperature variations in MERRA-1 over the period 1985-2003 are generally smaller in 515 magnitude than those found in NMC/NCEP, with the exception of a marked cooling at 1 hPa of $\sim$3 K  between 2001-2003, which is opposite to what is seen in NMC/NCEP. This cooling in MERRA-1 leads the decline in solar forcing during the downward phase of solar cycle 23 by around a year, and is also larger in amplitude than typical solar signals in temperature at this level (Mitchell et al., 2015a) . However, the sign is at least consistent with the expected tendency 520  upper stratospheric temperatures during the declining phase of the solar cycle.

A valid question is which representation of past stratospheric temperatures is likely to be most realistic? Mitchell et al. (2015a) compared MERRA-1 to Stratospheric Sounding Unit (SSU) satellite data and found considerable differences in upper stratospheric temperature variability between the two records. However, the  NMC/NCEP data show a  long-term warming trend in the upper stratosphere, which is opposite to the cooling expected from increasing atmospheric $CO_2$ and declining ozone abundances over this period. Both records therefore appear to exhibit differences compared to observed stratospheric temperature changes.

The evolution of  atmospheric temperatures will affect the geometric altitude of a given pressure surface, as well as the conversion from number density to mixing ratio. It is well known that  cooling will lower the altitude of pressure surfaces, a so-called 'atmospheric shrinking' effect. Therefore the presence of cooling near the stratopause in  MERRA-1 would tend to lead to a greater atmospheric shrinking than for the NMC/NCEP temperatures. Furthermore, the conversion from number density to mixing ratio is proportional to temperature, so a positive correlation between number density and temperature over the solar cycle would tend to increase the  magnitude of the SOR on a given pressure surface. Figure 7 shows the annual mean solar cycle signals in stratospheric temperatures derived for  the NMC/NCEP  MERRA-1 datasets over the period 1985-2003. Although the  sign of the temperature signals are  consistent in most regions, the maximum warming in the tropics at solar maximum occurs at 4 hPa in  MERRA-1 as compared to 2 hPa in NMC/NCEP. The peak  magnitude of the solar cycle temperature response is also around 25% smaller in  MERRA-1 compared to NMC/NCEP. The impact of these differences on the SOR in SAGE II mixing ratio data are explored in the next section.

However, the

**4.1.2 Dependence of SOR in SAGE II mixing ratios on temperature record**

 test the impact of the differences between NMC/NCEP

 and MERRA-1 temperatures on the SOR in SAGE II, we perform our own conversion of  the SAGE II v6.2 data from number densities to mixing ratios. Each monthly and zonal mean ozone profile is first converted to number  densities on pressure levels  using the hydrostatic equation, and then to mixing ratios on pressure levels using the ideal gas law. The MLR in Equation 1 is then applied to the converted ozone mixing  ratios to derive a  SOR that can be compared to the published SAGE II mixing ratio datasets discussed above and shown in Figure 4.

As a first test, we convert  SAGE II v6.2 number  densities to mixing ratios using the full timeseries of temperatures from NMC/NCEP and  MERRA-1 in turn. The SORs diagnosed from these 'post-hoc' converted datasets are shown in Figures 8(a) and 8(b) for NMC/NCEP and MERRA-1, respectively,  with the difference between them shown in Figure 8(c). These can be compared to Figures 4(c)4(e). We stress that differences in the SORs are to be expected, since in the published SAGE II datasets each ozone profile is converted separately before averaging is performed, whereas here we  have converted the monthly, zonally and latitudinally averaged ozone number density profiles.

 SOR in the post-hoc converted data using NMC/NCEP temperatures (Figure 8(a)) shows a qualitatively similar structure to Figure 4(c), but the  magnitude is underestimated by  ∼2% in the is overestimated, which is an issue in all of the post-hoc converted fields. The tropical upper stratosphere. The SOR in the data converted using  MERRA-1 temperatures (Figure 8(b)) compares more closely with the original SAGE II v7.0 vmr dataset (Figure 4(d)). In particular, the reduced magnitude of the SOR in the tropical upper stratosphere is captured.

, which allows us to explore how differences in linear trends and solar cycle signals in temperature between NMC/NCEP and  MERRA-1 impact on the diagnosed SOR.

Figures 8(d) and 8(e) show the SOR for the SAGE II v6.2 data converted to mixing ratios using a monthly temperature climatology from MERRA-1 added to a latitude-height-time dependent linear trend and solar cycle term (see Figure 7) extracted from either NMC/NCEP (Figure 8(d)) or MERRA-1 (Figure 8(e)). The difference between Figures 8(d) and 8(e) is shown in Figure 8(f) for reference. Figures 8(d-f) are very similar to Figures 8(a-c) indicating that the majority of the difference in SOR in Figure 8(c) can be intepreted as due to differences in long-term trends and solar cycle variability in temperatures between NMC/NCEP and MERRA-1. Further tests (not shown) show that the diagnosed SORs are not affected by the choice of base temperature climatology (MERRA-1 or NMC/NCEP).

The remaining panels Figures 8(g-i) and 8(j-l) show equivalent results to Figures 8(d-f), but component of the temperature with the conversion to mixing ratios performed with the temperature climatology added to either the linear trend (Figures 8(g-i)) or solar cycle (Figures 8(j-l)) components of temperature variability from the two datasets. In both of these further tests, the SOR in the tropical upper stratosphere is larger for the SAGE II data converted using NMC/NCEP data (Figures 8(g,j)). This indicates that both components of the temperature variability contribute to the differences in SOR in Figure 8(c).

In conclusion, the SORs in SAGE II v6.2 and v7.0 are much more consistent in terms of number densities on altitude surfaces than they are for mixing ratios on pressure surfaces. The differences in SORs in the latter occur particularly in the upper stratosphere, and these have been shown to be sensitive to the details of the temperature records used for conversion. The long-term warming trend in the upper stratosphere in NMC/NCEP data is at odds with the understanding of recent changes in stratospheric composition and its impact on temperatures (Randel et al., 2009) ; however, the peak of the solar cycle signal in stratospheric temperatures in MERRA-1 is at lower altitude than predicted from theory and models. Recent analysis suggests that the relationship between ozone and temperature in the upper stratosphere that is anticipated from photochemical theory is more

realistic for the SAGE II v7.0 mixing ratio data than for v6.2 (Dhomse et al., 2015).

630  Nevertheless, there remain questions around which of the SAGE II mixing ratio datasets  is likely to be most credible for diagnosing the SOR. These results raise issues for the representation of the SOR in the CMIP5  ozone database, which was largely based on SAGE II v6.2

635  mixing ratios (Cionni et al. (2011) ; see also Maycock et al., in prep.).

**4.2  The SOR in extended SAGE II datasets**

 Given the uncertainties in the SOR for the SAGE II mixing ratio datasets discussed above, we focus our analysis of the extended SAGE II records on the three SI$^2$N datasets that are currently available as number densities (see Section

640 ): SAGE-GOMOS 1, SAGE-GOMOS 2, and SAGE-OSIRIS. Extending SAGE II using these more recent measurements increases the number of data points included in the MLR model by almost a factor of 2 in the tropics and by $\sim$50% in the subtropics (see Supplementary Material Figures S1 and S2). Figure 9 shows timeseries of monthly tropical percent ozone anomalies at select altitudes for the three SI$^2$N datasets. The datasets do not agree perfectly over the SAGE II era (1984-2004)

645 because the anomalies are defined relative to the entire timeseries, but overall they show similar behaviour to SAGE II v7.0 number densities (green line) in Figure 3, as expected. In the post-2004 period, where either GOMOS or OSIRIS data are included, the datasets show generally consistent behaviour in the mid-stratosphere during the overlap period up to 2011. QBO-like variations in ozone are visible in the timeseries at 24 and 31 km. At 36 km, there is a decline in ozone from 2004-09 in

650 all three datasets, with increases subsequent to this. However, in the upper stratosphere (48 km) there are more substantial differences between the datasets, particularly between the SAGE-GOMOS and SAGE-OSIRIS records. SAGE-OSIRIS shows mean positive anomalies from 2004-13, particularly in the latter part of the record, whereas the two SAGE-GOMOS datasets show negative anomalies between 2007-10, which coincide with the minimum of solar cycle 23. These differences in ozone

655 variability during the post-SAGE II period may affect the SORs diagnosed in the extended datasets, as compared to that found for the SAGE II v7.0 data alone (Figure 4(b)).

Figures 10(a-c) show the SORs in the three extended SAGE II datasets  and Figures 10(d-f) show their associated 95% confidence intervals in terms of percent ozone. An indication of the importance of how the  satellite records are merged for the

660  SOR can be seen by comparing Figures 10(a) and 10(b), which show the SOR in SAGE-GOMOS 1 and SAGE-GOMOS 2, respectively. The SOR in SAGE-GOMOS 1 shows  a generally smoother spatial structure as compared to SAGE-GOMOS 2,  although the magnitudes are not distinguishable from one another given the estimated confidence

intervals (Figures 10(d-e)). Nevertheless, since statistical uncertainties in the SOR are not typically
accounted for in solar-climate studies (e.g Gray et al. (2009) ) or in climate model ozone datasets
(e.g. Cionni et al. (2011) ), differences in the 'best estimate' of the SOR between the datasets remain
important to characterise. The differences in SOR between SAGE-GOMOS
1 and 2 must arise from differences in the data merging procedures, which are summarised by Tummon et al. (2015), and are described in  detail by Kyrölä et al. (2015) and Penckwitt et al.
(2015).  Analysis of the SOR in the two
SAGE-GOMOS datasets over the SAGE II period alone (1984-2004) reveals similar differences in
magnitude and structure (not shown), which suggests that the use of SAGE II or GOMOS as a reference to which the other record is adjusted is a key factor

for the differences in SOR. The uncertainties in the SOR in SAGE-GOMOS 2 (Figure 10()~~shows the merged SAGE II OSIRIS dataset. These data mostly show significant increases in ozone at solar maximum in the southern subtropics and in the tropics. This is similar to the results of Bourassa et al. (2014) , but they also find that the increase in the northern midlatitudes is statistically significant for the period 1985 to near present day (see their Figure 9). The absence of a significant change in ozone in the Northern hemisphere in the SAGE II OSIRIS dataset is also in contrast to the two~~ e)) are similar to
those found in the SAGE II v7.0 number density dataset (Figure 5(b)), whereas the magnitude of the
uncertainties in the SOR in SAGE-GOMOS ~~datasets, which both show significant increases in ozone in this region at pressures less than ~10hPa. Hubert et al. (2015) identified a significant positive drift of 5-8decade$^{-1}$ in OSIRIS data above 35km compared to ozonesondes and lidar measurements; this may contribute to the differences in the solar-ozone signal between~~ 1 (Figure 10(d)) are reduced
compared to SAGE II v7.0, particularly in the upper stratosphere.
The  SOR in the SAGE-OSIRIS dataset (Figure 10(c)) shows
 significant positive values in the subtropics between ~30-40  km. This is consistent with the results of Bourassa et al. (2014) who conducted a
similar MLR analysis to assess long-term ozone trends in SAGE-OSIRIS (see also Tummon et al. (2015) ).
 the SOR is smaller and less significant in the tropical upper stratosphere and northern
tropics as compared to the two SAGE-GOMOS datasets and the SAGE II  v7.0

significant increase in ozone in the lower tropical stratosphere of up to 5at 50 significant positive drift of 5-8

705~~There are several common features in the solar-ozone signals across the five extended datasets. These include a statistically significant increase in ozone in the mid and upper stratosphere, and an absence of ozone changes in the tropical mid stratosphere at ~10 hPa.~~

 .

decade$^{-1}$ in OSIRIS data above 35 km compared to ozonesondes and lidar measurements, which may contribute to the differences in SOR in the upper stratosphere.

Although there are  broad similarities in the SOR 715 the three extended SAGE II datasets there are also  some differences. This is despite the fact that  all of the datasets use SAGE II v7.0 number densities as a basis . There is therefore a trade-off between generating the longest climate data record possible, which is desirable for analysing quasi-decadal 720signals, and the introduction of additional sources of uncertainty from combining multiple satellite records with different sampling properties and drifts. There appear to be variations in ozone in the  OSIRIS record that reduce the magnitude of the SOR in the extended SAGE-OSIRIS record compared to the ~~solar-ozone signals differ by up to a factor of 3-4. These differences will have implications for the contribution of the solar-ozone response to stratospheric heating. They 725may therefore be important for understanding the climate response to solar forcing, including the contribution of the 'top-down' pathway to the surface climate response. From the results in this section, we conclude that whilst longer ozone records can be obtained by merging multiple datasets, this does not necessarily reduce the uncertainty in the solar-ozone response owing to the dependence of the signals on data selection and merging procedures.~~

730SAGE II period alone. When the SAGE-GOMOS datasets are analysed over the SAGE II period (1984-2004), SAGE-GOMOS 1 shows the greatest resemblance to the original SAGE II v7.0 data in Figure 4(b) (not shown) and we therefore conclude that this record is likely the most reliable estimate of the SOR from the datasets considered.

**4.3**

735 ### 4.3 The SOR in SBUV records

~~In addition to SAGE, the other long-term satellite record for stratospheric ozone is the SBUV dataset, which extends from 1970 to near the present day. Unlike for SAGE II , which represents measurements from a single instrument in continuous orbit, the SBUV record is formed from multiple instruments which have been launched on various satellite platforms. Thus, whilst the nadir-viewing SBUV instruments provide greater spatial and temporal sampling than the limb-viewing SAGE II , there are issues around inter-instrument calibration and merging the separate records. Owing to its viewing geometry, the vertical resolution of SBUV below ~15hPa is much poorer than for SAGE II making it more challenging to extract information about ozone in the middle and lower stratosphere.~~

Figure 11 shows timeseries of monthly percent ozone anomalies at select  stratospheric levels (as in Figure 3) for the  SBUVMOD VN8.0 (black line), SBUVMOD VN8.6 (red line), and SBUV Merged Cohesive VN8.6 (blue line) datasets.  At 1 hPa, the  ozone anomalies in the different datasets are in  good agreement between 1979-1994. After 1994, the main differences are found between the SBUMOV VN8.0 and the two SBUV VN8.6 datasets, the latter being largely consistent with one another. In particular, SBUVMOD VN8.0 shows a larger positive trend in ozone from the mid-1990s to the mid-2000s than in the  SBUV VN8.6 records; this partly coincides with the ascending phase of solar cycle 23. At 3 hPa,  a comparison of the three SBUV records reveals somewhat different behaviour. Here, the SBUVMOD VN8.0 and SBUV Merged Cohesive VN8.6  datasets show more similar ozone variations, and instead the SBUVMOD VN8.6 is an outlier exhibiting a larger decline in ozone compared to the other two records of ~7-8% over 1979-2012. At 5 hPa,  the three SBUV datasets generally show similar temporal variations in  ozone in the early and latter parts of the records, with some differences in offsets linked to different behaviours in the late 1990s and early 2000s when data come from the NOAA-11, 14, 16 and 17 satellites. At 30 hPa, the three SBUV records are largely consistent with one another in their short and long-term variations, with some exceptions during the 1990s when the data come mainly from the NOAA-11 and NOAA-14 satellites (see e.g. Tummon et al. (2015) ).

 Figures 5(a-c) show the annual mean SORs in the (a)  SBUVMOD VN8.0, (b) SBUVMOD VN8.6, and (c) SBUV Merged Cohesive VN8.6 datasets. ~~The SBUV VN8.0 and SBUV Merged Cohesive VN8.6 records show larger increases in ozone at solar maximum of up to 2-3in the upper stratosphere peaking at ~3hPa. In contrast, SBUVMOD VN8.6 only shows statistically significant increases of 1-2in the subtropics at around ±30°. All of the records show an equatorial minimum in the~~

mid stratosphere, although this is at higher pressure (5-10hPa) in SBUV VN8.0 compared to the SBUV VN8.6 records (3-5hPa). Between 10-50hPa, the SBUV VN8.6 based records show generally  Figures 5(d-f) show the associated 95% confidence intervals in terms of percent ozone. All three SBUV records show a significant positive SOR in some parts of the upper stratosphere of up to 2-3%  . The SOR in the tropical upper stratosphere is smaller and not highly statistically significant in SBUVMOD VN8.6, which is in contrast to the two other records and somewhat resembles the SOR in SAGE II v7.0 ~~. However, we note that the poor vertical resolution (~10km) of the SBUV instruments at pressures greater than ~15hPa means that there are large uncertainties in these features.~~

mixing ratios (Figure 4(d)). The modifications to the data processing algorithm between  SBUVMOD VN8.0 and SBUVMOD VN8.6 are documented by Bhartia et al. (2013); these include the use of new ozone absorption cross-sections, a new a priori ozone climatology, and a new cloud-height climatology. In addition, changes were also made to the inter-instrument calibration, which is now achieved at the radiance level during periods of overlap between the SBUV instruments (DeLand et al., 2012; Bhartia et al., 2013). It seems  plausible that calibration changes could impact on the diagnosis of quasi-decadal variability in ozone, and it seems possible that the new  processing procedure may have smoothed out  part of the SOR in the upper stratosphere in SBUVMOD VN8.6.

 Note that the difference in SOR in the tropical upper stratosphere between the two SBUV VN8.6 ~~datasets, Figure 5 highlights that a key issue for isolating the solar-ozone response relates to how the different records are merged to create a coherent timeseries. The selection of data included in a given time window, along with treatment of inter-instrument offsets and drifts, can have a substantial impact on the apparent variability and trends in ozone. Thus despite the same input data being used by the two SBUVrecords, they show markedly different solar-ozone signals in the upper stratosphere . Tummon et al. (2015) found that the SBUV Merged Cohesivedataset shows substantially different long-term ozone trends compared to a range of other satellite records . For example, in the tropical upper stratosphere SBUV Merged Cohesive VN8.6 showed a positive ozone trend over 1984-1997, whereas almost all other datasets analysed showed a substantial decline of several percent per decade. In constrast, they found that the trends in SBUVMOD VN8.6 were more consistent with the other records. However, it is not~~

810
* * *
815
820
  The SORs in the three SBUV records show further differences between 10-50   between 5-40  hPa, with
825  MOD  .6 showing a larger and more  VN8.  .6 shows a weaker SOR. However, we note that the poor vertical resolution ($\sim$10  km) of the SBUV instruments at pressures greater than $\sim$15
830

835  hPa makes it challenging to resolve features in the mid and lower stratosphere. Note that the confidence intervals for all the SBUV records are smaller than those for SAGE II based records (see Figure 5 and Figures 10(d-f)). This is likely to be because the

840   number of data points included in the MLR analysis is around 2-3 times higher for the SBUV datasets than for the SAGE records (see Supplementary Material Figures S1 and S3).

845 It is desirable for the purposes of e.g. chemistry-climate model evaluation to determine which SBUV dataset might be most reliable for estimating the annual mean SOR. Lean (2014) analysed total column ozone measurements from SBUVMOD VN8.0 and SBUVMOD VN8.6 and found a smaller SOR in SBUVMOD VN8.6 near-global column ozone, which appeared to be related to instrument effects around the 1996 time frame. However, Hood (1997) analysed the SOR in total 850 column ozone data and found that most of the

Instead, the main features in HALOE consist of increases in ozone in the subtropics of 3 signal is associated with ozone changes in the lower stratosphere that are linked to dynamical 855 processes. Column ozone measurements are therefore unlikely to be particularly helpful for constraining the SOR in the upper stratosphere where differences are found amongst many of the datasets analysed here and where the SOR is strongly determined by photochemical processes.

 in the 860  over 1984-1997, whereas almost all other datasets analysed, including SBUVMOD VN8.6 , showed a significant decline of several percent per decade over this period

**4.3.1    Sensitivity to time period**

870  . Instead, SBUV Merged Cohesive VN8.6 showed larger negative ozone trends that the other datasets between 5-10  hPa.

875 Wild and Long (2015) and Tummon et al. (2015) explain how the adjustments used to combine data from the ascending node of NOAA-11 with NOAA-9 and NOAA-14 in SBUV Merged Cohesive VN8.6 were determined from the overlap of the descending node of NOAA-11 with NOAA-16 because of known issues with the quality of data from NOAA-9 and NOAA-14 (Kramavora et al., 2013) . Since the NOAA-9 and NOAA-14 data coincide with the end of the trend analysis period used by

880 Tummon et al. (2015) , this could have had a particularly pronounced impact on their linear trend calculations, but may not be as important for diagnosing the SOR.

 From the timeseries of 1 hPa ozone anomalies shown in Figure 11, it would appear that differences between the two SBUV VN8.6 datasets in the early 2000s may be more important for determining the differences in SOR in the 885  upper stratosphere. During this period, which coincides with the maximum of solar cycle 22, SBUVMOD VN8.6 shows persistently more negative ozone anomalies than SBUV Merged Cohesive VN8.6. Further analysis of the SOR for the period up to the year 2000 (not shown) does produce a slightly larger and more significant SOR in the tropical upper stratosphere in SBUVMOD VN8.6 890 ~~Figure ??(g) shows the full 1979-2009 period for comparison. The signals diagnosed in the earlier part of the record show larger increases in ozone in the upper stratosphere and tropical lower stratosphere than are found in the later periods. The magnitude of the signal extracted for the full 31 year period lies in between these two representations. All of the six sub-periods shown in Figures ??(a-f) include the two major tropical volcanic eruptions that have occurred in the past 35 years (El 895 Chichn in April 1982 and Mt Pinatubo in June 1991), so it is unlikely that the differences amongst them are related to volcanic effects. The MLR in Equationdoes not include a linear trend term to represent CO$_2$ because for much of the period being considered EESC is also increasingdecline in EESC since the mid-1990s (see Figure 1(c)) , which has occurred alongside a continued increase in CO$_2$, could affect the results. To test this , we add a linear trend term into the MLR; however, this does not strongly affect the results as compared to Figure ?? (not shown). Therefore the differences betweensix sub-periods must arise from other time-dependent factors, such as inter-instrument calibration and merging, or indeed time-dependence of the solar-ozone signal itself.~~

tropical upper stratosphere in the SBUV records are small compared to the associated statistical 910 uncertainties (Figures 5(d-f)) and small compared to the differences in SOR between the two SAGE II mixing ratio datasets in this region. We therefore conclude that using the longest climate data record is most favourable for diagnosing the SOR, particularly on seasonal timescales (see Section 4.4), and in this case that is SBUVMOD VN8.6.

**4.4 Seasonality in the solar-ozone response**

The analysis thus far has described the annual mean SOR in satellite ozone datasets. However, the SOR is expected to exhibit a seasonal dependence; for example, in regions close to photochemical steady-state the annual cycle in solar zenith angle would be expected to produce a larger SOR in the summer hemisphere (Haigh, 1994). Furthermore, given the hypothesis that solar variability modifies the strength of the stratospheric polar vortex (Kuroda and Kodera, 2002), there may also be seasonal signatures in the SOR arising from dynamical processes, particularly in the winter hemispheres. Seasonal variations in the SOR could potentially influence the overall climate response to solar forcing through coupling to radiation (e.g. Hood et al. (2015)), and it is therefore important to characterise these in observations and chemistry-climate models.

Constraining the SOR on seasonal timescales requires high spatial and temporal data coverage; this is to ensure that any seasonal component of the signal can be resolved, but also to increase the number of degrees of freedom (i.e. the number of data points) available for the regression. Such coverage is not adequately provided by limb-viewing instruments, such as SAGE II, which have relatively sparse and infrequent sampling. The coverage is considerably better for nadir-viewing instruments like SBUV; however, as described above their vertical resolution is much poorer in the middle and lower stratosphere. There is therefore a trade-off between the information that can be usefully extracted from different data sources.

Given the denser sampling of SBUV compared to SAGE II, we focus here on the SBUVMOD VN8.6 dataset to examine the seasonality of the SOR. Figure 5 shows the monthly SOR in SBUVMOD VN8.6 for the period 1970-2012. These values are calculated by applying the MLR model to timeseries for individual months, and therefore no autocorrelation term has been included, since separate months are approximately uncorrelated from year-to-year. We note that the detailed magnitudes and structure of the monthly SORs are more sensitive to the choice of analysis epoch than for the annual mean SOR (not shown), but the broad features are generally consistent. The key point to take from Figure 5 is that there are substantially enhanced meridional and vertical gradients in the monthly SORs as compared to the annual mean SOR for SBUVMOD VN8.6 in Figure 5(b). This is similar to the conclusion reached by Hood et al. (2015).

950 Of the ozone response was determined by photochemical processes alone, one would expect a seasonal component associated with the annual cycle in solar zenith angle, with the largest anomalies expected in the summer hemisphere (Haigh, 1994) . However, away from regions in approximate photochemical steady-state localised ozone anomalies are also intimately tied to stratospheric dynamical variability, particularly in the winter hemisphere where intraseasonal variability in the polar vortex 955 key driver. Given the hypothesis that solar variability can modify the strength of the polar vortex (Kuroda and Kodera, 2002) , it follows that there may be a dynamical signature in the ozone changes, particularly in the winter hemisphere. This is evident in Figure 5 where there are particularly large gradients in ozone across the extratropics in July in the Southern hemisphere and in March in the Northern hemisphere.

960     Although much of the localised changes in ozone are clearly driven by variations in the SOR are *driven by* dynamical processes, it is also possible that they could feedback onto circulation through their impact the radiative impacts of ozone on stratospheric heating rates and temperatures. Hood et al. (2015) concluded that the three chemistry-climate models from CMIP5 that simulated simulate strong gradients in ozone in the winter upper stratosphere, which more closely resembled 965 observations, tended most closely resemble observations, tend to have high latitude dynamical responses that compared more favourably with are most similar to reanalysis data. It may therefore be important for such seasonal aspects of the solar-ozone response to be included in model simulations that lack interactive chemistry Seasonal variations in the SOR may therefore play a role in the ability of a model to simulate the climate response to solar variability. However, given the tight coupling 970 between ozone and dynamics, attribution of the importance of such radiative feedbacks is particularly challenging. To our knowledge, the importance of this two-way coupling for the solar-climate response climate response to solar variability has not been explicitly tested. This is important to clarify for modeling the impact of solar variability on climate because it is not known whether it is sufficient to simply prescribe a seasonally-varying solar-ozone signal SOR, or whether a fully interactive 975 chemistry-climate model is required which can to capture the coupling and feedbacks between composition, radiation and dynamics over the solar cycle. The representation of the solar-ozone response in global atmospheric SOR in global climate models is discussed in more detail in Part II of this study (Maycock et al., in prep.).

**5   Discussion**

980 **5   Conclusions**

The representation of the annual mean solar-ozone response has been analysed in many of the available satellite ozone datasets. Despite there being considerable differences between individual instruments and the techniques adopted to merge multiple records, there are some consistent features in all of the datasets. Every dataset shows a statistically significant increase in ozone at solar maximum

somewhere in the region 1-50hPa, ±60°. The magnitude of the peak increase in ozone ranges from ~1-5. An increase in ozone upon an increase in solar ultraviolet radiation is consistent with our understanding of photochemical processes in the stratosphere and results from chemistry-climate models (Haigh, 1994; Austin et al., 2008) .

However, despite all the datasets showing an increase in ozone there are marked differences in the vertical and horizontal structures of the signals. Some of the differences have been shown to be particularly sensitive to the post-processing of data; for example, the importance of the stratospheric temperature record for the conversion of SAGE II from number density to mixing ratio (see Section 4.1) , or the method employed for merging SBUV records to create a consistent timeseries (see Section 4.3). More recent combined ozone datasets, which append other records to the SAGE II timeseries, can provide a longer record which is useful for constraining quasi-decadal signals. However, the results from these records are also sensitive to the methods for combining independent records, as demonstrated by the differences between two versions of the SAGE-GOMOS dataset.

The differences in the magnitude and structure of the solar-ozone response across datasets have important implications for understanding the (SOR) forms an important part of the climate response to solar variability. The impact of changes in irradiance and ozone on stratospheric heating rates, and therefore on stratospheric temperatures , are strongly height-dependent. Therefore the different solar-ozone signals would lead to different solar cycle signatures in stratospheric temperatures 11-year solar cycle variability through its impact on stratospheric temperatures (e.g. Shibata and Kodera (2005); Gray et al. (2009)). Since one of the leading 'top-down' mechanisms for solar-climate coupling is related to radiatively-driven changes in meridional temperature gradients in the upper stratosphere (e.g. Gray et al. (2010) ), it is important to constrain the contribution of ozone to this anomalous heating.

Soukharev and Hood (2006) concluded from their MLR analysis of 3-month mean ozone data that the solar signals in SAGE II v6.2 vmr, SBUV VN8.0 and HALOE were comparable enough to create a multi-instrument mean response; this was subsequently used to evaluate the CCMVal-1 models (Austin et al., 2008) . However, the analysis presented here shows that the differences between individual records, which have recently been reprocessed in different ways, are often as large as the mean response, and we conclude that this precludes the formulation of a multi-instrument mean signal.

Chemistry-climate modelsmay be useful tools for constraining the solar-ozone response, at least in the annual mean; however, there remain considerable uncertainties in the characteristics of spectral solar irradiance (SSI) variability (Ermolli et al., 2013) , which must be prescribed in models, and which will therefore strongly determine the solar-ozone response (Haigh et al., 2010) . Other studies have developed methods aimed at using ozone observations to constrain SSI variability (Ball et al., 2014) , but as has been shown here, the differences between individual records are typically too large to provide a stringent constraint.

This papers forms the first of a two-part study that aims to quantify the SOR in current satellite observations and chemistry-climate models.

It is therefore important to quantify the solar-ozone signal to improve our understanding and ability to model the influence of solar variability on climate. Many global climate models, such as those participating in CMIP exercises, do not currently represent stratospheric chemical processes. As a result, several ozone databases have been created for long-term climate model studies. To allow for a realistic representation of the impacts of solar variability on climate, these datasets must include a solar-ozone signal. Such a signal could be derived from observations and Part I has focused on comparing the SOR in recently updated and extended versions of long-term satellite ozone datasets (e.g. SAGE II, SBUV) with their previous counterparts (e.g. Soukharev and Hood (2006); Austin et al. (2008) ).

The SAGE II dataset has been widely used for ozone studies because of its long-term stability. SAGE II ozone data are available as number densities on altitude levels and post-processed to mixing ratios on pressure levels. The SAGE II version 6.2 (v6.2) mixing ratio dataset shows a positive annual mean SOR with a peak magnitude of ∼5% near the tropical stratopause. However, the more recent SAGE II v7.0 dataset shows substantially smaller SOR at the tropical stratopause of ∼1%. Conversely, the SORs in the equivalent SAGE II number density datasets are much more consistent for v6.2 and v7.0, and show a three peaked structure in the tropics/subtropics with a magnitude of up to 3-4%.

By applying a post-hoc method to convert SAGE II number densities to mixing ratios, we have shown that the differences in SOR mostly arise from the change in independent temperature record used by the SAGE II team to convert number densities to mixing ratios: v6.2 uses NMC/NCEP and v7.0 uses MERRA-1 temperatures. Differences between these temperature records in both long-term trends and solar cycle variations contribute to the

 changes in SOR described above. Since both temperature records contain known issues (e.g. Damadeo et al. (2013); Mitchell et al. (2015a) ), we conclude that the latest SAGE II v7.0 ozone number densities are likely to be most  reliable for estimating the SOR at the present time. This is an important conclusion because several of the existing ozone datasets developed for use in global climate models have been based on SAGE II v6.2 mixing ratio data, including the dataset developed for CMIP5 simulations (Cionni et al., 2011) .

 We further analysed the annual mean SOR in three extended SAGE II  datasets that have merged more recent GOMOS (2002-11) or OSIRIS (2002-13) data with SAGE II v7.0 number densities. Two SAGE-GOMOS datasets were analysed that adopt different methods for merging the satellite records (Kyrölä et al., 2015; Penckwitt et al., 2015) . These records show broadly similar SORs, but the dataset that uses SAGE II as a reference and adjusts GOMOS using seasonally-varying offsets at each latitude and altitude (Penckwitt et al., 2015) was found to have a SOR with a noisier spatial structure. The SAGE-OSIRIS dataset (Bourassa et al., 2014) shows a significant positive SOR of ∼2% between 30-40 km, but a weaker and less significant SOR in the tropical upper stratosphere than is found in the SAGE-GOMOS datasets . Thus the inclusion of OSIRIS data results in a markedly different SOR to that found in the SAGE II v7.0 number densities that underpin the first part of the record. Given these various issues, we conclude that the SAGE-GOMOS 1 dataset (Kyrölä et al., 2015) is likely to be the most reliable extended SAGE II dataset for estimating the SOR at the present time.

Analysis of the  recently released SBUVMOD VN8.6 data produced by NASA show a smaller  SOR in the tropical upper stratosphere by ∼1% compared to the previous SBUVMOD VN8.0 data (Soukharev and Hood, 2006). However, the SBUV Merged Cohesive VN8.6 dataset from NOAA, which takes a different approach for combining  the individual SBUV VN8.6 records, shows a  SOR that more closely matches  SBUVMOD VN8.0 . Nevertheless, the differences in the magnitude of the SOR between the various SBUV records are generally smaller than those

between the SAGE II v6.2 and v7.0 mixing ratio datasets and are not highly statistically significant given the estimated uncertainties in the SOR from the regression model. We therefore suggest that the SBUVMOD VN8.6 dataset is most appropriate for analysing the SOR since it is the longest of the currently available SBUV records (1970-2012).

 1100  Analysis of the SOR on monthly timescales in the SBUVMOD VN8.6 dataset reveals larger horizontal and 1105 vertical gradients in the SOR, particularly in the winter extratropics. Hood et al. (2015) analysed CMIP5 models with interactive chemistry and concluded that the models with seasonal variations in the SOR that best matched observations simulated changes in high latitude zonal winds that more closely resemble reanalysis data. Seasonal variations in the SOR may therefore be important for the climate response to solar variability, but the quantitative importance of this feedback for stratospheric 1110 dynamics remains to be tested.

 To allow for a realistic representation of the climate impacts of solar variability in models, simulations should include the effects of both the SOR and variations in spectral solar irradiance (Matthes et al., 2016) . Our results raise issues for how to best  1115  represent the SOR in 'non-interactive' climate models for which the SOR much be externally prescribed. For example, ozone databases for climate models are usually created using a variety of ozone measurements, and therefore implicitly include a representation of the SOR that 1120 emerges from whichever combinations of data are included (e.g. Cionni et al. (2011); Bodeker et al. (2013) ). However, the differences in the  magnitude and structure of the 'best estimate' SOR between the various satellite datasets presented here would likely result in different climate responses to solar forcing. There is therefore a need for new studies to explore the effects of uncertainties in the SOR for climate simulation, particularly in light of the 1125 substantial, but largely unexplained, spread in ~~stratospheric temperature responses across CMIP5 models (Mitchell et al., 2015a) . This should therefore be improved in CMIP6. It is also desirable for seasonal effects , which were excluded in models without chemistry in CMIP5 (see Maycock et al., in prep.; Hood et al. (2015) ), and which are evident in the available observational records (Figure 5), to be incorporated, although the importance of having full coupling between chemistry and dynamicsremains unclear. We conclude that if a more consistent representation of the solar-ozone response can~~

be achieved in CMIP6 it will aid in understanding the response to solar variability in models climate responses to the 11 year solar cycle across CMIP5 models (Mitchell et al., 2015b; Hood et al., 2015) .

*Acknowledgements.* ACM acknowledges funding from an AXA Postdoctoral Fellowshipand , the ERC ACCI grant. Grant Project No. 267760, and a NERC Independent Research Fellowship (NE/M018199/1). ACM also acknowledges funding from the COST action ES1005 Towards a more complete assessment of the impact of solar variability on the Earth's climate (TOSCA) for a Short-term Scientific Mission to GEOMAR in September 2014 which initiated this work. Parts of the work at GEOMAR Helmholtz Centre for Ocean Research Kiel was performed within the Helmholtz-University Young Investigators Group NATHAN, funded by the Helmholtz-Association and GEOMAR. We thank Ray Wang for providing useful information about the SAGE II record and Stacey Frith for providing useful information about the SBUV recordrecords. We also thank the many instrument scientists and groups who have contributed to the development of the merged SAGE-GOMOS 1, SAGE-GOMOS 2, and SAGE II OSIRIS , SWOOSH and GOZCARDS datasets, and for having made their data available for this study.

**References**

J. Austin, K. Tourpali, E. Rozanov, H. Akiyoshi, S. Bekki, G. Bodeker, C. Brühl, N. Butchart, M. Chipperfield, M. Deushi, V. I. Fomichev, M. A. Giorgetta, L. Gray, K. Kodera, F. Lott, E. Manzini, D. Marsh, K. Matthes, T. Nagashima, K. Shibata, R. S. Stolarski, H. Struthers, and W. Tian. Coupled chemistry climate model simulations of the solar cycle in ozone and temperature. *J. Geophys. Res.*, 113:D11306, 2008.

W. T. Ball, D. J. Mortlock, J. S. Egerton, and J. D. Haigh. Assessing the relationship between spectral solar irradiance and stratospheric ozone using Bayesian inference. *J. Space Weather Space Clim.*, 4:A25, 2014.

W. T. Ball, J. D. Haigh, E. V. Rozanov, A. Kuchar, T. Sukhodolov, F. Tummon, A. V. Shapiro, and W. Schmutz. High solar cycle spectral variations inconsistent with stratospheric ozone observations. *Nature Geoscience*, 9:3, 206-209, 2016.

P. K. Bhartia, R. D. McPeters, L. E. Flynn, S. Taylor, N. A. Kramarova, S. Frith, B. Fisher, and M. DeLand. Solar Backscatter UV (SBUV) total ozone and profile algorithm *Atmos. Meas. Tech.*, 6, 2533-2548 2013.

G. E. Bodeker, B. Hassler, P. J. Young, and R. W. Portmann. A vertically resolved, global, gap-free ozone database for assessing or constraining global climate model simulations. *Earth Syst. Sci. Data*, 5:31–43, 2013.

A. E. Bourassa, D. A. Degenstein, W. J. Randel, J. M. Zawodny, E. Kyrölä, C. A. McLinden, C. E. Sioris, and C. Z. Roth. Trends in stratospheric ozone derived from merged SAGE II and Odin-OSIRIS satellite observations. *Atmos. Chem. Phys.*, 14:6983–6994, 2014.

G. Chiodo, D. R. Marsh, R. Garcia-Herrera, N. Calvo, and J. A. Garcia. On the detection of the solar signal in the tropical stratosphere. *Atmos. Chem. Phys.*, 14:5251-5269, 2014.

I. Cionni, V. Eyring, J-F. Lamarque, W. J. Randel, D. S. Stevenson, F. Wu, G. E. Bodeker, T. G. Shepherd, D. T. Shindell, and D. W. Waugh. Ozone database in support of CMIP5 simulations: Results and corresponding radiative forcing. *Atmos. Chem. Phys.*, 11:11,267–11,292, 2011.

R. P. Damadeo, J. M. Zawodny, L. W. Thomason, and N. Iyer. SAGE version 7.0 algorithm: application to SAGE II. *Atmos. Meas. Tech.*, 6:3539–3561, 2013.

S. M. Davis, K. H. Rosenlof, B. Hassler, D. F. Hurst, W. G. Read, H. Vomel, H. Selkirk, M. Fujiwara, and  Damadeo. The Stratospheric Water and Ozone Satellite Homogenized (SWOOSH) database: A long-term database for climate studies. *Earth Syst. Sci. Data Discuss*,  doi:10.5194/essd-2016-16, submitted, 2016.

J. de Grandpré, S. R. Beagley, V. I. Fomichev, E. Griffioen, J. C. McConnell, A. S. Medvedev, and T. G. Shepherd. Ozone climatology using interactive chemistry: Results from the Canadian Middle Atmosphere Model. *J. Geophys. Res.*, 105:26,475–26,491, 2000.

D. P. Dee, S. M. Uppala, A. J. Simmons, P. Berrisford, P. Poli, S. Kobayashi, U. Andrae, M. A. Balmaseda, G. Balsamo, P. Bauer, P. Bechtold, A. C. M. Beljaars, L. van de Berg, J. Bidlot, N. Bormann, C. Delsol, R. Dragani, M. Fuentes, A. J. Geer, L. Haimberger, S. B. Healy, H. Hersbach, E. V. Hólm, L. Isaksen, P. Kållberg, M. Köhler, M. Matricardi, A. P. McNally, B. M. Monge-Sanz, J.-J. Morcrette, B.-K. Park, C. Peubey, P. de Rosnay, C. Tavolato, J.-N. Thépaut, and F. Vitart. The ERA-Interim reanalysis: configuration and performance of the data assimilation system. *Q. J. R. Meteorol. Soc.*, 137(656):553–597, 2011.

M. T. DeLand, S. L. Taylor, L. K. Huang, and B. L. Fisher. Calibration of the SBUV version 8.6 ozone data product *Atmos. Meas. Tech.*, 5:2951-2967, 2012.

S. Dhomse, M. P. Chipperfield, W. Feng, and J. D. Haigh. Solar response in tropical stratospheric ozone: a 3-D chemical transport model study using ERA reanalyses. *Atmos. Chem. Phys.*, 11:12773–12786, 2011.

S. Dhomse, M. P. Chipperfield, R. P. Damadeo, J. M. Zawodny, W. Ball, W. Feng, R. Hossaini, G. W. Mann and J. D. Haigh. On the ambiguous nature of the 11-year solar cycle signal profile in stratospheric ozone. *Geophys. Res. Letts.*, submitted,  2016.

I. Ermolli, K. Matthes, T. Dudok de Wit, N. A. Krivova, K. Tourpali, M. Weber, Y. C. Unruh, L. Gray, U. Langematz, P. Pilewskie, E. Rozanov, W. Schmutz, A. Shapiro, S. K. Solanki, G. Thuillier, and T. N. Woods. Recent variability of the solar spectral irradiance and its impact on climate modelling. *Atmos. Chem. Phys.*, 13: 3945–3977, 2013.

P. M. Forster, V. I. Fomichev, E. Rozanov, C. Cagnazzo, A. I. Jonsson, U. Langematz, B. Fomin, M. J. Iacono, B. Mayer, E. Mlawer, G. Myhre, R. W. Portmann, H. Akiyoshi, V. Falaleeva, N. Gillett, A. Karpechko, J. Li, P. Lemennais, O. Morgenstern, S. Oberländer, M. Sigmond, and K. Shibata. Evaluation of radiation scheme performance within chemistry-climate models. *J. Geophys. Res.*, 116:D10302, 2011.

T. H. A. Frame and L. J. Gray. The 11-yr solar cycle in ERA-40 date: An update to 2008. *J. Climate*, 23: 2213–2222, 2010.

S. M. Frith, N. A. Kramarova, R. S. Stolarski, R. D. McPeters, P. K. Bhartia and G. J. Labow. Recent changes in column ozone based on the SBUV version 8.6 merged ozone database. *J. Geophys. Res.*, 119:9735–9751, 2014.

L. Froidevaux, J. Anderson, H.-J. Wang, R. A. Fuller, M. J. Schwartz, M. L. Santee, N. J. Livesey, H. C. Pumphrey, P. Bernath, J. M. Russell III and M. P. McCormick. Global Ozone Chemistry And Related Datasets for the Stratosphere (GOZCARDS): methodology and sample results with a focus on HCl, H2O, and O3. *Atmos. Chem. Phys. Discuss.*, 15:5849-5957, 2015.

L. J. Gray, S. Rumbold, and K. P. Shine. Stratospheric temperatures and radiative forcing response to 11-year solar cycle changes in irradiance and ozone. *J. Atmos. Sci.*, 66:2402–2417, 2009.

L. J. Gray, J. Beer, M. Geller, J. D. Haigh, M. Lockwood, K. Matthes, U. Cubasch, D. Fleitmann, G. Harrison, L. Hood, J. Luterbacher, G. A. Meehl, D. Shindell, B. van Geel, and W. White. Solar influences on climate. *Rev. Geophys.*, 48:RG4001, 2010.

J. U. Grooß and J. M. Russell. Technical note: A stratospheric climatology for $o_3$, $h_2o$ and $ch_4$ derived from HALOE measurements. *Atmos. Chem. Phys.*, 5:2797–2807, 2005.

J. D. Haigh. The role of stratospheric ozone in modulating the solar radiative forcing of climate. *Nature*, 370: 544–546, 1994.

J. D. Haigh., A. R. Winning, R. Toumi, and J. W. Harder. An influence of solar spectral variations on radiative forcing of climate. *Nature*, 467:696–699, 2010.

N. R. P. Harris, B. Hassler, F. Tummon, G. E. Bodeker, D. Hubert, I. Petropavlovsikh, W. Steinbrecht, J. Anderson, P. K. Bhartia, C. D. Boone, A. Bourassa, S. M. Davis, D. Degenstein, A. Delcloo, S. M. Frith, L. Froidevaux, S. Godin-Beekmann, N. Jones, M. J. Kurylo, E. Kyrölä, M. Laine, S.T. Leblanc, J.C. Lambert, E. Mahieu, A. C. Maycock, M. de Maziere, A. Parrish, R. Querel, K. H. Rosenlof, C. Roth, C. Sioris, B. Liley, J. Staehelin, R. S. Stolarski, R. Stubi, J. Tamminen, C. Vigouroux, K. Walker, H. J. Wang, J. Wild, and

J. M. Zawodny. Past changes in the Vertical Distribution of Ozone, Part III: Analysis and Interpretation of Trends. *Atmos. Chem. Phys. Diss.*, 15:8565–8608, 2015.

1225 M. I. Hegglin, D. Plummer, O. Morgenstern. An ozone dataset for CMIP6. In preparation for *Geosci. Mod. Dev. CMIP6 special issue*.

L. L. Hood. The solar cycle variation of total ozone: Dynamical forcing in the lower stratosphere. *J. Geophys. Res.*, 102, 1355–1370. doi:10.1029/96JD00210.

L. L. Hood, S. Misios, D. M. Mitchell, E. Rozanov, L. J. Gray, K. Tourpali, K. Matthes, H. Schmidt, G. Chiodo,
1230 R. Thiéblemont, D. Shindell, and A. Krivolutsky. Solar signals in cmip-5 simulations: The ozone response. *Q. J. Roy. Meteorol. Soc.*,  141, 2670-2689, 2015. doi:10.1002/qj.2553.

D. Hubert, J. -C. Lambert, T. Verhoelst, J. Granville, A. Keppens, J. -L. Baray, U. Cortesi, D. A. Degenstein, L. Froidevaux, S. Godin-Beekmann, K. W. Hoppel, E. Kyrölä, T. Leblanc, G. Lichtenberg, C. T. McElroy, D. Murtagh, H. Nakane, J. M. Russell III, J. Salvador, H. G. J. Smit, K. Stebel, W. Steinbrecht, K. B. Straw-
1235 bridge, R. Stübi, D. P. J. Swart, G. Taha, A. M. Thompson, J. Urban, J. A. E. van Gijsel, P. von der Gathen, K. A. Walker, E. Wolfram, and J. M. Zawodny. Ground-based assessment of the bias and long-term stability of fourteen limb and occultation ozone profile data records. *Atmos. Meas. Tech. Diss.*, 8:6661-6757, 2015.

S. Ineson, A. Scaife, Jeff R. Knight, James C. Manners, Nick J. Dunstone, Lesley J. Gray, and Joanna D. Haigh. Solar forcing of winter climate variability in the Northern Hemisphere. *Nature Geoscience*, 4:753–757, 2011.
1240 , A. C. Maycock, L. J. Gray, A. A. Scaife, N. J. Dunstone, J. W. Harder, J. R. Knight, M. Lockwood, J. C. Manners, and R. A. Wood. Regional climate impacts of a possible future grand solar minimum. *Nature Communications*,  6, 7535, 2015.

Kramarova, N. A., Frith, S. M., Bhartia, P. K., McPeters, R. D., Taylor, S. L., Fisher, B. L., Labow, G. J.,
1245 and DeLand, M. T. Validation of ozone monthly zonal mean profiles obtained from the version 8.6 Solar Backscatter Ultraviolet algorithm. *Atmos. Chem. Phys.*, 13, 6887-6905 2013.

Y. Kuroda and K. Kodera. Effect of solar activity on the Polar-night Jet Oscillation in the Northern and Southern hemisphere winter. *J. Met. Soc. Japan*, 80:973–984, 2002.

E. Kyrölä, M. Laine, V. Sofieva, J. Tamminen, S. M. Päivärinta, S. Tukiainen, J. Zawodny and L. Thoma-
1250 son. Combined SAGE II-GOMOS ozone profile data set for 1984–2011 and trend analysis of the vertical distribution of ozone. *Atmos. Chem. Phys.*, 13:10645–10658, 2013.

J. Lean. Evolution of the Sun's Spectral Irradiance Since the Maunder Minimum. *Geophys. Res. Letts.*, 27:16, 2425-2428, 2000.

J. Lean. Evolution of Total Atmospheric Ozone from 1900 to 2100 Estimated with Statistical Models. *J. Atmos.*
1255 *Sci.*, 71:6, 1956-1984, 2014.

A. C. Maycock, K. Matthes, S. Tegtmeier, R. Thiéblemont, L. L. Hood. Solar cycle signals in stratospheric ozone. Part II: Analysis of Climate Models. *Atmos. Chem. Phys. Disc.*, in preparation.

K. Matthes, U. Langematz, L. J. Gray, K. Kodera, and K. Labitzke. Improved 11-year solar signal in the Freie Universität Berlin Climate Middle Atmosphere Model (FUB-CMAM). *J. Geophys. Res.*, 109:D06101, 2004.
1260 K. Matthes, Y. Kuroda, K. Kodera, and U. Langematz. Transfer of the solar signal from the stratosphere to the troposphere: Northern winter. *J. Geophys. Res.*, 111:D06108, 2006.

K. Matthes, et al. Solar forcing recommendation for CMIP6. *Geosci. Mod. Devel. Discuss.*, submitted, 2016.

C. A. McLinden, S. Tegtmeier, and V. Fioletov. Technical note: A SAGE-corrected SBUV zonal-mean ozone data set. *Atmos. Chem. Phys.*, 9:7963–7972, 2009.

R. D. McPeters, T. Miles, L. E. Flynn, C. G. Wellemeyer, and J. M. Zawodny. Comparison of SBUV and SAGE II ozone profiles: Implications for ozone trends. *J. Geophys. Res.*, 99:20513–20524, 1994.

R. D. McPeters, P. K. Bhartia, D. Haffner, G. J. Labow, and L. Flynn. The version 8.6 SBUV ozone data record: An overview. *J. Geophys. Res.*, 1168:8032–8039, 2013.

D. M. Mitchell, L. J. Gray, M. Fujiwara, T. Hibino, J. Anstey, Y. Harada, C. Long, S. Misios, P. A. Stott, and D. Tan. Signatures of natural variability in the atmosphere using multiple reanalysis datasets. *Q. J. Roy. Meteorol. Soc.*, in press, 2015a. doi:10.1002/qj.2492.

D. M. Mitchell, S. Misios, L. J. Gray, K. Tourpali, K. Matthes, L. L. Hood, H. Schmidt, G. Chiodo, R. Thiéble-mont, E. Rozanov, D. Shindell, and A. Krivolutsky. Solar signals in cmip-5 simulations: the stratospheric pathway. *Q. J. Roy. Meteorol. Soc.*, in press, 2015b. doi:10.1002/qj.2530.

K. M. Nissen, K. Matthes, U. Langematz, and B. Mayer. Towards a better representation of the solar cycle in general circulation models. *Atmos. Chem. Phys.*, 7:5391–5400, 2007.

A. A. Penckwitt, G. E. Bodeker, L. E. Revell, L. Richter, E. Kyrölä and P. Young. Construction and analysis of a new merged SAGE II-GOMOS ozone profile data set for 1984–2012. *Earth Syst. Sci. Data*, in prep., 2015.

V. Ramaswamy, M. -L. Chanin, J. Angell, J. Barnett, D. Gaffen, M. Gelman, P. Keckhut, Y. Koshelkov, K. Labitzke, J. -J. R. Lin, A. O'Neill, J. Nash, W. Randel, R. Rood, K. Shine, M. Shiotani, R. Swinbank. Stratospheric temperature trends: Observations and model simulations. *Rev. Geophys.*, 39:71-122, 2001.

W. J. Randel and F. Wu. A stratospheric ozone profile data set for 1979–2005: Variability, trends, and comparisons with column ozone data. *J. Geophys. Res.*, 112:D06313, 2007.

W. J. Randel, K. P. Shine, J. Austin, J. Barnett, C. Claud, N. P. Gillett, P. Keckhut, U. Langematz, R. Lin, C. Long, C. Mears, A. Miller, J. Nash, D. J. Seidel, D. W. J. Thompson, F. Wu, S. Yoden An update of observed stratospheric temperature trends. *J. Geophys. Res.*, 114:D010421, 2009.

E. E. Remsberg. Decadal-scale responses in middle and upper stratospheric ozone from SAGE II version 7 data. *Atmos. Chem. Phys.*, 14:1039–1053, 2014.

E. E. Remsberg and G. Lingenfelser. Analysis of SAGE II ozone of the middle and upper stratosphere for its response to a decadal-scale forcing. *Atmos. Chem. Phys.*, 10:11779–11790, 2010.

M. Sato, J. E. Hansen, M. P. McCormick, and J.B. Pollack. Stratospheric aerosol optical depth, 1850-1990. *J. Geophys. Res.*, 98:22,987–22,994, 1993.

K Shibata and K. Kodera. Simulation of radiative and dynamical responses of the middle atmosphere to the 11-year solar cycle. *J. Atmos. Sol. Terr. Phys.*, 67:125–143, 2005.

C. E. Sioris, C. A. McLinden, V. E. Fioletov, C. Adams, J. M. Zawodny, A. E. Bourassa, C. Z. Roth and D. A. Degenstein. Trend and variability in ozone in the tropical lower stratosphere over 2.5 solar cycles observed by SAGE II and OSIRIS. *Atmos. Chem. Phys*, 14:3479–3496, 2014.

B. E. Soukharev and L. L. Hood. Solar cycle variation of stratospheric ozone: Multiple regression analysis of long-term satellite data sets and comparisons with models. *J. Geophys. Res.*, 111:D20314, 2006.

SPARC CCMVal. SPARC report on the evaluation of Chemistry-Climate Models [V. Eyring, T. G. Shepherd and D. Waugh (Eds.)], SPARC Report No. 5, WCRP-132,WMO/TD-No. 1526, 2010.

S. Tegtmeier, M. I. Hegglin, J. Anderson, A. Bourassa, S. Brohede, D. Degenstein, L. Froidevaux, R. Fuller, B. Funke, J. Gille, A. Jones, Y. Kasai, K. Krüger, E. Kyrölä, G. Lingenfelser, J. Lumpe, B. Nardi, J. Neu, D. Pendlebury, E. Remsberg, A. Rozanov, L. Smith, M. Toohey, J. Urban, T. von Clarmann, K. A. Walker, and R. H. J. Wang. SPARC Data Initiative: A comparison of ozone climatologies from international satellite limb sounders. *J. Geophys. Res.*, 118:12,229–12,247, 2013.

R. Thiéblemont, K. Matthes, N.-E. Omrani, K. Kodera, F. Hansen. Solar forcing synchronizes decadal North Atlantic climate variability. *Nature Communications*, 6:8268, 2015.

D. W. J. Thompson, D. J. Seidel, W. J. Randel, C.-Z. Zou, A. H. Butler, C. Mears, A. Osso, C. Long, and R. Lin. The mystery of recent stratospheric temperature trends. *Nature*, 491:692–697, 2012.

M. Toohey, M. I. Hegglin, S. Tegtmeier, J. Anderson, J. A. Añel, A. Bourassa, S. Brohede, D. Degenstein, L. Froidevaux, R. Fuller, B. Funke, J. Gille, A. Jones, Y. Kasai, K. Krüger, E. Kyrölä, J. Neu, A. Rozanov, L. Smith, J. Urban, T. von Clarmann, K. A. Walker, and R. H. J. Wang. Characterizing sampling biases in the trace gas climatologies of the SPARC Data Initiative. *J. Geophys. Res.*, 118:11,847–11,862, 2013.

F. Tummon, B. Hassler, N. R. P. Harris, J. Staehelin, W. Steinbrecht, J. Anderson, G. E. Bodeker, A. Bourassa, S. M. Davis, D. Degenstein, S. M. Frith, L. Froidevaux, E. Kyrölä, M. Laine, C. Long, A. A. Penckwitt, C. E. Sioris, K. H. Rosenlof, C. Roth, H. J. Wang, and J. Wild. Intercomparison of vertically resolved merged satellite ozone data sets: interannual variability and long-term trends. *Atmos. Chem. Phys.*, 15:3021-3043, 2015.

H. J. Wang, D. M. Cunnold, L. W. Thomason, J. M. Zawodny, and G. E. Bodeker. Assessment of SAGE version 6.1 ozone data quality. *J. Geophys. Res.*, 107:D234691, 2002.

Y.-M. Wang, J. L. Lean, and N. R. Shelley. Modeling the Sun's magnetic field and irradiance since 1713. *J. Astrophys.*, 625:522–538, 2005.

J. D. Wild and C. S. Long A Coherent Ozone Profile Dataset from SBUV, SBUV/2: 1979 to 2013. *in prep.*, 2015.

[Figure]

**Figure 1.** Timeseries of the six basis functions used in  the MLR analysis. (a) Solar forcing based on F10.7cm solar radio flux; (b)  a trend term based on the monthly $CO_2$ concentration at Mauna Loa; (c) Equivalent effective stratospheric chlorine; (d) the Nino 3.4 index for ENSO; (e, f) two  QBO indices  based on tropical zonal winds at 50 and 30 hPa. The timeseries are in units of standard deviation and the time period is 1970-2015. A volcanic term is not included because the 2 year periods following the two major tropical eruptions in this epoch (El Chichon and Mt Pinatubo) are excluded from the regression analysis.

[Figure]

**Figure 2.** Timeseries of  the fraction of sunrise to total (sunrise + sunset) profiles used to generate monthly mean ozone values in the tropics (30°S-30°N)  at  1 hPa for the SAGE II v7.0 vmr dataset.

[Figure]

**Figure 3.** Timeseries of percent tropical mean (30°S-30°N) ozone anomalies for 1984-2004 at (a) 1 hPa (48 km), (b) 3 hPa (40 km), (c) 5 hPa (36 km), (d) 10 hPa (31 km), and (e) 30 hPa (24 km). Data are shown for SAGE II v6.2 volume mixing ratios (vmr) (black), SAGE II v7.0 vmr (red), SAGE II v6.2 number densities (nd) (blue), and SAGE II v7.0 nd (green). The thick red lines denote the periods excluded from the MLR analysis following major tropical volcanic eruptions. The bottom panel shows the F10.7cm solar flux for reference.

[Figure]

$$O_3 \text{ [\%] } S_{max} - S_{min}$$

**Figure 4.**  (%) annual solar-ozone response (SOR)  (per 130 SFU) for the (a, (10hPa SAGE II v6.2 data and (b, e) SAGE II v7.0 data in terms of (a, b) number density-altitude units and (d, e)  volume mixing  ratio-pressure units. Panel (c) shows (b)  minus (a),  and panel (f) shows (e)  minus (d). The contour interval is 1%. The hatching denotes regions where the SOR is not statistically distinguishable from zero at the 95% confidence level.

[Figure]

**Figure 5.** The 95% confidence intervals (CI$_{95\%}$)  on the SORs (SOR$\pm$CI$_{95\%}$) shown in  Figure 4 for the (a, c) SAGE II v6.2 data and (b, d) SAGE II v7.0 data in terms of (a, b) number density-altitude units and (c, d) volume mixing ratio-pressure units. The contour interval is 0.5%. The hatching is as in Figure 4.

[Figure]

**Figure 6.** Timeseries of tropical mean temperature anomalies  from the NMC/NCEP (dashed) and  MERRA-1 (solid) datasets for (top-to-bottom) 1, 2, 5, 10, 30 hPa, respectively. The time period is 1979-2013. The thick red lines denote the periods excluded from the MLR analysis following major volcanic eruptions. The bottom panel shows the F10.7cm solar flux for reference.

[Figure]

**Figure 7.**  11 year solar cycle signals in temperature (K) from the (a)  MERRA-1 and (b) NMC/NCEP datasets. Shading as in Figure 4. The contour interval is 0.25 K. These temperature fields are used in the  'post-hoc' conversion of SAGE II v6.2  number  densities to mixing  ratios (see Section  4.1.2 for details).

[Figure]

**Figure 8.** The  percent (%)  annual solar-ozone response (SOR) (per 130 SFU) in SAGE II v6.2 data  onverted from number  densities to mixing  ratios for the period 1985-2003 using the method described in Section 4.1.2. The  conversion is first conducted using full  timeseries of monthly (a) NMC/NCEP and (b)  MERRA-1 temperatures. Panel (c) (b) minus (a). A comparison of  panels (a-c) with Figures 4(a-c)  demonstrates the performance of  the 'post-hoc' conversion (d-f)  As in (a-c) but with the  number density to mixing ratio conversion performed using a monthly temperature climatology from  MERRA-1 added to a linear trend and solar signal in stratospheric temperatures extracted from (d) NMC/NCEP and (e) MERRA-1. The remaining rows show the same as (d-f) but  with the  conversion  performed with the (g-i) linear trend or (j-l) solar cycle  temperature  terms alone. The shading is as is Figure 4. The contour interval is 1% in the left and middle columns and 0.5% in the right-hand column.

[Figure]

**Figure 9.** As in Figure 3, but for the three extended SAGE II number density datasets: SAGE-GOMOS 1 (black), SAGE-GOMOS 2 (orange), and SAGE-OSIRIS (green). The  time period is 1984-2013.

[Figure]

**Figure 10.** (a-c)  As in  Figure 4, but for the extended SAGE II number density datasets: (a) SAGE-GOMOS 1, (b) SAGE-GOMOS 2, (c) SAGE-OSIRIS. SORs are derived  for different periods as stated in the headers. The contour interval is 1%.  (d-f) As in Figure 5, but for the datasets as shown in (a-c). The contour interval is 0.5%.

[Figure]

**Figure 11.** As in Figure 3, but for the  SBUVMOD VN8.0 (black), SBUVMOD VN8.6 (red), and SBUV Merged Cohesive VN8.6 (blue) datasets. The time period is 1970-2015.

[Figure]

**Figure 12.** The annual percent () differences As in ozone per 130SFU Figure 10 for the (a,d) SBUV SBUVMOD VN8.0(McPeters et al., 1994) , (b,e) SBUVMOD VN8.6dataset (McPeters et al., 2013; Frith et al., 2014) , and (c,f) SBUV Merged Cohesive VN8.6 datasets(Wild and Long, 2015) . Signals SORs are derived for different periods as stated in the period 1984-2004 inclusiveheaders. The contour interval is 1. The hatching denotes regions that are not statistically significant at the 95confidence level.

[Figure]

SBUVMOD VN8.6 1970−2012

The monthly percent solar-ozone response (SOR) differences in ozone (per 130 SFU) in the SBUVMOD VN8.6 dataset for the period 1984-20041970-2012. The contour interval is 1%. The grey shading denotes regions that are where the SOR is not statistically significant distinguishable from zero at the 95% confidence level.

The monthly percent solar-ozone response (SOR) differences in ozone (per 130 SFU) in the SBUVMOD VN8.6 dataset for the period 1984-20041970-2012. The contour interval is 1%. The grey shading denotes regions that are where the SOR is not statistically significant distinguishable from zero at the 95% confidence level.

**Figure 13.** (a-f) The percent (%) differences in ozone per 130SFU in the SBUVMOD VN8.6 dataset for 21 year periods separated by 2 year intervals covering 1979-2009. Panel (g) shows the result for the full 1979-2009 period. The contour interval is 1. The hatching denotes regions that are not statistically significant at the 95confidence level.

The monthly percent solar-ozone response (SOR) differences in ozone (per 130 SFU) in the SBUVMOD VN8.6 dataset for the period 1984-20041970-2012. The contour interval is 1%. The grey shading denotes regions that are where the SOR is not statistically significant distinguishable from zero at the 95% confidence level.

| Dataset | Type | Time period | Units | Reference |
|---|---|---|---|---|
| SAGE II v6.2 | Raw satellite product: solar occultation instrument | 1984 - 2004 | ppmv/cm$^{-3}$ | Wang et al. (2002) |
|  SAGE II v7.0 | Raw satellite product: solar occultation instrument | 1984 - 2004 | ppmv/cm$^{-3}$ | Damadeo et al. (2013) |
| SAGE-GOMOS 1 | Combined satellite product, including SAGE II v7.0 | 1984 -  2011 | cm$^{-3}$ | Penckwitt et al. (2015) |
| SAGE-GOMOS 2 | Combined satellite product, including SAGE II v7.0 | 1984 -  2011 | cm$^{-3}$ | Kyrölä et al. (2015) |
| SAGE-OSIRIS | Combined satellite product, including SAGE II v7.0 | 1984 -  2013 | cm$^{-3}$ | Bourassa et al. (2014) |
|  SBUVMOD VN8.0 | Raw satellite product: nadir-viewing instrument |  1970 -  2009 | ppmv |  |
| SBUVMOD VN8.6 | Raw satellite product: nadir-viewing instrument |  1970 -  2012 | ppmv | McPeters et al. (2013); Frith et al. (2014) |
| SBUV Merged Cohesive VN8.6 | Raw satellite product: nadir-viewing instrument |  1978 -  2012 | ppmv | Wild and Long (2015) |
|  |  | | | |

**Table 1.**  Overview of the satellite ozone datasets used in this study.

---

## Referee Report (RR1)

I would like to thank the authors for having thoughtfully responded to each of my points raised during the review of the original manuscript.  I recommend this revised manuscript be published as is.  There is little doubt this paper will be frequently cited.  I found the quality of the writing, figures and inclusion of relevant references to be first rate.